# LEARNING WITH LOCAL SEARCH MCMC LAYERS

## ABSTRACT

Integrating combinatorial optimization layers into neural networks has recently attracted significant research interest. However, many existing approaches lack theoretical guarantees or fail to perform adequately when relying on inexact solvers. This is a critical limitation, as many operations research problems are NP-hard, often necessitating the use of neighborhood-based local search heuristics. In this paper, we introduce a theoretically-principled approach for learning with such inexact solvers. Inspired by the connection between simulated annealing and Metropolis-Hastings, we propose to transform problem-specific neighborhood systems used in local search heuristics into proposal distributions, implementing MCMC on the combinatorial space of feasible solutions. This allows us to construct differentiable, stochastic combinatorial layers and associated loss functions. Replacing an exact solver by a local search strongly reduces the computational burden of learning on many applications. We demonstrate our approach on a dynamic vehicle routing problem with time windows, a multi-dimensional knapsack problem, and on binary vector and k-subset prediction tasks.

## 1 INTRODUCTION

Models that combine neural networks and combinatorial optimization have recently attracted significant attention (Sadana et al., 2024; Mandi et al., 2024; Donti et al., 2017; Berthet et al., 2020; Bengio et al., 2020; Blondel and Roulet, 2024). They enrich combinatorial optimization algorithms with context-dependent features, making decisions more resilient to uncertainty. An important subset of this line of research integrates, within a neural network, a linear programming layer of the form:

$$\theta \mapsto \operatorname*{argmax}_{y \in \mathcal{Y}} \langle \theta, y \rangle \subseteq \operatorname*{argmax}_{y \in \operatorname{conv}(\mathcal{Y})} \langle \theta, y \rangle, \tag{1}$$

where $\mathcal{Y}$ is a finite set of feasible outputs. In the graphical models and structured prediction literature, Eq. (1) is known as the *maximum a posteriori* (MAP) problem (Wainwright and Jordan, 2008). Such layers enable the transformation of learned, continuous latent representations into structured, discrete outputs, that satisfy complex constraints. This setting is known as *decision-focused learning* (DFL), where a fixed solver is parameterized by $\theta$, predicted from features $x$, in contrast to *neural combinatorial optimization* (NCO), which aims to replace the solver entirely with ML-based heuristics.

The main challenge, however, lies in end-to-end training: as piecewise-constant functions, such layers lack meaningful gradients. Many relaxations and loss functions have been proposed for this setting; see Section 2 for a review. Table 1 contrasts them based on the type of oracle they assume access to. Some rely on an oracle for a regularized version of Eq. (1), while others use a solver for the original problem (i.e., a MAP oracle), performing multiple calls per instance for smoothing reasons. However, theoretical guarantees for these approaches typically assume exact solutions.

Unfortunately, many problems in operations research are NP-hard in nature, making exact oracles impractical. Instead, applications often rely on *local search heuristics* (e.g., simulated annealing), which iteratively generate and then accept or reject a neighbor of the current solution. We aim to provide a principled approach for learning with such inexact combinatorial solvers. This is crucial for exploiting popular heuristics from the operations research literature as layers in neural networks.

To do so, we open the solver "black box", by bridging local search heuristics and Markov chain Monte-Carlo (MCMC) methods. These lines of research have evolved quite separately, and their links remain unexploited for designing principled combinatorial optimization layers.

Table 1: The proposed approach leverages the neighborhood systems used by local search heuristics (inexact solvers) to obtain a differentiable combinatorial layer when usual oracles are not available.

| | Regularization | Oracle | Approach |
|---|---|---|---|
| Differentiable DP (2009; 2018) | Entropy | Exact marginal | DP |
| SparseMAP (2018) | Quadratic | Exact MAP | Frank-Wolfe |
| Barrier FW (2015) | TRW Entropy | Exact MAP | Frank-Wolfe |
| IntOpt (2020) | Log barrier | Interior point solver | Primal-Dual |
| Perturbed optimizers (2020) | Implicit via noise | Exact MAP | Monte-Carlo |
| DYS-net (2024) | Quadratic | Projection oracles | Davis-Yin Splitting |
| Blackbox solvers (2020) | None | Exact MAP | Interpolation |
| Contrastive divergences (2000) | Entropy | Gibbs / Langevin sampler | MCMC |
| Proposed | Entropy | Local search | MCMC |

We make the following contributions:

- We integrate local search heuristics as differentiable, stochastic layers in neural networks, by converting their neighborhood systems to proposal distributions, turning the local search oracle into a discrete MCMC sampler over the combinatorial set of solutions.

- We extend our framework to handle local search heuristics that leverage a diversity of neighborhood systems, enabling this class of powerful solvers to be used as a unified MCMC sampler.

- We show that the proposed layer yields stochastic gradients of a Fenchel-Young loss (Blondel et al., 2020) (even with a single MCMC iteration), leading to principled learning algorithms for conditional and unconditional settings, for which we provide a convergence analysis.

- The proposed layer reduces the computational bottleneck, especially with few MCMC iterations, enabling larger training instances and better generalization at scale (Parmentier, 2021; 2022).

- We demonstrate our approach on the EURO Meets NeurIPS 2022 challenge (Kool et al., 2023), a large-scale dynamic vehicle routing problem with time windows, and on binary vector prediction tasks. Abundant additional experiments are included in Section A of the appendix.

## 2 BACKGROUND AND RELATED WORK

### 2.1 PROBLEM SETUP

In this paper, our goal is to learn models that incorporate optimization layers of the form:

$$\widehat{\boldsymbol{y}} : \boldsymbol{\theta} \mapsto \operatorname*{argmax}_{\boldsymbol{y} \in \mathcal{Y}} \langle \boldsymbol{\theta}, \, \boldsymbol{y} \rangle + \varphi(\boldsymbol{y}), \tag{2}$$

where $\mathcal{Y} \subset \mathbb{R}^d$ is a finite but combinatorially-large set, and $\varphi$ encodes structural costs or preferences on outputs (e.g., routing distances, fixed costs) that do not depend on $\boldsymbol{\theta}$ (not to be confused with a regularization term). This formulation therefore extends the standard linear objective in Eq. (1) by allowing additional problem-specific structure.

We focus on settings where Eq. (2) is intractable and only heuristic algorithms are available to obtain an approximate solution. Our goal is to integrate NP-hard problems arising in operations research (e.g., routing, scheduling, network design), within a neural network. Unfortunately, many existing approaches lack formal guarantees or simply do not work when used with inexact solvers.

We distinguish between two settings. In the unconditional setting, our goal will be to learn $\boldsymbol{\theta} \in \mathbb{R}^d$ from observations $\boldsymbol{y}_1, \ldots, \boldsymbol{y}_N \in \mathcal{Y}$. In the conditional setting, we will assume that $\boldsymbol{\theta} = g_W(\boldsymbol{x})$ and our goal will be to learn the parameters $W$ from observation pairs $(\boldsymbol{x}_1, \boldsymbol{y}_1), \ldots, (\boldsymbol{x}_N, \boldsymbol{y}_N)$.

### 2.2 COMBINATORIAL OPTIMIZATION AS A LAYER

Since the layer defined in Eq. (1) is piecewise constant, a frequent strategy consists in introducing regularization in the problem so as to obtain a continuous relaxation. In some cases, we may have access to an oracle for directly solving the regularized problem. For instance, dynamic programming solvers can handle entropic regularization through a change of semi-ring (Li and Eisner, 2009) or algorithmic smoothing (Mensch and Blondel, 2018). As another example, interior point solvers can be used to compute a logarithmic barrier regularized solution (Mandi and Guns, 2020). More recently, McKenzie et al. (2024) handle quadratic regularization by leveraging projection oracles.

We focus on settings where only a MAP oracle is available for the original, unregularized optimization problem. While prior work is often limited to the linear form in Eq. (1) for the latter, our framework also handles the more general Eq. (2). Frank-Wolfe-like methods can be used to solve the regularized problem using only MAP oracle calls (Niculae et al., 2018; Krishnan et al., 2015). Another strategy consists in injecting noise perturbations (Berthet et al., 2020) in the oracle, which can be shown to be implicitly using regularization. In both cases, a Fenchel-Young loss can be associated, enabling principled learning. However, formal guarantees require an exact oracle, often called multiple times during the forward pass. Our proposal enjoys guarantees even with inexact solvers and a single call.

Regarding differentiation, several strategies are possible. When the approach only needs to differentiate through a (regularized) $\max$, as is the case of Fenchel-Young losses, we can use Danskin's theorem (Danskin, 1966). When the approach needs to differentiate a (regularized) $\arg\max$, we can either use autodiff on the unrolled solver iterations or implicit differentiation (Amos and Kolter, 2017; Agrawal et al., 2019; Blondel et al., 2022). Differently, Vlastelica et al. (2020) propose to compute gradients via continuous interpolation of the solver.

### 2.3 CONTRASTIVE DIVERGENCES

An alternative approach to learning in combinatorial spaces is to use energy-based models (EBMs) (Lecun et al., 2006), which define a distribution over outputs via a parameterized energy function $E_{\boldsymbol{\theta}}$:

$$\boldsymbol{p_\theta}(\boldsymbol{y}) \propto \exp(E_{\boldsymbol{\theta}}(\boldsymbol{y})), \quad \text{with} \quad \nabla_{\boldsymbol{\theta}} \log \boldsymbol{p_\theta}(\boldsymbol{y}) = \nabla_{\boldsymbol{\theta}} E_{\boldsymbol{\theta}}(\boldsymbol{y}) - \mathbb{E}_{Y \sim \boldsymbol{p_\theta}} \left[ \nabla_{\boldsymbol{\theta}} E_{\boldsymbol{\theta}}(Y) \right].$$

Therefore, we can perform maximum likelihood estimation (MLE) if we can sample from $\boldsymbol{p_\theta}$, but this is hard both in continuous and combinatorial settings, due to its intractable normalization constant. Contrastive divergences (Hinton, 2000; Carreira-Perpiñán and Hinton, 2005; Song and Kingma, 2021) address this by using MCMC to obtain (biased) stochastic gradients. Originally developed for restricted Boltzmann machines with $\mathcal{Y} = \{0, 1\}^d$ and a Gibbs sampler, they have also been applied in continuous domains via Langevin dynamics (Du and Mordatch, 2020; Du et al., 2021).

**MCMC in discrete spaces.** Contrastive divergences rely on MCMC to sample the model distribution. Unfortunately, designing MCMC samplers is often case-by-case, and discrete domains have received less attention than continuous ones. Recent efforts adapt continuous techniques, such as Langevin dynamics (Zhang et al., 2022; Sun et al., 2023a) or gradient-informed proposals (Grathwohl et al., 2021; Rhodes and Gutmann, 2022), to discrete settings. However, these works often assume simple state spaces (e.g., the hypercube or categorical codebooks), and do not handle complex constraints ubiquitous in operations research. Sun et al. (2023b) allow structured spaces via relaxed constraints in the energy function, yet ignore these structures in their proposal supports. Notably, we emphasize that all these works focus on sampling, not on designing differentiable MCMC layers.

## 3 LOCAL SEARCH-BASED MCMC LAYERS

This section introduces our core contribution. We first connect local search heuristics and MCMC methods, then use this link to define a stochastic layer based on a single neighborhood system (Algorithm 1), and subsequently generalize it to leverage diverse neighborhood systems (Algorithm 2).

### 3.1 FROM LOCAL SEARCH TO MCMC

**Local search and neighborhood systems.** Local search heuristics (Gendreau et al., 2010) iteratively generate a neighbor $\boldsymbol{y}' \in \mathcal{N}(\boldsymbol{y}^{(k)})$ of the current solution $\boldsymbol{y}^{(k)}$, and either accept it or reject it based on an acceptance rule, that depends on the objective function, $\boldsymbol{y}^{(k)}$ and $\boldsymbol{y}'$. In this context, a neighborhood system $\mathcal{N}$ defines, for each feasible solution $\boldsymbol{y} \in \mathcal{Y}$, a set of neighbors $\mathcal{N}(\boldsymbol{y}) \subseteq \mathcal{Y}$.

Neighborhoods are problem-specific, and must respect the structure of the problem, i.e., must maintain solution feasibility. They are typically defined implicitly via a set of allowed *moves* from $\boldsymbol{y}$. For instance, Table 2 lists example moves for a vehicle routing problem.

Formally, we denote the neighborhood graph by $G_{\mathcal{N}} := (\mathcal{Y}, E_{\mathcal{N}})$, where edges are defined by $\mathcal{N}$. We assume the graph is undirected, i.e., $\boldsymbol{y}' \in \mathcal{N}(\boldsymbol{y})$ if and only if $\boldsymbol{y} \in \mathcal{N}(\boldsymbol{y}')$, and without self-loops – i.e., $\boldsymbol{y} \notin \mathcal{N}(\boldsymbol{y})$. A stochastic neighbor generating function is also provided, in the form of a proposal distribution $q(\boldsymbol{y}, \cdot)$ with support either equal to $\mathcal{N}(\boldsymbol{y})$ or $\mathcal{N}(\boldsymbol{y}) \cup \{\boldsymbol{y}\}$.

**Algorithm 1** SA / MH as a layer

**Inputs:** $\boldsymbol{\theta} \in \mathbb{R}^d$, $\boldsymbol{y}^{(0)} \in \mathcal{Y}$, $(t_k)$, $K \in \mathbb{N}$, $\mathcal{N}$, $q$
**for** $k = 0 : K$ **do**
    Sample a neighbor in $\mathcal{N}(\boldsymbol{y}^{(k)})$:
    $\boldsymbol{y}' \sim q\left(\boldsymbol{y}^{(k)}, \cdot\right)$
    $\alpha(\boldsymbol{y}^{(k)}, \boldsymbol{y}') \leftarrow 1$ (SA) or
    $\alpha(\boldsymbol{y}^{(k)}, \boldsymbol{y}') \leftarrow \frac{q(\boldsymbol{y}', \boldsymbol{y}^{(k)})}{q(\boldsymbol{y}^{(k)}, \boldsymbol{y}')}$ (MH)
    $U \sim \mathcal{U}([0, 1])$
    $\Delta^{(k)} \leftarrow \langle \boldsymbol{\theta}, \boldsymbol{y}' \rangle + \varphi(\boldsymbol{y}') - \langle \boldsymbol{\theta}, \boldsymbol{y}^{(k)} \rangle - \varphi(\boldsymbol{y}^{(k)})$
    $p^{(k)} \leftarrow \alpha(\boldsymbol{y}^{(k)}, \boldsymbol{y}') \exp\left(\Delta^{(k)}/t_k\right)$
    If $U \leq p^{(k)}$, accept move: $\boldsymbol{y}^{(k+1)} \leftarrow \boldsymbol{y}'$
    If $U > p^{(k)}$, reject move: $\boldsymbol{y}^{(k+1)} \leftarrow \boldsymbol{y}^{(k)}$
**end for**
**Output:** $\widehat{\boldsymbol{y}}(\boldsymbol{\theta}) \approx \boldsymbol{y}^{(K)}$ (SA) or
$\widehat{\boldsymbol{y}}_t(\boldsymbol{\theta}) = \mathbb{E}_{\pi_{\boldsymbol{\theta}, t}}[Y] \approx \frac{1}{K} \sum_{k=1}^{K} \boldsymbol{y}^{(k)}$ (MH)

**Algorithm 2** Neighborhood mixture MCMC

**Inputs:** $\boldsymbol{\theta} \in \mathbb{R}^d$, $\boldsymbol{y}^{(0)} \in \mathcal{Y}$, $t$, $K \in \mathbb{N}$, $(\mathcal{N}_s, q_s)_{s=1}^{S}$
**for** $k = 0 : K$ **do**
    Sample a neighborhood system:
    $s \sim \mathcal{U}(Q(\boldsymbol{y}^{(k)}))$
    Sample a neighbor in $\mathcal{N}_s(\boldsymbol{y}^{(k)})$:
    $\boldsymbol{y}' \sim q_s(\boldsymbol{y}^{(k)}, \cdot)$
    $\alpha_s(\boldsymbol{y}^{(k)}, \boldsymbol{y}') \leftarrow \frac{|Q(\boldsymbol{y}^{(k)})|}{|Q(\boldsymbol{y}')|} \frac{q_s(\boldsymbol{y}', \boldsymbol{y}^{(k)})}{q_s(\boldsymbol{y}^{(k)}, \boldsymbol{y}')}$
    $U \sim \mathcal{U}([0, 1])$
    $\Delta^{(k)} \leftarrow \langle \boldsymbol{\theta}, \boldsymbol{y}' \rangle + \varphi(\boldsymbol{y}') - \langle \boldsymbol{\theta}, \boldsymbol{y}^{(k)} \rangle - \varphi(\boldsymbol{y}^{(k)})$
    $p^{(k)} \leftarrow \alpha_s(\boldsymbol{y}^{(k)}, \boldsymbol{y}') \exp\left(\Delta^{(k)}/t\right)$
    If $U \leq p^{(k)}$, accept move: $\boldsymbol{y}^{(k+1)} \leftarrow \boldsymbol{y}'$
    If $U > p^{(k)}$, reject move: $\boldsymbol{y}^{(k+1)} \leftarrow \boldsymbol{y}^{(k)}$
**end for**
**Output:** $\widehat{\boldsymbol{y}}_t(\boldsymbol{\theta}) = \mathbb{E}_{\pi_{\boldsymbol{\theta}, t}}[Y] \approx \frac{1}{K} \sum_{k=1}^{K} \boldsymbol{y}^{(k)}$

**Link between simulated annealing and Metropolis-Hastings.** A well-known example of local search heuristic is simulated annealing (SA) (Kirkpatrick et al., 1983). It is intimately related to Metropolis-Hastings (MH) (Hastings, 1970), an instance of a MCMC algorithm. We provide a unified view of both in Algorithm 1.

The difference lies in the acceptance rule, which incorporates a proposal correction ratio for MH, and in the choice of the sequence of temperatures $(t_k)_{k \in \mathbb{N}}$. In the case of SA, it is chosen to verify $t_k \to 0$. In the case of MH, it is such that $t_k \equiv t$. In this case, the iterates $\boldsymbol{y}^{(k)}$ of Algorithm 1 follow a time-homogenous Markov chain on $\mathcal{Y}$, defined by the following transition kernel:

$$P_{\boldsymbol{\theta}, t}(\boldsymbol{y}, \boldsymbol{y}') = \begin{cases} q(\boldsymbol{y}, \boldsymbol{y}') \min\left[1, \frac{q(\boldsymbol{y}', \boldsymbol{y})}{q(\boldsymbol{y}, \boldsymbol{y}')} \exp\left(\frac{\langle \boldsymbol{\theta}, \boldsymbol{y}' \rangle + \varphi(\boldsymbol{y}') - \langle \boldsymbol{\theta}, \boldsymbol{y} \rangle - \varphi(\boldsymbol{y})}{t}\right)\right] & \text{if } \boldsymbol{y}' \in \mathcal{N}(\boldsymbol{y}), \\ 1 - \sum_{\boldsymbol{y}'' \in \mathcal{N}(\boldsymbol{y})} P_{\boldsymbol{\theta}, t_k}(\boldsymbol{y}, \boldsymbol{y}'') & \text{if } \boldsymbol{y}' = \boldsymbol{y}, \\ 0 & \text{else.} \end{cases} \quad (3)$$

In past work, the link between the two algorithms has primarily been used to show that SA converges to the exact MAP solution in the limit of infinite iterations (Mitra et al., 1986; Faigle and Schrader, 1988). Under mild conditions – if the neighborhood graph $G_{\mathcal{N}}$ is connected and the chain is aperiodic, the iterates $(\boldsymbol{y}^{(k)})_{k \in \mathbb{N}}$ of Algorithm 1 (MH case) converge in distribution to the Gibbs distribution (see Section E.1 for a proof):

$$\pi_{\boldsymbol{\theta}, t}(\boldsymbol{y}) \propto \exp\left([\langle \boldsymbol{\theta}, \boldsymbol{y} \rangle + \varphi(\boldsymbol{y})]/t\right). \quad (4)$$

**Proposed layer.** Algorithm 1 and this result motivate us to define the combinatorial MCMC layer

$$\widehat{\boldsymbol{y}}_t(\boldsymbol{\theta}) := \mathbb{E}_{\pi_{\boldsymbol{\theta}, t}}[Y], \quad (5)$$

where $\boldsymbol{\theta} \in \mathbb{R}^d$ are logits and $t > 0$ is a temperature parameter, defaulting to $t = 1$. Computing $\widehat{\boldsymbol{y}}_t(\boldsymbol{\theta})$ is known as the *marginal inference* problem in the graphical models literature. Naturally, the estimate of $\widehat{\boldsymbol{y}}_t(\boldsymbol{\theta})$ returned by Algorithm 1 (MH case) is biased, as the Markov chain cannot perfectly mix in a finite number of iterations, except if it is initialized at $\pi_{\boldsymbol{\theta}, t}$. In Section 4, we will show that this does not hinder the convergence of the proposed learning algorithms. The next proposition, proved in Section E.2, states some useful properties of the proposed layer.

**Proposition 1.** *Let $\boldsymbol{\theta} \in \mathbb{R}^d$. We have the following properties:*

$$\widehat{\boldsymbol{y}}_t(\boldsymbol{\theta}) \in \mathsf{relint}(\mathcal{C}), \quad \widehat{\boldsymbol{y}}_t(\boldsymbol{\theta}) \xrightarrow[t \to 0^+]{} \underset{\boldsymbol{y} \in \mathcal{Y}}{\mathrm{argmax}} \langle \boldsymbol{\theta}, \boldsymbol{y} \rangle + \varphi(\boldsymbol{y}), \quad \text{and} \quad \widehat{\boldsymbol{y}}_t(\boldsymbol{\theta}) \xrightarrow[t \to \infty]{} \frac{1}{|\mathcal{Y}|} \sum_{\boldsymbol{y} \in \mathcal{Y}} \boldsymbol{y}.$$

*Moreover, $\widehat{\boldsymbol{y}}_t$ is differentiable and its Jacobian matrix is given by $J_{\boldsymbol{\theta}} \widehat{\boldsymbol{y}}_t(\boldsymbol{\theta}) = \frac{1}{t} \mathsf{cov}_{\pi_{\boldsymbol{\theta}, t}}[Y]$.*

## 3.2 Mixing neighborhood systems

Central to local search algorithms in combinatorial optimization is the use of multiple neighborhood systems to more effectively explore the solution space (Mladenović and Hansen, 1997; Blum and Roli, 2003). In this section, we propose a tractable way to incorporate such diversity of neighborhood systems into the combinatorial MCMC layer, while preserving the correct stationary distribution.

**Definitions.** Let $(\mathcal{N}_s)_{s=1}^{S}$ be a set of different neighborhood systems. Typically, all neighborhood systems are not defined on all solutions $\boldsymbol{y} \in \mathcal{Y}$, so we note $Q(\boldsymbol{y}) \subseteq [\![1, S]\!]$ the set of neighborhood systems defined on $\boldsymbol{y}$ (i.e., the set of allowed moves on $\boldsymbol{y}$). Let $(q_s)_{s \in Q(\boldsymbol{y})}$ be the corresponding proposal distributions, such that the support of $q_s(\boldsymbol{y}, \cdot)$ is either $\mathcal{N}_s(\boldsymbol{y})$ or $\mathcal{N}_s(\boldsymbol{y}) \cup \{\boldsymbol{y}\}$. Let $\bar{\mathcal{N}}$ be the aggregate neighborhood system defined by $\bar{\mathcal{N}} : \boldsymbol{y} \mapsto \bigcup_{s \in Q(\boldsymbol{y})} \mathcal{N}_s(\boldsymbol{y})$.

**Computational challenge of neighborhood mixing.** A standard way to combine these neighborhood systems would be to use Algorithm 1 by defining an aggregated proposal $q(\boldsymbol{y}, \cdot)$ as, e.g.:

$$q(\boldsymbol{y}, \boldsymbol{y}') := \frac{1}{|Q(\boldsymbol{y})|} \sum_{s \in Q(\boldsymbol{y})} q_s(\boldsymbol{y}, \boldsymbol{y}'), \quad \text{giving:} \quad \alpha(\boldsymbol{y}, \boldsymbol{y}') = \frac{|Q(\boldsymbol{y})|}{|Q(\boldsymbol{y}')|} \cdot \frac{\sum_{s \in Q(\boldsymbol{y}')} q_s(\boldsymbol{y}', \boldsymbol{y})}{\sum_{s \in Q(\boldsymbol{y})} q_s(\boldsymbol{y}, \boldsymbol{y}')}.$$

However, this leads to intractable updates. Indeed, computing the correction ratio $\alpha(\boldsymbol{y}, \boldsymbol{y}')$ is prohibitively expensive as it involves summing the forward proposal probabilities for all move types in $Q(\boldsymbol{y})$ and the reverse probabilities for all move types in $Q(\boldsymbol{y}')$. The difficulty is that multiple, distinct proposal types can generate the same solution $\boldsymbol{y}'$ from $\boldsymbol{y}$. For example, in our vehicle routing application in Section 5.1, relocating a pair of clients before the first one in a route of three gives the same solution $\boldsymbol{y}'$ as relocating the first client at the last position (see the `relocate` and `relocate pair` moves from Table 2). Identifying and calculating all these potential forward and reverse pathways for every step is a significant computational hurdle.

**Proposed efficient sampler.** In contrast, the update we propose in Algorithm 2 circumvents this summation entirely by sampling the move type $s$ first. It only requires computing the single individual ratio $\alpha_s(\boldsymbol{y}, \boldsymbol{y}') := \frac{|Q(\boldsymbol{y})|}{|Q(\boldsymbol{y}')|} \cdot \frac{q_s(\boldsymbol{y}', \boldsymbol{y})}{q_s(\boldsymbol{y}, \boldsymbol{y}')}$ for the unique move type $s$ that was actually sampled.

> **Proposition 2.** *If each neighborhood graph $G_{\mathcal{N}_s}$ is undirected and without self-loops, and the aggregate neighborhood graph $G_{\bar{\mathcal{N}}}$ is connected, the iterations $(\boldsymbol{y}^{(k)})_{k \in \mathbb{N}}$ produced by Algorithm 2 follow a Markov chain with unique stationary distribution $\pi_{\boldsymbol{\theta}, t}$.*

See Section E.3 for the proof. Importantly, our method is not an approximation: it targets the exact same stationary distribution as the naive approach, but does so efficiently. Furthermore, only the *aggregate* neighborhood graph $G_{\bar{\mathcal{N}}}$ is required to be connected. This enables combining neighborhood systems $\mathcal{N}_s$ that could not connect $\mathcal{Y}$ if used individually, and an irreducible Markov chain can be obtained by mixing the proposal distributions of reducible ones. As a concrete example, the moves used as proposals in our dynamic vehicle routing experiment (Section 5.1) are defined in Table 2.

## 4 Loss functions and theoretical analysis

Building upon the differentiable MCMC layer developed in Section 3, this section constructs the corresponding learning framework. We derive principled Fenchel-Young loss functions for our layer, present practical stochastic gradient algorithms for both conditional and unconditional learning, and provide theoretical convergence guarantees for these algorithms.

### 4.1 Negative log-likelihood and associated Fenchel-Young loss

We now show that the proposed layer $\widehat{\boldsymbol{y}}_t(\boldsymbol{\theta})$ can be viewed as the solution of a regularized optimization problem on $\mathcal{C} = \text{conv}(\mathcal{Y})$. Let $A_t(\boldsymbol{\theta}) := t \cdot \log \sum_{\boldsymbol{y} \in \mathcal{Y}} \exp\left([\langle \boldsymbol{\theta}, \boldsymbol{y} \rangle + \varphi(\boldsymbol{y})]/t\right)$ be the cumulant function (Wainwright and Jordan, 2008) associated to $\pi_{\boldsymbol{\theta}, t}$, scaled by $t$. We define the regularization function $\Omega_t$ and the corresponding Fenchel-Young loss (Blondel et al., 2020) as:

$$\Omega_t(\boldsymbol{\mu}) := A_t^*(\boldsymbol{\mu}) = \sup_{\boldsymbol{\theta} \in \mathbb{R}^d} \langle \boldsymbol{\mu}, \boldsymbol{\theta} \rangle - A_t(\boldsymbol{\theta}), \quad \text{and} \quad \ell_t(\boldsymbol{\theta}; \boldsymbol{y}) := (\Omega_t)^*(\boldsymbol{\theta}) + \Omega_t(\boldsymbol{y}) - \langle \boldsymbol{\theta}, \boldsymbol{y} \rangle.$$

Since $\Omega_t = A_t^*$ is strictly convex on $\mathrm{relint}(\mathcal{C})$ (see Section E.4 for a proof) and $\widehat{\boldsymbol{y}}_t(\boldsymbol{\theta}) = \nabla_{\boldsymbol{\theta}} A_t(\boldsymbol{\theta})$, the proposed layer is the solution of the regularized optimization problem

$$\widehat{\boldsymbol{y}}_t(\boldsymbol{\theta}) = \operatorname*{argmax}_{\boldsymbol{\mu} \in \mathcal{C}} \left\{ \langle \boldsymbol{\theta}, \boldsymbol{\mu} \rangle - \Omega_t(\boldsymbol{\mu}) \right\}, \tag{6}$$

the Fenchel-Young loss $\ell_t$ is differentiable, satisfies $\ell_t(\boldsymbol{\theta}, \boldsymbol{y}) = 0 \Leftrightarrow \widehat{\boldsymbol{y}}_t(\boldsymbol{\theta}) = \boldsymbol{y}$, and has gradient $\nabla_{\boldsymbol{\theta}} \ell_t(\boldsymbol{\theta} \,; \boldsymbol{y}) = \widehat{\boldsymbol{y}}_t(\boldsymbol{\theta}) - \boldsymbol{y}$ (Blondel et al., 2020). It is therefore equivalent, up to a constant, to the negative log-likelihood loss, as we have $-\nabla_{\boldsymbol{\theta}} \log \pi_{\boldsymbol{\theta},t}(\boldsymbol{y}) = (\widehat{\boldsymbol{y}}_t(\boldsymbol{\theta}) - \boldsymbol{y})/t$. Algorithms 1 and 2 can thus be used to perform MLE, by returning a (biased) stochastic estimate of the gradient of $\ell_t$.

## 4.2 Empirical risk minimization

In the conditional learning setting, we are given observations $(\boldsymbol{x}_i, \boldsymbol{y}_i)_{i=1}^N \in (\mathbb{R}^p \times \mathcal{Y})^N$, and want to fit a model $g_W : \mathbb{R}^p \to \mathbb{R}^d$ such that $\widehat{\boldsymbol{y}}_t(g_W(\boldsymbol{x}_i)) \approx \boldsymbol{y}_i$. This is motivated by a generative model where, for some weights $W_0 \in \mathbb{R}^p$, the data is generated with $\boldsymbol{y}_i \sim \pi_{g_{W_0}(\boldsymbol{x}_i),t}$. We aim at minimizing the empirical risk $L_N$, defined below along with its exact gradient:

$$L_N(W) := \frac{1}{N} \sum_{i=1}^N \ell_t\left(g_W(\boldsymbol{x}_i)\,; \boldsymbol{y}_i\right), \quad \text{with} \quad \nabla_W L_N(W) = \frac{1}{N} \sum_{i=1}^N J_W g_W(\boldsymbol{x}_i)\left(\widehat{\boldsymbol{y}}_t(g_W(\boldsymbol{x}_i)) - \boldsymbol{y}_i\right).$$

**Doubly stochastic gradient estimator.** In practice, we cannot compute the exact gradient above. Using Algorithm 1 or 2 to get a MCMC estimate of $\widehat{\boldsymbol{y}}_t(g_W(\boldsymbol{x}_i))$, we propose the following estimator:

$$\nabla_W L_N(W) \approx J_W g_W(\boldsymbol{x}_i)\left(\frac{1}{K}\sum_{k=1}^K \boldsymbol{y}_i^{(k)} - \boldsymbol{y}_i\right),$$

where $\boldsymbol{y}_i^{(k)}$ is the $k$-th iterate of the algorithm with maximization direction $\boldsymbol{\theta}_i = g_W(\boldsymbol{x}_i)$ and temperature $t$. This estimator is doubly stochastic, since we sample both data points and Markov iterations, and can be seamlessly used with batches. The term $J_W g_W(\boldsymbol{x}_i)$ is computed via autodiff.

**Markov chain initialization.** Following the contrastive divergence literature (Hinton, 2000), in the conditional setting, we initialize the Markov chains at the corresponding ground-truth, by setting $\boldsymbol{y}_i^{(0)} = \boldsymbol{y}_i$. In the unconditional setting, we use a persistent initialization (Tieleman, 2008) instead.

## 4.3 Associated Fenchel-Young loss with a single MCMC iteration

To obtain an unbiased gradient estimator for the Fenchel-Young loss $\ell_t$ associated with $\widehat{\boldsymbol{y}}_t$, the MCMC sampler must be run until it reaches its stationary distribution $\pi_{\boldsymbol{\theta},t}$. This requirement makes any practical estimator with a finite number of steps $K$ inherently biased.

Although our convergence analysis in Section 4.4 shows that this bias does not hinder the convergence of the proposed learning algorithms, we now demonstrate that when a single MCMC iteration is used ($K = 1$), there exists *another* target-dependent Fenchel-Young loss such that the gradient estimator is *unbiased* with respect to that loss. See Section E.7 for the construction of $\Omega_{\boldsymbol{y}}$ and the proof.

> **Proposition 3** (Existence of a Fenchel-Young loss when $K = 1$). *Let $\boldsymbol{p}_{\boldsymbol{\theta},\boldsymbol{y}}^{(1)}$ denote the distribution of the first iterate of the Markov chain defined by Eq. (3), with proposal distribution $q$ and initialized at ground-truth $\boldsymbol{y} \in \mathcal{Y}$. There exists a target-dependent regularization function $\Omega_{\boldsymbol{y}}$ with the following properties: $\Omega_{\boldsymbol{y}}$ is $t/\mathbb{E}_{q(\boldsymbol{y}, \cdot)}\|Y - \boldsymbol{y}\|_2^2$-strongly convex, it is such that:*
>
> $$\mathbb{E}_{\boldsymbol{p}_{\boldsymbol{\theta},\boldsymbol{y}}^{(1)}}[Y] = \operatorname*{argmax}_{\boldsymbol{\mu} \in \mathrm{conv}(\mathcal{N}(\boldsymbol{y}) \cup \{\boldsymbol{y}\})} \left\{ \langle \boldsymbol{\theta}, \boldsymbol{\mu} \rangle - \Omega_{\boldsymbol{y}}(\boldsymbol{\mu}) \right\},$$
>
> *and the Fenchel-Young loss $\ell_{\Omega_{\boldsymbol{y}}}$ generated by $\Omega_{\boldsymbol{y}}$ is $\mathbb{E}_{q(\boldsymbol{y}, \cdot)}\|Y - \boldsymbol{y}\|_2^2/t$-smooth in its first argument, and such that $\nabla_{\boldsymbol{\theta}} \ell_{\Omega_{\boldsymbol{y}}}(\boldsymbol{\theta} \,; \boldsymbol{y}) = \mathbb{E}_{\boldsymbol{p}_{\boldsymbol{\theta},\boldsymbol{y}}^{(1)}}[Y] - \boldsymbol{y}$.*

A similar result in the unconditional setting with data-based initialization is given in Proposition 6. In contrast, Sutskever and Tieleman (2010) showed that the expected CD-1 update with Gibbs sampling for restricted Boltzmann machines is not the gradient of any function, let alone a convex one.

Table 2: Local search moves used for creating neighborhoods in our vehicle routing experiments.

| Name | Description |
| --- | --- |
| relocate | Removes a single request from its route and re-inserts it at a different position in the solution. |
| relocate pair | Removes a pair of consecutive requests from their route and re-inserts them at a different position in the solution. |
| swap | Exchanges the position of two requests in the solution. |
| swap pair | Exchanges the positions of two pairs of consecutive requests in the solution. |
| 2-opt | Reverses a route segment. |
| serve request | Inserts a currently unserved request into the solution. |
| remove request | Removes a request from the solution. |

### 4.4 CONVERGENCE ANALYSIS IN THE UNCONDITIONAL SETTING

In the unconditional setting, we are given observations $(\boldsymbol{y}_i)_{i=1}^N \in \mathcal{Y}^N$ and want to fit a model $\pi_{\boldsymbol{\theta},t}$, motivated by an underlying generative model such that $\boldsymbol{y}_i \sim \pi_{\boldsymbol{\theta}_0,t}$ for some true parameter $\boldsymbol{\theta}_0$. We assume here that $\mathcal{C} = \text{conv}(\mathcal{Y})$ is of full dimension in $\mathbb{R}^d$ (if not, the model is specified only up to vectors $\boldsymbol{\mu}$ orthogonal to the affine subspace spanned by $\mathcal{C}$, as $\pi_{\boldsymbol{\theta}+\boldsymbol{\mu},t} = \pi_{\boldsymbol{\theta},t}$). We have the corresponding empirical $L_N$ and population $L_{\boldsymbol{\theta}_0}$ Fenchel-Young losses:

$$L_N(\boldsymbol{\theta}; \boldsymbol{y}_1, \ldots, \boldsymbol{y}_N) := \frac{1}{N} \sum_{i=1}^N \ell_t(\boldsymbol{\theta}; \boldsymbol{y}_i) , \quad L_{\boldsymbol{\theta}_0}(\boldsymbol{\theta}) := \mathbb{E}_{(\boldsymbol{y}_i)_{i=1}^N \sim (\pi_{\boldsymbol{\theta}_0,t})^{\otimes N}} [L_N(\boldsymbol{\theta}; \boldsymbol{y}_1, \ldots, \boldsymbol{y}_N)],$$

which are minimized for $\boldsymbol{\theta}$ such that $\widehat{\boldsymbol{y}}_t(\boldsymbol{\theta}) = \bar{Y}_N := \frac{1}{N} \sum_{i=1}^N \boldsymbol{y}_i$, and for $\boldsymbol{\theta}$ such that $\widehat{\boldsymbol{y}}_t(\boldsymbol{\theta}) = \widehat{\boldsymbol{y}}_t(\boldsymbol{\theta}_0)$, respectively. Let $\boldsymbol{\theta}_N^\star$ as the minimizer of the empirical loss $L_N$. For it to be defined, we assume that $\bar{Y}_N \in \text{int}(\mathcal{C})$ (which is always the case for $N$ large enough, as $\pi_{\boldsymbol{\theta}_0,t}$ has dense support on $\mathcal{Y}$). A slight variation on Proposition 4.1 in Berthet et al. (2020) gives the following asymptotic normality:

**Proposition 4** (Convergence of the empirical loss minimizer to the true parameter)**.**

$$\sqrt{N}(\boldsymbol{\theta}_N^\star - \boldsymbol{\theta}_0) \xrightarrow[N\to\infty]{\mathcal{D}} \mathcal{N}\left(\mathbf{0},\, t^2 \operatorname{cov}_{\pi_{\boldsymbol{\theta}_0,t}}[Y]^{-1}\right).$$

The proof is given in Section E.5. We now consider the sample size as fixed to $N$ samples, and define $\hat{\boldsymbol{\theta}}_n$ as the $n$-th iterate of the following stochastic gradient algorithm:

$$\hat{\boldsymbol{\theta}}_{n+1} = \hat{\boldsymbol{\theta}}_n + \gamma_{n+1} \left[ \bar{Y}_N - \frac{1}{K_{n+1}} \sum_{k=1}^{K_{n+1}} \boldsymbol{y}^{(n+1,k)} \right], \tag{7}$$

where $\boldsymbol{y}^{(n+1,k)}$ is the $k$-th iterate of Algorithm 1 with temperature $t$, maximization direction $\hat{\boldsymbol{\theta}}_n$, and initialized at $\boldsymbol{y}^{(n+1,1)} = \boldsymbol{y}^{(n,K_n)}$. This initialization corresponds to the persistent contrastive divergences (PCD) algorithm (Tieleman, 2008), and is further discussed in Section B.3.

**Proposition 5** (Convergence of the stochastic gradient estimate)**.** *Suppose the following hold for the step sizes $(\gamma_n)_{n\geq 1}$, sample sizes $(K_n)_{n\geq 1}$, and proposal distribution $q$:*

**(i)** $\gamma_n = an^{-b}$, *with* $b \in \left(\frac{1}{2}, 1\right]$ *and* $a > 0$.

**(ii)** $K_{n+1} > \lfloor 1 + a' \exp(\frac{8R_\mathcal{C}}{t}\|\hat{\boldsymbol{\theta}}_n\|) \rfloor$, *with* $a' > 0$ *and* $R_\mathcal{C} = \max_{\boldsymbol{y}\in\mathcal{Y}} \|\boldsymbol{y}\|$.

**(iii)** $\frac{1}{\sqrt{K_n}} - \frac{1}{\sqrt{K_{n-1}}} \leq a''n^{-c}$, *with* $a'' > 0$ *and* $c > 1 - \frac{b}{2}$.

**(iv)** $q(\boldsymbol{y}, \boldsymbol{y}') = \begin{cases} \frac{1}{2d^*}, & \boldsymbol{y}' \in \mathcal{N}(\boldsymbol{y}), \\ 1 - \frac{d(\boldsymbol{y})}{2d^*}, & \boldsymbol{y}' = \boldsymbol{y}, \\ 0, & else, \end{cases}$ *where* $d(\boldsymbol{y}) := |\mathcal{N}(\boldsymbol{y})|$ *and* $d^* := \max_{\boldsymbol{y}\in\mathcal{Y}} d(\boldsymbol{y})$.

*Then the iterates $\hat{\boldsymbol{\theta}}_n$ defined by Eq. (7) converge almost surely: $\hat{\boldsymbol{\theta}}_n \xrightarrow{a.s.} \boldsymbol{\theta}_N^\star$.*

See Section E.6 for the proof. The assumptions on $q$ are used for obtaining a closed-form convergence rate bound for the Markov chain, using graph-based geometric bounds (Ingrassia, 1994).

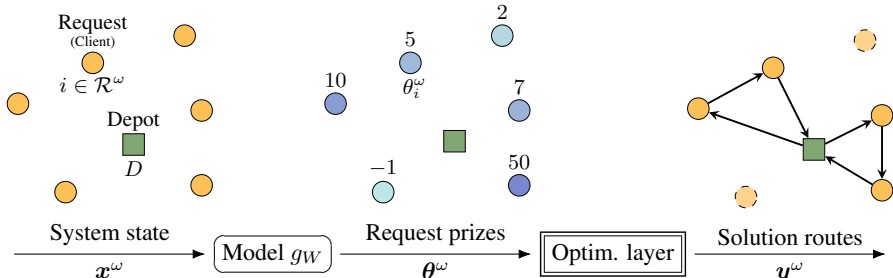

Figure 1: Overview of the vehicle routing pipeline, represented at request wave $\omega$.

## 5 NUMERICAL EXPERIMENTS

### 5.1 DYNAMIC VEHICLE ROUTING

We empirically validate our approach on the dynamic vehicle routing problem with time windows (DVRPTW) from the *EURO Meets NeurIPS 2022 Vehicle Routing Competition* (Kool et al., 2023). A detailed introduction to the challenge with precise notations is given in Section C.

**Reduction to supervised learning.** In this DVRPTW, requests arrive in delivery waves $\omega$, at the start of which a dispatching and vehicle routing problem for the current set of requests $\mathcal{R}^\omega$ must be solved, to get a feasible solution $\boldsymbol{y}^\omega \in \mathcal{Y}(\mathcal{R}^\omega)$. Following Baty et al. (2023), we frame each dispatching and routing problem as a prize-collecting (PC-)VRP, where a model $g_W$ predicts a "prize" vector $\boldsymbol{\theta}^\omega$ for serving each request. This PC-VRP fits the general formulation of Eq. (2):

$$\widehat{\boldsymbol{y}}(\boldsymbol{\theta}^\omega) = \underset{\boldsymbol{y} \in \mathcal{Y}(\mathcal{R}^\omega)}{\arg\max} \, \langle \boldsymbol{\theta}^\omega, \boldsymbol{y} \rangle + \varphi(\boldsymbol{y}), \tag{8}$$

where $\varphi(\boldsymbol{y}) := -\langle \boldsymbol{c}, \boldsymbol{y} \rangle$ is the negative routing cost. The overall pipeline is shown in Fig. 1. The model is trained to imitate an anticipative oracle $f^\star$, i.e., we use its output as ground-truth for supervised learning. We compute $f^\star$ by solving a static VRPTW where all future information in the instance is revealed from the start, turning dispatching waves into time windows.

**Approach and baseline.** The baseline Baty et al. (2023), winner of the competition, relies on a perturbation-based method (Berthet et al., 2020) with the state-of-the-art PC-HGS heuristic $\tilde{\boldsymbol{y}}$ (Vidal, 2022) as a combinatorial optimization layer. Since $\tilde{\boldsymbol{y}}$ is an inexact solver, the theoretical guarantees granted by the framework of Berthet et al. (2020) no longer hold. Our approach instead uses a local search MCMC layer to train $g_W$. We use a mixture of proposals (Algorithm 2) defined precisely in Section C.5, derived from the local search moves used by the PC-HGS solver itself (which are summarized in Table 2). At inference time, we follow the baseline, and use $f_W := \tilde{\boldsymbol{y}} \circ g_W$.

**Results.** We use the competition's metric: the routing cost over full instances with multiple dispatching waves, relative to the anticipative oracle $f^\star$. In Fig. 2, initializing the Markov chain with the ground-truth solution clearly outperforms a random start (even more so when refined by the fast initialization heuristic used by $\tilde{\boldsymbol{y}}$), and performance increases with the MCMC iteration number $K$.

In Table 3, we compare training methods under a fixed time budget for the layer's forward pass (the main computational bottleneck). We observe that our approach significantly outperforms the perturbation-based method in low time-limit regimes (1-100ms), thus enabling faster and more efficient training. Full experimental details and additional results are in Section C.8.

Table 3: Best test relative cost (%) w.r.t. $f^\star$ for different training methods and time limits.

| Time limit (ms) | 1 | 5 | 10 | 50 | 100 | 1000 |
|---|---|---|---|---|---|---|
| Perturbed inexact oracle | $65.2 \pm 5.8$ | $13.1 \pm 3.4$ | $8.7 \pm 1.9$ | $6.5 \pm 1.1$ | $6.3 \pm 0.76$ | $\mathbf{5.5} \pm 0.4$ |
| Proposed ($\boldsymbol{y}^{(0)} = \boldsymbol{y}$) | $10.0 \pm 1.7$ | $12.0 \pm 2.6$ | $11.8 \pm 2.8$ | $9.1 \pm 1.7$ | $8.4 \pm 1.7$ | $7.7 \pm 1.1$ |
| Proposed ($\boldsymbol{y}^{(0)} = $ heuristic($\boldsymbol{y}$)) | $\mathbf{7.8} \pm 0.8$ | $\mathbf{7.2} \pm 0.6$ | $\mathbf{6.3} \pm 0.7$ | $\mathbf{6.2} \pm 0.8$ | $\mathbf{5.9} \pm 0.7$ | $5.9 \pm 0.6$ |

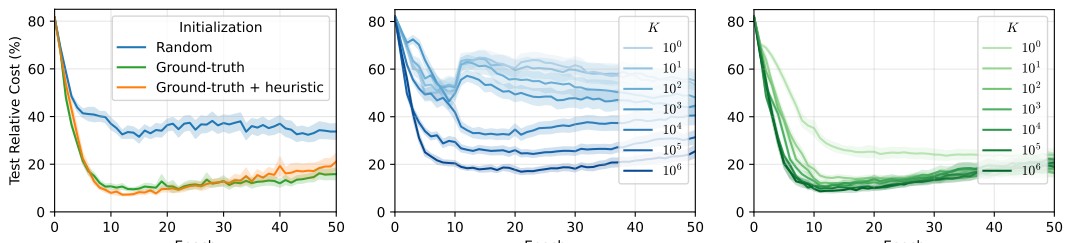

Figure 2: Test relative cost (%) w.r.t. $f^\star$. **Left**: varying initialization method. **Center**: varying number of Markov iterations $K$, random initialization. **Right**: varying $K$, ground-truth initialization.

## 5.2 MULTI-DIMENSIONAL KNAPSACK PROBLEM

In this section, we evaluate our framework on the multi-dimensional knapsack problem (MKP) (Martello and Toth, 1990; Kellerer et al., 2004), which is a resource allocation problem involving subset selection under multiple constraints. We benchmark our method against a broader landscape of differentiable optimization baselines provided by the PyEPO library (Tang and Khalil, 2023). Experimental and methodological details are given in Section D.

**Problem formulation.** We consider the decision-focused learning setup where the goal is to select a subset of items to maximize a total value while respecting $M$ capacity constraints. The item values $\boldsymbol{\theta}$ are predicted from features $\boldsymbol{x}$. Formally, the combinatorial optimization problem is defined as:

$$\widehat{\boldsymbol{y}}(\boldsymbol{\theta}) := \underset{\boldsymbol{y} \in \{0,1\}^d}{\operatorname{argmax}} \sum_{i=1}^{d} \theta_i y_i \quad = \underset{\boldsymbol{y} \in \mathcal{Y}}{\operatorname{argmax}} \langle \boldsymbol{\theta}, \boldsymbol{y} \rangle, \tag{9}$$

$$\text{s.t. } \forall j \in [M], \ \sum_{i=1}^{d} w_{i,j} y_i \leq C_j$$

where $\boldsymbol{\theta} = g_W(\boldsymbol{x}) \in \mathbb{R}^d$ are the item values, $w_{i,j} \geq 0$ is the weight of item $i$ in dimension $j$, and $C_j$ is the capacity of dimension $j$. The feasible set is $\mathcal{Y} := \{\boldsymbol{y} \in \{0,1\}^d \mid \forall j \in [M], \ \sum_{i=1}^{d} w_{i,j} y_i \leq C_j\}$. We are given a training set $(\boldsymbol{x}_i, \boldsymbol{y}_i)_{i=1}^{N}$ (the SPO+ baseline also requires access to the true values $\boldsymbol{\theta}_i$). At test time, given only $\boldsymbol{x}$, the goal is to predict $\boldsymbol{y}$ with minimal regret compared to the ground-truth.

**Proposed layer.** For our Local Search-MCMC layer $\widehat{\boldsymbol{y}}_t$, we use Algorithm 2 with ground-truth initialization, temperature $t = 1.0$, and a mixture of three proposal distributions, detailed in Section D.1.

**Baselines.** We compare against four established decision-focused learning methods from the PyEPO library: smart predict-then-optimize (SPO+, Elmachtoub and Grigas (2020)), perturbed optimizers using $K = 5$ Monte-Carlo samples (PFY, Berthet et al. (2020)), negative identity backpropagation (NID, Sahoo et al. (2023)), and noise-contrastive estimation (NCE, Mulamba et al. (2021)).

**Compute and performance benchmark.** We generate a dataset $(\boldsymbol{x}_i, \boldsymbol{y}_i)_{i=1}^{N}$ using PyEPO with $N = 2000$, $d = 100$ and $J = 50$ (we also give $\boldsymbol{\theta}_i$ to the SPO+ loss). Our approach achieves competitive test relative regret (Fig. 3, left) while drastically reducing the computational burden (Fig. 3, center). The variance of the LS-MCMC gradients remains consistently lower than that of other methods (Fig. 3, right), showing that the proposed method provides a stable signal for learning.

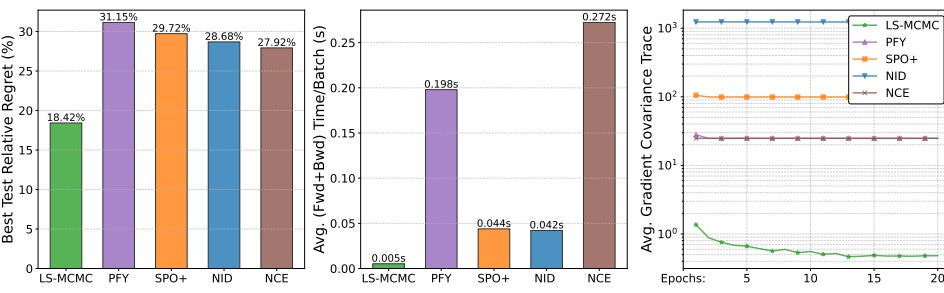

Figure 3: Benchmark results on the MKP.

### 5.3 LEARNING TO PREDICT BINARY VECTORS

**Setup.** To further validate the proposed gradient estimators, we use a synthetic unconditional learning task with hypercube output space, $\mathcal{Y} = \{0,1\}^d$. This setting is ideal for controlled experiments because the Gibbs distribution $\pi_{\boldsymbol{\theta},t}$ is fully factorized, leading to trivial sampling and a tractable closed-form expectation $\mathbb{E}_{\pi_{\boldsymbol{\theta},t}}[Y] = \sigma(\boldsymbol{\theta}/t)$, where $\sigma$ is the logistic sigmoid function. This allows us to both faithfully generate datasets from a known distribution $\pi_{\boldsymbol{\theta},t}$, and to minimize the population Fenchel-Young loss $L_{\boldsymbol{\theta}}$ directly (see Section 4.4 for its definition). The latter lets us decouple the noise from our MCMC estimator from the statistical noise inherent in finite datasets.

In all experiments, the goal is to recover a known "true" parameter vector $\boldsymbol{\theta}_0$ from independent samples $(\boldsymbol{y}_i)_{i=1}^N \sim (\pi_{\boldsymbol{\theta}_0,t})^{\otimes N}$. We summarize our key findings in Fig. 4, which shows the distance to $\boldsymbol{\theta}_0$ along a stochastic gradient trajectory, either minimizing $L_N$ (left) or $L_{\boldsymbol{\theta}_0}$ (center, right). Full experimental and theoretical details are available in Section A, together with additional results on both the hypercube and the top-$\kappa$ polytope.

**Results.** The results highlight three important aspects for effective learning. First, persistent and data-based initializations for the MCMC chains are critical (see Section B.3 for a detailed discussion), vastly outperforming random restarts, which introduce systematic bias in the gradient estimation (Fig. 4, center). Second, a larger dataset size $N$ provides a better approximation of the population loss, leading to a more accurate parameter recovery (Fig. 4, left), in line with Proposition 4. Finally (defining Hamming distance-based neighborhood systems $(\mathcal{N}_{r_s})_{s=1}^S$ by $\boldsymbol{y}' \in \mathcal{N}_{r_s}(\boldsymbol{y}) \Leftrightarrow d_H(\boldsymbol{y}, \boldsymbol{y}') = r_s$), using a mixture of proposals with Algorithm 2 (e.g., with $r_s \in \{1,2,3,6\}$) enables more effective exploration, improving convergence compared to a single proposal type (Fig. 4, right).

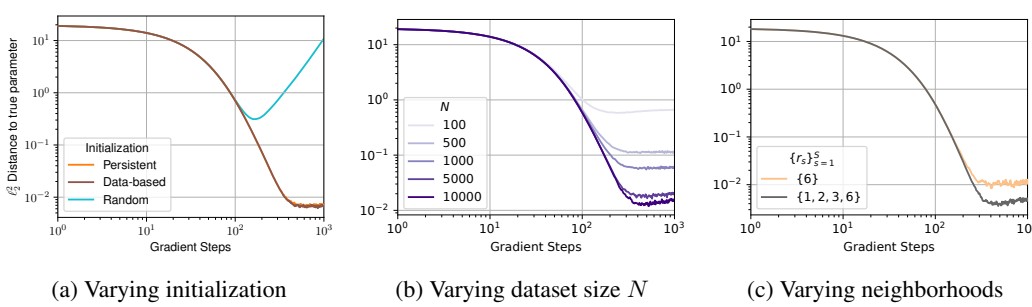

    (a) Varying initialization      (b) Varying dataset size $N$      (c) Varying neighborhoods

Figure 4: Squared distance $||\hat{\boldsymbol{\theta}}_n - \boldsymbol{\theta}_0||_2^2$ to the true parameter over optimization steps.

## 6 CONCLUSION

In this paper, we introduced a principled framework for integrating NP-hard combinatorial optimization layers into neural networks without relying on exact solvers. Our approach adapts neighborhood systems from the metaheuristics community, to design structure-aware proposal distributions for combinatorial MCMC. This leads to significant training speed-ups, enabling to tackle larger problem instances, which is crucial in operations research, where scaling up leads to substantial value creation by reducing marginal costs. In future work, we plan to extend our framework to large neighborhood search algorithms, which are heuristics that leverage neighborhood-wise exact optimization oracles.

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

## NOTATION

| Notation | Description |
|---|---|
| $\langle \boldsymbol{\theta}, \boldsymbol{y} \rangle$ | Euclidean inner product between two vectors $\boldsymbol{\theta}, \boldsymbol{y} \in \mathbb{R}^d$. |
| $\mathrm{conv}(\mathcal{Y})$ | Convex hull of a set $\mathcal{Y}$. |
| $\mathrm{dom}(\Omega)$ | Domain of a function $\Omega : \mathbb{R}^d \to \mathbb{R} \cup \{\infty\}$, defined as $\{\boldsymbol{\mu} \in \mathbb{R}^d : \Omega(\boldsymbol{\mu}) < \infty\}$. |
| $\Omega^*$ | Fenchel conjugate of $\Omega$, defined as $\Omega^*(\boldsymbol{\theta}) := \sup_{\boldsymbol{\mu} \in \mathbb{R}^d} \langle \boldsymbol{\theta}, \boldsymbol{\mu} \rangle - \Omega(\boldsymbol{\mu})$. |
| $\nabla\Omega$ | Gradient of $\Omega$. |
| $\partial\Omega$ | Subgradient of $\Omega$. |
| $J_{\boldsymbol{x}} f(\boldsymbol{x}, \boldsymbol{y})$ | Jacobian of a function $f : \mathcal{X} \times \mathcal{Y} \to \mathbb{R}^d$ with $\mathcal{X} \subseteq \mathbb{R}^n$ at point $(\boldsymbol{x}, \boldsymbol{y})$ w.r.t. $\boldsymbol{x}$, viewed as a matrix $J_{\boldsymbol{x}} f(\boldsymbol{x}, \boldsymbol{y}) \in \mathbb{R}^{n \times d}$. |
| $\mathcal{U}(\mathcal{X})$ | Uniform distribution on a set $\mathcal{X}$. |
| $\mathcal{N}(\boldsymbol{x}, \Sigma)$ | Normal distribution with mean $\boldsymbol{x} \in \mathbb{R}^d$ and covariance $\Sigma \in \mathbb{R}^{d \times d}$. |

## A  EXPERIMENTS ON EMPIRICAL CONVERGENCE OF GRADIENTS AND PARAMETERS

In this section, we evaluate the proposed approach on two discrete output spaces: sets and $\kappa$-subsets. These output spaces are for instance useful for multilabel classification. We focus on these output spaces because the exact MAP and marginal inference oracles are available, allowing us to compare our gradient estimators to exact gradients. We set $\varphi \equiv 0$ in these experiments.

### A.1  POLYTOPES AND CORRESPONDING ORACLES

The vertex set of the first polytope is the set of binary vectors in $\mathbb{R}^d$, which we denote $\mathcal{Y}^d := \{0, 1\}^d$, and $\mathrm{conv}(\mathcal{Y}^d) = [0, 1]^d$ is the "hypercube". The vertex set of the second is the set of binary vectors with exactly $\kappa$ ones and $d - \kappa$ zeros (with $0 < \kappa < d$),

$$\mathcal{Y}_\kappa^d := \big\{ \boldsymbol{y} \in \{0, 1\}^d : \langle \boldsymbol{y}, \mathbf{1} \rangle = \kappa \big\},$$

and $\mathrm{conv}(\mathcal{Y}_\kappa^d)$ is referred to as "top-$\kappa$ polytope" or "hypersimplex". Although these polytopes would not provide relevant use cases of the proposed approach in practice, since exact marginal inference oracles are available (see below), they allow us to compare the Fenchel-Young loss value and gradient estimated by our algorithm to their true value.

**Marginal inference.** For the hypercube, we have:

$$\mathbb{E}_{\pi_{\boldsymbol{\theta}, t}}[Y_i] = \sum_{\boldsymbol{y} \in \mathcal{Y}^d} \frac{\exp\left(\langle \boldsymbol{\theta}, \boldsymbol{y} \rangle / t\right)}{\sum_{\boldsymbol{y}' \in \mathcal{Y}^d} \exp\left(\langle \boldsymbol{\theta}, \boldsymbol{y}' \rangle / t\right)} y_i = \sum_{\boldsymbol{y} \in \{0,1\}^d} \frac{\exp\left(\sum_{j=1}^d \theta_j y_j / t\right)}{\sum_{\boldsymbol{y}' \in \{0,1\}^d} \exp\left(\sum_{j=1}^d \theta_j y_j' / t\right)} y_i$$

$$= \sum_{y_i \in \{0,1\}} \sum_{\boldsymbol{y}_{-i} \in \{0,1\}^{d-1}} \frac{\exp\left(\theta_i y_i / t + \sum_{j \neq i} \theta_j y_j / t\right)}{\sum_{y_i' \in \{0,1\}} \sum_{\boldsymbol{y}_{-i}' \in \{0,1\}^{d-1}} \exp\left(\theta_i y_i' / t + \sum_{j \neq i} \theta_j y_j' / t\right)} y_i$$

$$= \sum_{y_i \in \{0,1\}} \frac{\exp\left(\theta_i y_i / t\right)}{\sum_{y_i' \in \{0,1\}} \exp\left(\theta_i y_i' / t\right)} y_i \sum_{\boldsymbol{y}_{-i} \in \{0,1\}^{d-1}} \frac{\exp\left(\sum_{j \neq i} \theta_j y_j / t\right)}{\sum_{\boldsymbol{y}_{-i}' \in \{0,1\}^{d-1}} \exp\left(\sum_{j \neq i} \theta_j y_j' / t\right)}$$

$$= \sum_{y_i \in \{0,1\}} \frac{\exp\left(\theta_i y_i / t\right)}{\sum_{y_i' \in \{0,1\}} \exp\left(\theta_i y_i' / t\right)} y_i$$

$$= \frac{\exp\left(\theta_i / t\right)}{1 + \exp\left(\theta_i / t\right)}$$

$$= \sigma\left(\frac{\theta_i}{t}\right),$$

which gives $\mathbb{E}_{\pi_{\boldsymbol{\theta},t}}[Y] = \sigma\left(\frac{\boldsymbol{\theta}}{t}\right)$, where the logistic sigmoid function $\sigma$ is applied component-wise. The cumulant function is also tractable, as we have

$$
\begin{aligned}
\log \sum_{\boldsymbol{y} \in \mathcal{Y}^d} \exp\left(\langle \boldsymbol{\theta}, \boldsymbol{y} \rangle / t\right) &= \log \sum_{\boldsymbol{y} \in \{0,1\}^d} \exp\left(\sum_{i=1}^d \theta_i y_i / t\right) \\
&= \log \sum_{\boldsymbol{y}_1=0}^1 \sum_{\boldsymbol{y}_2=0}^1 \cdots \sum_{\boldsymbol{y}_d=0}^1 \exp\left(\sum_{i=1}^d \theta_i y_i / t\right) \\
&= \log \prod_{i=1}^d \sum_{y_i=0}^1 \exp\left(\theta_i y_i / t\right) \\
&= \log \prod_{i=1}^d \left(\exp(0) + \exp\left(\theta_i / t\right)\right) \\
&= \log \prod_{i=1}^d \left(1 + \exp\left(\theta_i / t\right)\right) \\
&= \sum_{i=1}^d \log\left(1 + \exp\left(\theta_i / t\right)\right).
\end{aligned}
$$

Another way to derive this is via the Fenchel conjugate.

For the top-$\kappa$ polytope, such closed-form formulas do not exist for the cumulant and its gradient. However, we implement them with dynamic programming, by viewing the top-$\kappa$ MAP problem as a $0/1$-knapsack problem with constant item weights, and by changing the $(\max, +)$ semiring into a $(\mathrm{LSE}, +)$ semiring. This returns the cumulant function, and we leverage PyTorch's automatic differentiation framework to compute its gradient. This simple implementation allows us to compute true Fenchel-Young losses values and their gradients in $\mathcal{O}(d\kappa)$ time and space complexity.

**Sampling.** For the hypercube, sampling from the Gibbs distribution on $\mathcal{Y}^d$ has closed form. Indeed, the latter is fully factorized, and we can sample $\boldsymbol{y} \sim \pi_{\boldsymbol{\theta},t}$ by sampling independently each component with $y_i \sim \mathrm{Bern}\left(\sigma(\theta_i/t)\right)$. Sampling from $\pi_{\boldsymbol{\theta},t}$ is also possible on $\mathcal{Y}_\kappa^d$, by sampling coordinates iteratively using the dynamic programming table used to compute the cumulant function (see, e.g., Algorithm 2 in Ahmed et al. (2024) for a detailed explanation).

## A.2 Neighborhood Graphs

**Hypercube.** On $\mathcal{Y}^d$, we use a family of neighborhood systems $\mathcal{N}_{\leq}^r$ parameterized by a Hamming distance radius $r \in [d-1]$. The graph is defined by:

$$
\forall \boldsymbol{y}, \boldsymbol{y}' \in \mathcal{Y}^d : \ \boldsymbol{y}' \in \mathcal{N}_{\leq}^r(\boldsymbol{y}) \Leftrightarrow 1 \leq d_H\left(\boldsymbol{y}, \boldsymbol{y}'\right) \leq r.
$$

That is, two vertices are neighbors if their Hamming distance is at most $r$. This graph is regular, with degree $|\mathcal{N}_{\leq}^r(\boldsymbol{y})| = \sum_{i=1}^r \binom{d}{i}$. This graph is naturally connected, as any binary vector $\boldsymbol{y}'$ can be reached from any other binary vector $\boldsymbol{y}$ in $||\boldsymbol{y}' - \boldsymbol{y}||_1$ moves, by flipping each bit where $y_i' \neq y_i$, iteratively. Indeed, this trajectory consists in moves between vertices with Hamming distance equal to $1$, and are therefore along edges of the neighborhood graph, regardless of the value of $r$.

We also use a slight variation on this family of neighborhood systems, the graphs $\mathcal{N}_{=}^r$, defined by:

$$
\forall \boldsymbol{y}, \boldsymbol{y}' \in \mathcal{Y}^d : \ \boldsymbol{y}' \in \mathcal{N}_{=}^r(\boldsymbol{y}) \Leftrightarrow d_H\left(\boldsymbol{y}, \boldsymbol{y}'\right) = r.
$$

These graphs, on the contrary, are not always connected: indeed, if $r$ is even, they contain two connected components (binary vectors with an even sum, and binary vectors with an odd sum). We only use such graphs when experimenting with neighborhood mixtures (see Algorithm 2), by aggregating them into a connected graph.

**Top-$\kappa$ polytope.** On $\mathcal{Y}_\kappa^d$, we use a family of neighborhoods systems $\mathcal{N}^s$ parameterized by a number of "swaps" $s \in [\![1, \min(\kappa, d - \kappa)]\!]$. The graph is defined by

$$\forall \boldsymbol{y}, \boldsymbol{y}' \in \mathcal{Y}_\kappa^d : \ \boldsymbol{y}' \in \mathcal{N}^s(\boldsymbol{y}) \Leftrightarrow d_H(\boldsymbol{y}, \boldsymbol{y}') = 2s.$$

That is, two vertices are neighbors if one can be reached from the other by performing $s$ "swaps", each swap corresponding to flipping a 1 to a 0 and vice-versa. This ensures that the resulting vector is still in $\mathcal{Y}_\kappa^d$. All $s$ swaps must be performed on distinct components. The resulting graph is known as the *Generalized Johnson Graph* $J(d, \kappa, \kappa - s)$, or *Uniform Subset Graph* (Chen and Lih, 1987). It is a regular graph, with degree $|\mathcal{N}^s(\boldsymbol{y})| = \binom{\kappa}{s}\binom{d-\kappa}{s}$. It is proved to be connected in Jones (2005), except if $d = 2\kappa$ and $s = \kappa$ (in this case, it consists in $\frac{1}{2}\binom{d}{\kappa}$ disjoint edges).

When $s = 1$, the neighborhood graph is the Johnson Graph $J(d, \kappa)$, which coincides with the graph associated to the polytope $\text{conv}(\mathcal{Y}_\kappa^d) = \Delta_{d,\kappa}$ (Rispoli, 2008).

## A.3 Convergence to exact gradients

In this section, we conduct experiments on the convergence of the MCMC estimators to the exact corresponding expectation (that is, convergence of the stochastic gradient estimator to the true gradient). The estimators are defined as

$$\widehat{\boldsymbol{y}}_t(\boldsymbol{\theta}) = \mathbb{E}_{\pi_{\boldsymbol{\theta},t}}[Y] \approx \frac{1}{K - K_0} \sum_{k=K_0+1}^{K} \boldsymbol{y}^{(k)},$$

where $\boldsymbol{y}^{(k)}$ is the $k$-th iterate of Algorithm 1 with maximization direction $\boldsymbol{\theta}$, final temperature $t$, and $K_0$ is a number of burn-in (or warm-up) iterations. The obtained estimator is compared to the exact expectation by performing marginal inference as described in Section A.1 (with a closed-form formula in the case of $\mathcal{Y}^d$, and by dynamic programming in the case of $\mathcal{Y}_\kappa^d$).

**Setup.** For $T > K_0$, let $\tilde{\mathbb{E}}(\boldsymbol{\theta}, T) \coloneqq \frac{1}{T-K_0} \sum_{k=K_0+1}^{T} \boldsymbol{y}^{(k)}$ be the stochastic estimate of the expectation at step $T$. We proceed by first randomly generating $\boldsymbol{\Theta} \in \mathbb{R}^{M \times d}$, with $M$ being the number of instances, by sampling $\boldsymbol{\Theta}_{i,j} \sim \mathcal{N}(0, 1)$ independently. Then, we evaluate the impact of the following hyperparameters on the estimation of $\mathbb{E}_{\pi_{\boldsymbol{\Theta}_i,t}}[Y]$, for $i \in [M]$:

1. $K_0$, the number of burn-in iterations,

2. $t$, the temperature parameter,

3. $C$, the number of parallel Markov chains.

**Metric.** The metric used is the squared Euclidean distance to the exact expectation, averaged on the $M$ instances

$$\frac{1}{M} \sum_{i=1}^{M} ||\mathbb{E}_{\pi_{\boldsymbol{\Theta}_i,t}}[Y] - \tilde{\mathbb{E}}(\boldsymbol{\Theta}_i, T)||_2^2,$$

which we measure for $T \in [\![K_0 + 1, K]\!]$.

**Polytopes.** For the hypercube $\mathcal{Y}^d$ and its neighborhood system $\mathcal{N}_\leq^r$, we use $d = 10$ and $r = 1$, which gives $|\mathcal{Y}^d| = 2^{10}$ and $|\mathcal{N}_\leq^r(\boldsymbol{y})| = 10$. For the top-$\kappa$ polytope $\mathcal{Y}_\kappa^d$ and its neighborhood system $\mathcal{N}^s$, we use $d = 10$, $\kappa = 3$ and $s = 1$, which gives $|\mathcal{Y}_\kappa^d| = 120$ and $|\mathcal{N}^s(\boldsymbol{y})| = 30$. We also use a larger scale for both polytopes in order to highlight the varying impact of the temperature $t$ depending on the combinatorial size of the problem, in the second experiment. For the large scale, we use $d = 1000$ and $r = 10$ for the hypercube, which give $|\mathcal{Y}^d| = 2^{1000} \approx 10^{301}$ and $|\mathcal{N}_\leq^r(\boldsymbol{y})| \approx 2.7 \times 10^{23}$, and we use $d = 1000$, $\kappa = 50$ and $s = 10$ for the top-$\kappa$ polytope, which give $|\mathcal{Y}_\kappa^d| \approx 9.5 \times 10^{84}$ and $|\mathcal{N}^s(\boldsymbol{y})| \approx 1.6 \times 10^{33}$.

**Hyperparameters.** For each experiment, we use $K = 3000$. We average over $M = 1000$ problem instances for statistical significance. We use $K_0 = 0$, except for the first experiment, where it varies. We use a final temperature $t = 1$, except for the second experiment, where it varies. We use an initial temperature $t_0 = t = 1$ (leading to a constant temperature schedule), except for the first experiment, where it depends on $K_0$. We use only one Markov chain and thus have $C = 1$, except for the third experiment, where it varies.

**(1) Impact of burn-in.** First, we evaluate the impact of $K_0$, the number of burn-in iterations. We use a truncated geometric cooling schedule $t_k = \max(\gamma^k \cdot t_0, t)$ with $\gamma = 0.995$. The initial temperature $t_0$ is set to $1/(\gamma^{K_0})$, so that $\forall k \geq K_0 + 1$, $t_k = t = 1$. The results are gathered in Fig. 5.

**(2) Impact of temperature.** We then evaluate the impact of the final temperature $t$ on the difficulty of the estimation task (different temperatures lead to different target expectations). The results for the small scale are gathered in Fig. 6, and the results for the large scale are gathered in Fig. 7.

**(3) Impact of the number of parallel Markov chains.** Finally, we evaluate the impact of the number of parallel Markov chains $C$ on the estimation. The results are gathered in Fig. 8.

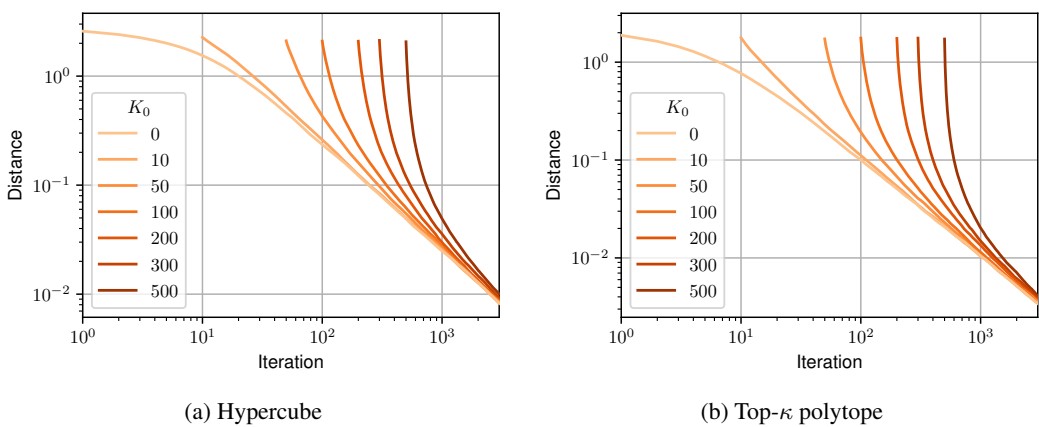

(a) Hypercube          (b) Top-$\kappa$ polytope

Figure 5: Convergence to exact expectation on the hypercube and the top-$\kappa$ polytope, for varying number of burn-in iterations $K_0$. We conclude that burn-in is not beneficial to the estimation, and taking $K_0 = 0$ is the best option.

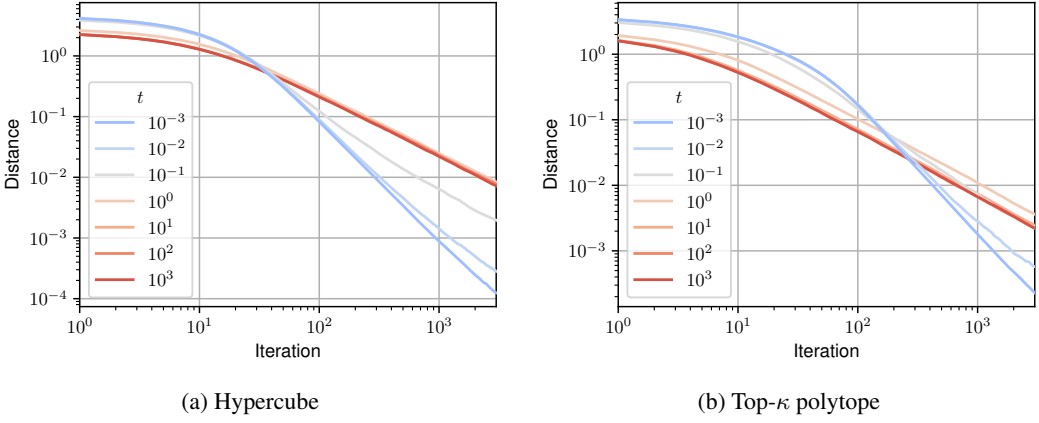

(a) Hypercube          (b) Top-$\kappa$ polytope

Figure 6: Convergence to exact expectation on the hypercube and the top-$\kappa$ polytope, for varying final temperature $t$ (small scale experiment). We conclude that lower temperatures facilitate the estimation.

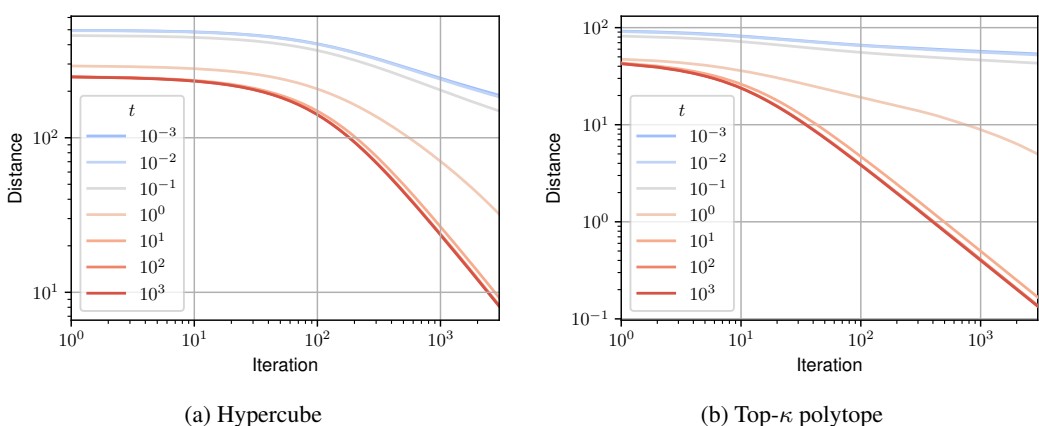

(a) Hypercube

(b) Top-$\kappa$ polytope

Figure 7: Convergence to exact expectation on the hypercube and the top-$\kappa$ polytope, for varying final temperature $t$ (large scale experiment). Contrary to the small scale case, larger temperatures are beneficial to the estimation when the solution set is combinatorially large.

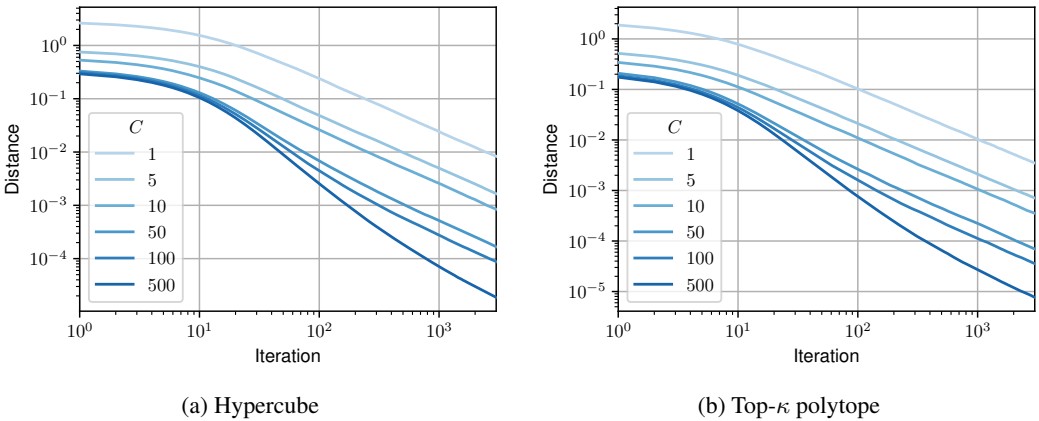

(a) Hypercube

(b) Top-$\kappa$ polytope

Figure 8: Convergence to exact expectation on the hypercube and the top-$\kappa$ polytope, for varying number of parallel Markov chains $C$. Running 10 times more chains in parallel provides roughly the same benefit as extending each chain by 10 times more iterations, highlighting the advantage of massively parallelized estimation.

In this section, we conduct experiments in the unconditional setting described in Section 4.4. As a reminder, the empirical $L_N$ and population $L_{\boldsymbol{\theta}_0}$ Fenchel-Young losses are given by:

$$
\begin{aligned}
L_N(\boldsymbol{\theta}; \boldsymbol{y}_1, \ldots, \boldsymbol{y}_N) &:= \frac{1}{N} \sum_{i=1}^{N} \ell_t(\boldsymbol{\theta}; \boldsymbol{y}_i) \\
&= A_t(\boldsymbol{\theta}) + \frac{1}{N} \sum_{i=1}^{N} \Omega_t(\boldsymbol{y}_i) - \langle \boldsymbol{\theta}, \bar{Y}_N \rangle \\
&= \ell_t(\boldsymbol{\theta}; \bar{Y}_N) + C_1(Y),
\end{aligned}
\tag{10}
$$

and

$$
\begin{aligned}
L_{\boldsymbol{\theta}_0}(\boldsymbol{\theta}) &:= \mathbb{E}_{(\boldsymbol{y}_i)_{i=1}^{N} \sim (\pi_{\boldsymbol{\theta}_0, t})^{\otimes N}} [L_N(\boldsymbol{\theta}; \boldsymbol{y}_1, \ldots, \boldsymbol{y}_N)] \\
&= A_t(\boldsymbol{\theta}) + \mathbb{E}_{\pi_{\boldsymbol{\theta}_0, t}} [\Omega_t(Y)] - \langle \boldsymbol{\theta}, \widehat{\boldsymbol{y}}_t(\boldsymbol{\theta}_0) \rangle \\
&= \ell_t(\boldsymbol{\theta}; \widehat{\boldsymbol{y}}_t(\boldsymbol{\theta}_0)) + C_2(\boldsymbol{\theta}_0),
\end{aligned}
\tag{11}
$$

where the constants $C_1(Y) = \frac{1}{N} \sum_{i=1}^{N} \Omega_t(\boldsymbol{y}_i) - \Omega_t(\bar{Y}_N)$ and $C_2(\boldsymbol{\theta}_0) = \mathbb{E}_{\pi_{\boldsymbol{\theta}_0, t}} [\Omega_t(Y)] - \Omega_t(\widehat{\boldsymbol{y}}_t(\boldsymbol{\theta}_0))$ do not depend on $\boldsymbol{\theta}$. As Jensen gaps, they are non-negative by convexity of $\Omega_t$.

**2D visualization.** As an introductory example, we display stochastic gradient trajectories in Fig. 9. The parameter $\boldsymbol{\theta} \in \mathbb{R}^d$ is updated following Eq. (7) to minimize the population loss $L_{\boldsymbol{\theta}_0}$ defined in Eq. (11), with $\boldsymbol{\theta}_0 = (1/2, 1/2)$. The polytope used is the 2-dimensional hypercube $\mathcal{Y}^2$, with neighborhood graph $\mathcal{N}_1$ (neighbors are adjacent vertices of the square). We present trajectories obtained using MCMC-sampled gradients, comparing results from both 1 and 100 Markov chain iterations with Algorithm 1. For comparison, we include trajectories obtained using Monte Carlo-sampled (i.e., unbiased) gradients, using 1 and 100 samples.

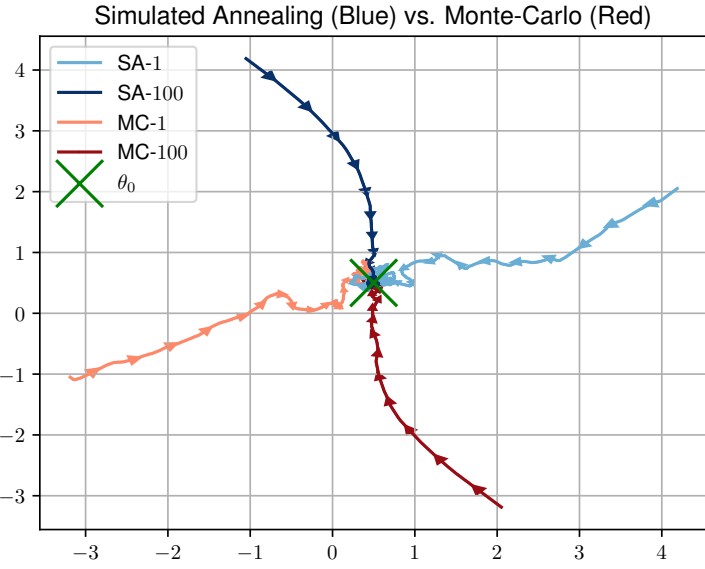

Figure 9: Comparison of stochastic gradient trajectories for a SA / M-H oracle on $\mathcal{Y}^2$ and unbiased stochastic gradients obtained via Monte Carlo sampling. Increasing the number of Markov chain iterations yields smoother trajectories, similar to the effect of using more Monte Carlo samples in the case of perturbation-based methods (Berthet et al., 2020).

**General setup.** We proceed by first randomly generating true parameters $\boldsymbol{\Theta}_0 \in \mathbb{R}^{M \times d}$, with $M$ being a number of problem instances we average on (in order to reduce noise in our observations), by sampling $\boldsymbol{\Theta}_{i,j} \sim \mathcal{N}(0, 1)$ independently. The goal is to learn each parameter vector $(\boldsymbol{\Theta}_0)_i \in \mathbb{R}^d$, $i \in [M]$, as $M$ independent problems. The model is randomly initialized at $\hat{\boldsymbol{\Theta}}_0$, and updated with Adam (Kingma and Ba, 2017) to minimize the loss. In order to better separate noise due to the optimization process and noise due to the sampling process, we use the population loss $L_{(\boldsymbol{\Theta}_0)_i}$ for general experiments, and use the empirical loss $L_N$ only when focusing on the impact of the dataset size $N$. In this case, we create a dataset $Y \in \mathbb{R}^{M \times N \times d}$, with $N$ being the number of samples, by sampling independently $Y_{i,j} \sim \pi_{(\boldsymbol{\Theta}_0)_i}$, $\forall i \in [M]$, $\forall j \in [N]$.

We study the impact of the following hyperparameters on learning:

1. $K$, the number of Markov chain iterations,

2. $C$, the number of parallel Markov chains,

3. the initialization method used for the chains (either random, persistent, or data-based),

4. $N$, the number of samples in the dataset.

**Metrics.** The first metric used is the objective function actually minimized, i.e., the population loss, averaged on the $M$ instances:

$$\frac{1}{M} \sum_{i=1}^{M} L_{(\boldsymbol{\Theta}_0)_i}((\hat{\boldsymbol{\Theta}}_n)_i),$$

where $(\hat{\boldsymbol{\Theta}}_n)_i$ is the $n$-th iterate of the optimization process for the problem instance $i \in [M]$. We measure this loss for $n \in [n_{\max}]$, with $n_{\max}$ the total number of gradient iterations. For the fourth experiment, where we evaluate the impact of the number of samples $N$, we measure instead the empirical Fenchel-Young loss:

$$\frac{1}{M} \sum_{i=1}^{M} L_N((\hat{\boldsymbol{\Theta}}_n)_i \, ; \, Y_{i,1}, \dots Y_{i,N})$$

In both cases, the best loss value that can be reached is positive but cannot be computed: it corresponds to the constants $C_1$ and $C_2$ in Eq. (10) and Eq. (11). Thus, we also provide "stretched" figures, where we plot the loss minus the best loss found during the optimization process.

The second metric used is the squared euclidean distance of the estimate to the true parameter, also averaged on the $M$ instances:

$$\frac{1}{M} \sum_{i=1}^{M} ||(\boldsymbol{\Theta}_0)_i - (\hat{\boldsymbol{\Theta}}_n)_i||_2^2.$$

As the top-$\kappa$ polytope is of dimension $d - 1$, the model is only specified up to vectors orthogonal to the direction of the smallest affine subspace it spans. Thus, in this case, we measure instead:

$$\frac{1}{M} \sum_{i=1}^{M} || \, \mathsf{P}^{\perp}_D \left((\boldsymbol{\Theta}_0)_i\right) - \mathsf{P}^{\perp}_D \left((\hat{\boldsymbol{\Theta}}_n)_i\right) ||_2^2,$$

where $\mathsf{P}^{\perp}_D$ is the orthogonal projector on the hyperplane $D = \{\boldsymbol{x} \in \mathbb{R}^d : \langle \mathbf{1}, \boldsymbol{x} \rangle = 0\}$, which is the corresponding direction.

**Polytopes.** For the hypercube $\mathcal{Y}^d$ and its neighborhood system $\mathcal{N}_{\leq}^r$, we use $d = 10$ and $r = 1$, except in the fifth experiment, where we use a mixture of $\mathcal{N}_{=}^r$ neighborhoods (detailed below). For the top-$\kappa$ polytope $\mathcal{Y}_\kappa^d$ and its neighborhood system $\mathcal{N}^s$, we use $d = 10$, $\kappa = 3$ and $s = 1$.

**Hyperparameters.** For each experiment, we perform 1000 gradient steps. We use $K_0 = 0$, final temperature $t = 1$ and initial temperature $t_0 = t = 1$ (leading to a constant temperature schedule). We use $K = 1000$ Markov chain iterations, except in the first experiment, where it varies. We use only one Markov chain and thus have $C = 1$, except for the second experiment, where it varies. We use a persistent initialization method for the Markov chains, except in the third experiment, where we compare the three different methods. For statistical significance, we average over $M = 100$ problem instances for each experiment, except in the third experiment, where we use $M = 1000$. We work in the limit case $N \to \infty$, except in the fourth experiment, where $N$ varies.

**(1) Impact of the length of Markov chains.** First, we evaluate the impact of $K$, the number of inner iterations, i.e., the length of each Markov chain. The results are gathered in Fig. 10.

**(2) Impact of the number of parallel Markov chains.** We now evaluate the impact of the number of Markov chains $C$ run in parallel to perform each gradient estimation on the learning process. The results are gathered in Fig. 11.

**(3) Impact of the initialization method.** Then, we evaluate the impact of the method to initialize each Markov chain used for gradient estimation. The persistent method consists in setting $\boldsymbol{y}^{(n+1,0)} = \boldsymbol{y}^{(n,K)}$, the data-based method consists in setting $\boldsymbol{y}^{(n+1,0)} = \boldsymbol{y}_i$ with $i \sim \mathcal{U}([N])$, and the random method consists in setting $\boldsymbol{y}^{(n+1,0)} \sim \mathcal{U}(\mathcal{Y})$ (see Section B.3 and Table 4 for a detailed explanation). The results are gathered in Fig. 12.

**(4) Impact of the dataset size.** We now evaluate the impact of the number of samples $N$ from $\pi_{\boldsymbol{\theta}_0}$ (i.e., the size of the dataset $(\boldsymbol{y}_i)_{i=1}^N$) on the estimation of the true parameter $\boldsymbol{\theta}_0$. The results are gathered in Fig. 13.

**(5) Impact of neigborhood mixtures.** Finally, we evaluate the impact of the use of neighborhood mixtures. To do so, we use mixtures $\{\mathcal{N}_{=}^{r_s}\}_{s=1}^S$, once with $\{r_s\}_{s=1}^S = \{5\}$ opposed to $\{r_s\}_{s=1}^S = \{1, 5\}$, and once with $\{r_s\}_{s=1}^S = \{6\}$ (which gives a reducible Markov chain as 6 is even, so that the individual neighborhood graph $\mathcal{N}_{=}^6$ is not connected, and has to connected components) opposed to $\{r_s\}_{s=1}^S = \{1, 2, 3, 6\}$. The results are gathered in Fig. 14.

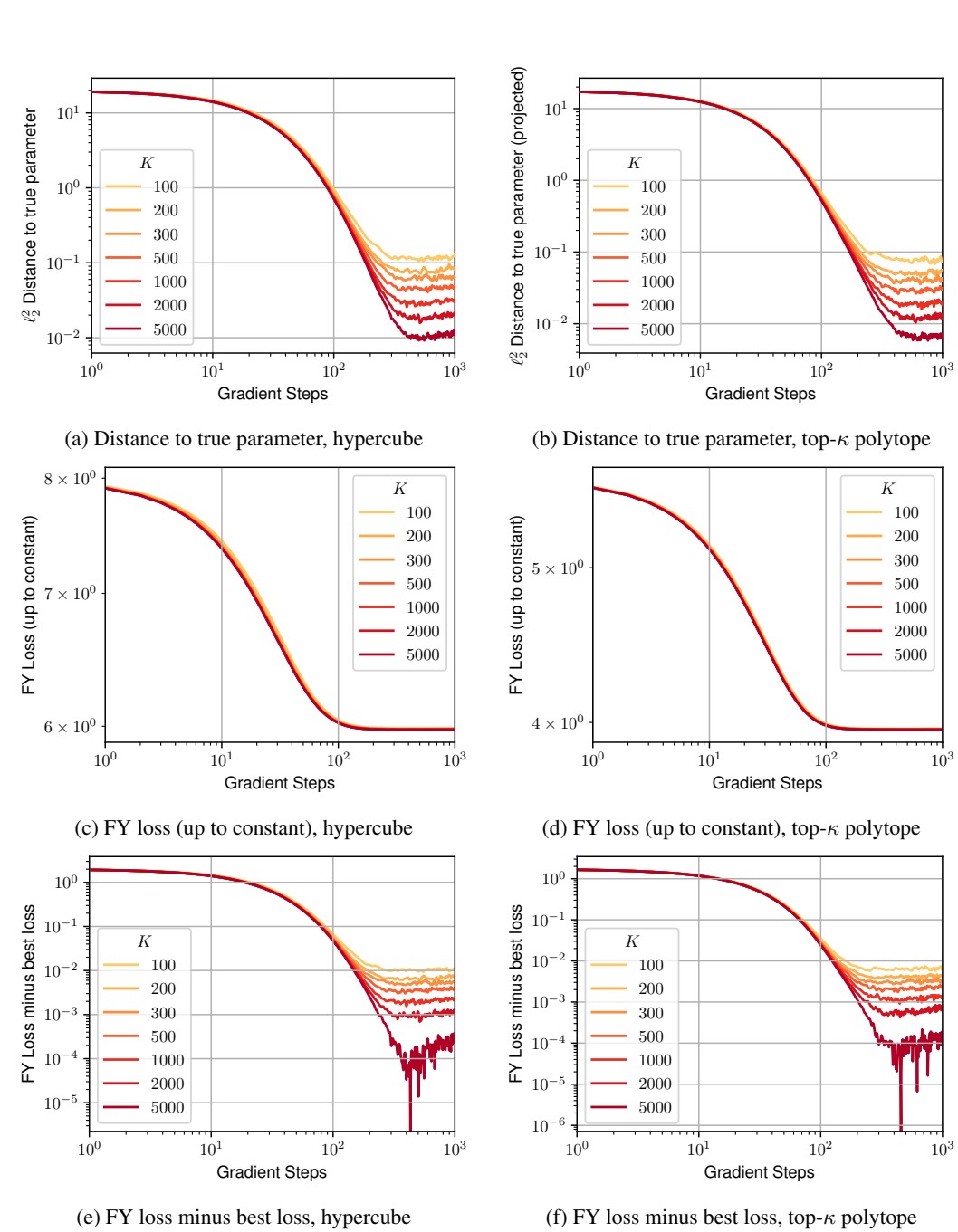

(a) Distance to true parameter, hypercube

(b) Distance to true parameter, top-$\kappa$ polytope

(c) FY loss (up to constant), hypercube

(d) FY loss (up to constant), top-$\kappa$ polytope

(e) FY loss minus best loss, hypercube

(f) FY loss minus best loss, top-$\kappa$ polytope

Figure 10: Convergence to the true parameter on the hypercube (left) and the top-$\kappa$ polytope (right), for varying number of Markov chain iterations $K$. Longer chains improve learning.

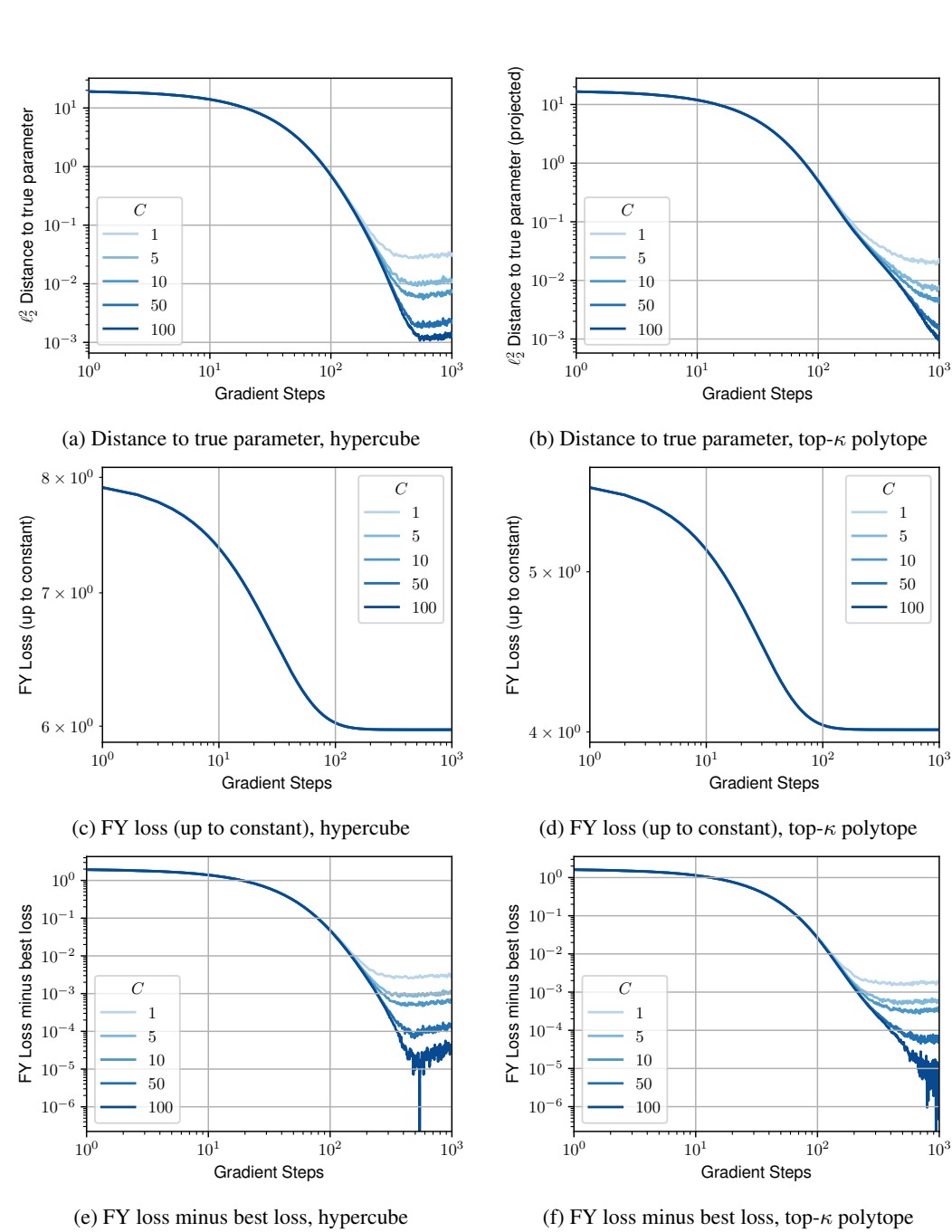

Figure 11: Convergence to the true parameter on the hypercube (left) and the top-$\kappa$ polytope (right), for varying number of parallel Markov chains $C$. Adding Markov chains improves estimation.

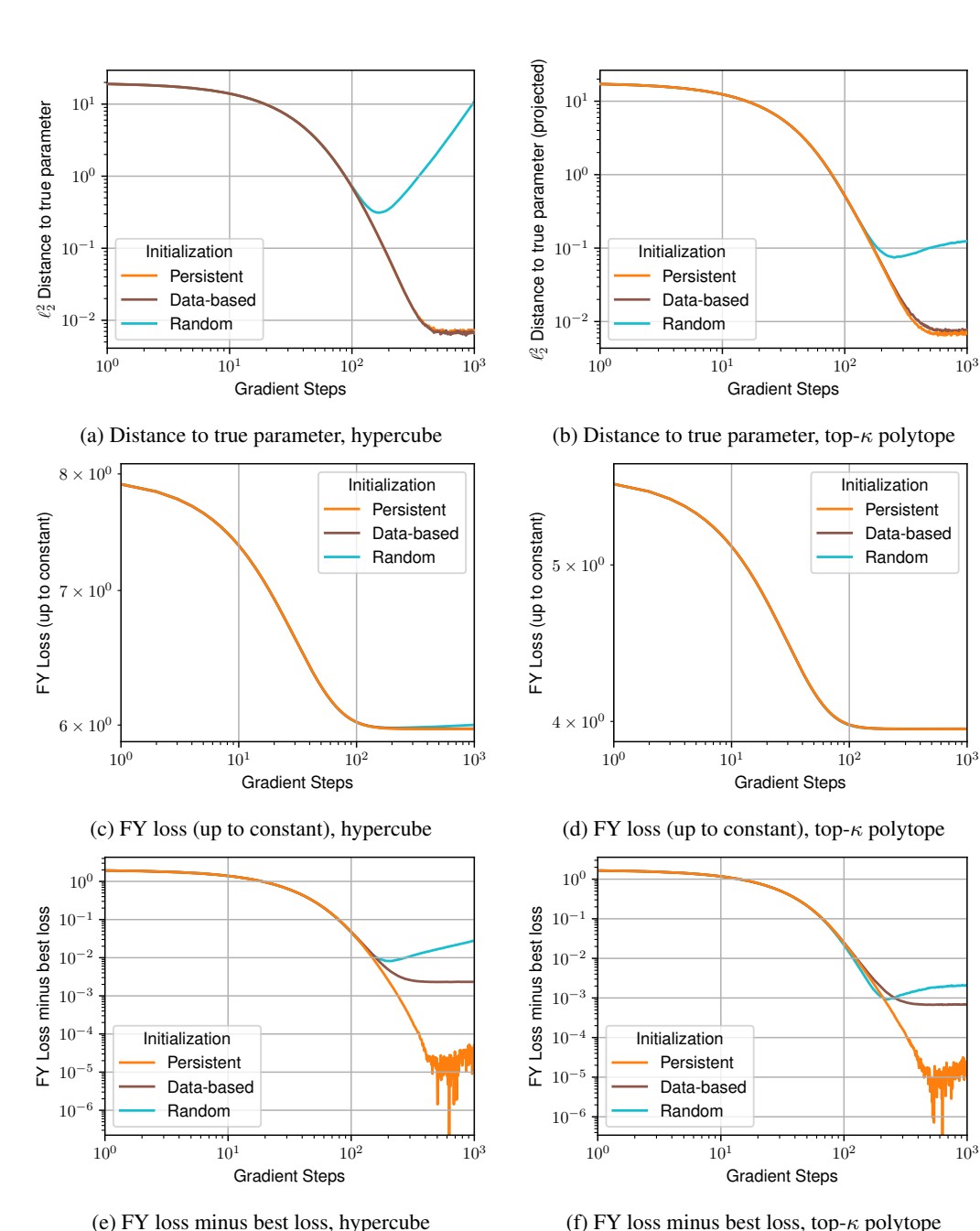

(a) Distance to true parameter, hypercube

(b) Distance to true parameter, top-$\kappa$ polytope

(c) FY loss (up to constant), hypercube

(d) FY loss (up to constant), top-$\kappa$ polytope

(e) FY loss minus best loss, hypercube

(f) FY loss minus best loss, top-$\kappa$ polytope

Figure 12: Convergence to the true parameter on the hypercube (left) and the top-$\kappa$ polytope (right), for varying Markov chain initialization method. The persistent and data-based initialization methods significantly outperform the random initialization method.

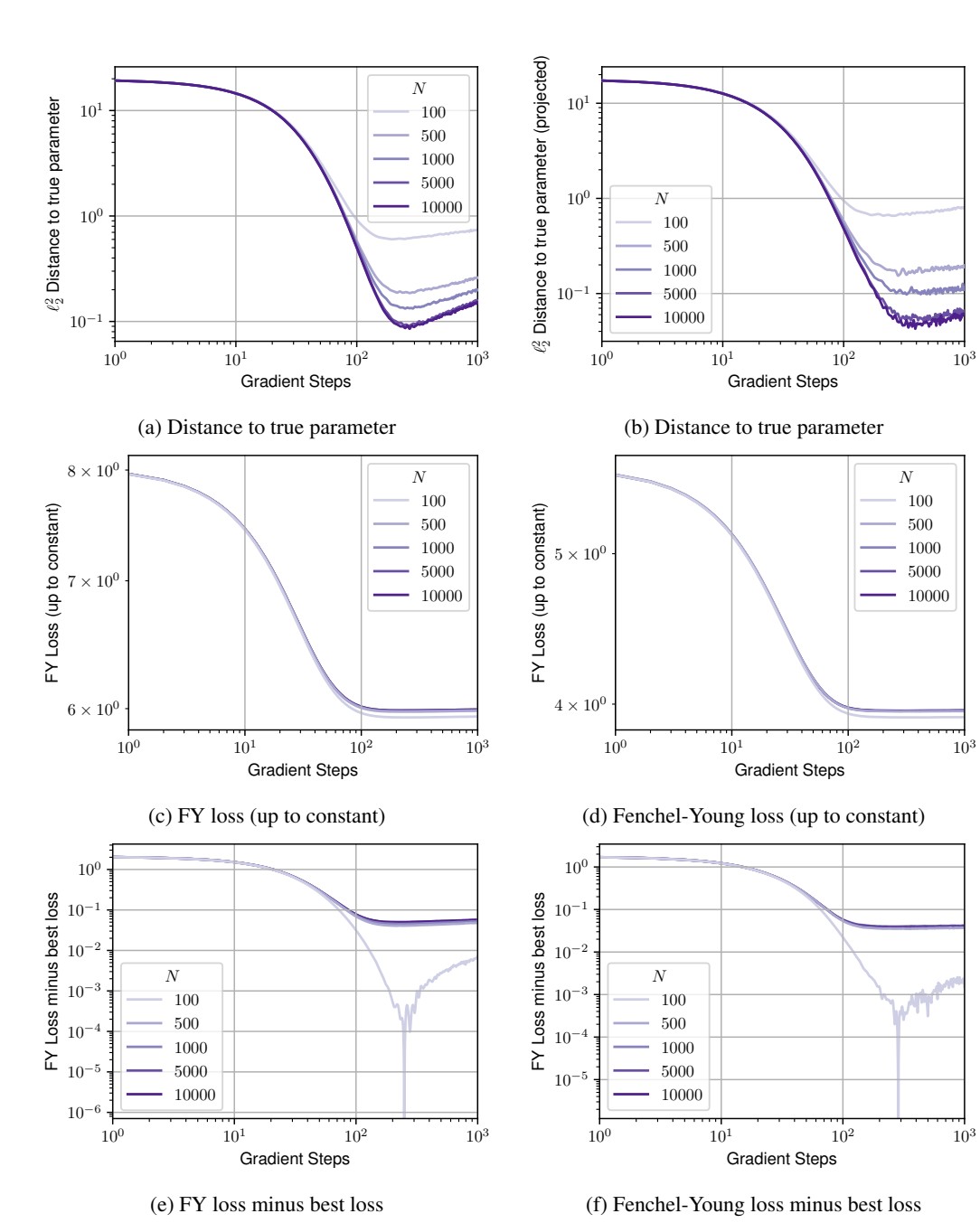

(a) Distance to true parameter

(b) Distance to true parameter

(c) FY loss (up to constant)

(d) Fenchel-Young loss (up to constant)

(e) FY loss minus best loss

(f) Fenchel-Young loss minus best loss

Figure 13: Convergence to the true parameter on the hypercube (left) and the top-$\kappa$ polytope (right), for varying number of samples $N$ in the dataset. As the dataset is different for each task, the empirical Fenchel-Young loss $L_N$, which is the minimized objective function (contrary to other experiments, where we minimize $L_{\boldsymbol{\theta}_0}$), also varies. Although empirical Fenchel-Young losses associated to smaller datasets appear easier to minimize, increasing the dataset size reduces the bias and thus the distance to $\boldsymbol{\theta}_0$, as expected.

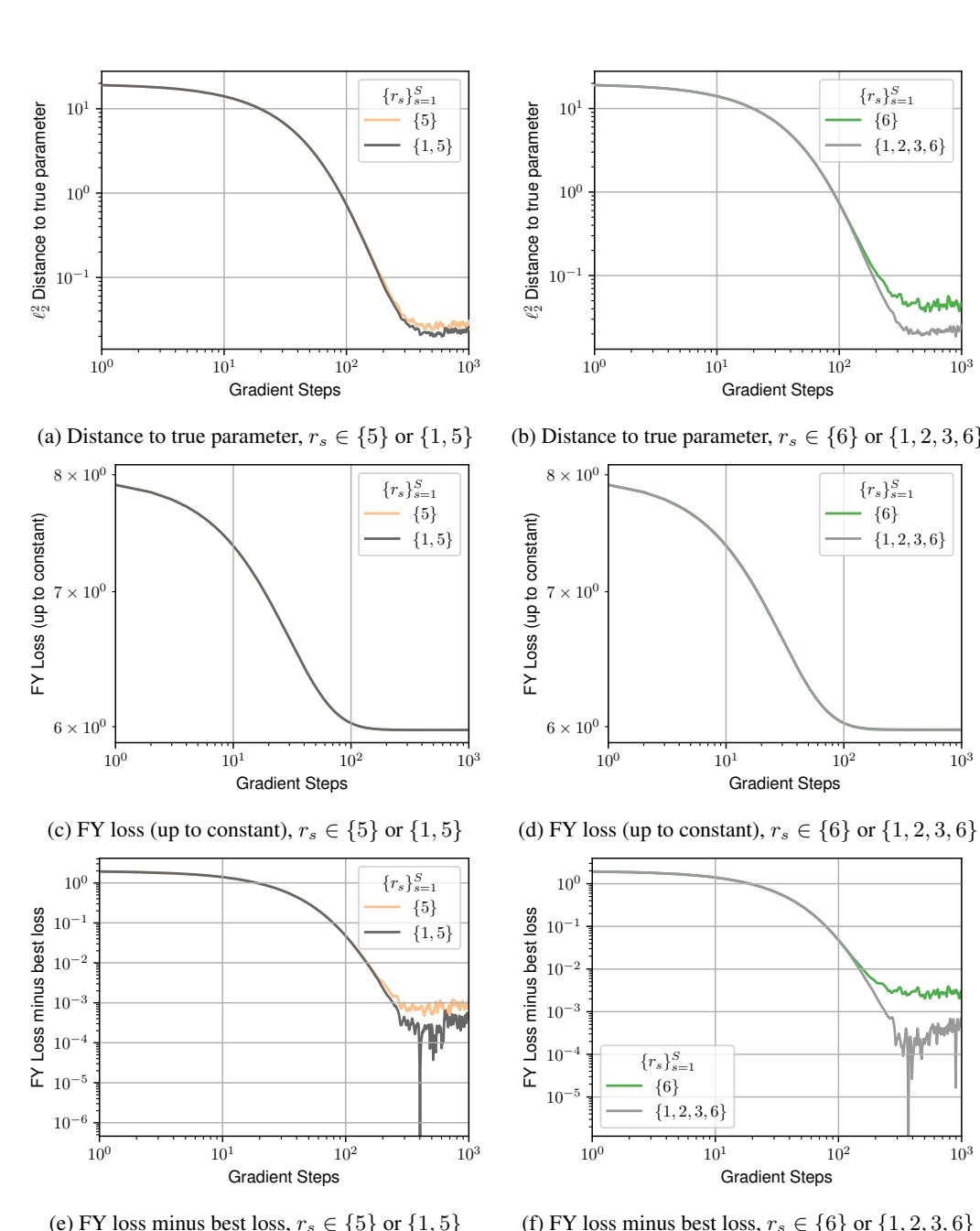

(a) Distance to true parameter, $r_s \in \{5\}$ or $\{1,5\}$

(b) Distance to true parameter, $r_s \in \{6\}$ or $\{1,2,3,6\}$

(c) FY loss (up to constant), $r_s \in \{5\}$ or $\{1,5\}$

(d) FY loss (up to constant), $r_s \in \{6\}$ or $\{1,2,3,6\}$

(e) FY loss minus best loss, $r_s \in \{5\}$ or $\{1,5\}$

(f) FY loss minus best loss, $r_s \in \{6\}$ or $\{1,2,3,6\}$

Figure 14: Convergence to the true parameter on the hypercube, with different mixtures of neighborhood systems $\{\mathcal{N}_=^{r_s}\}_{s=1}^S$: comparing $r_s \in \{5\}$ to $r_s \in \{1,5\}$ (left), and comparing $r_s \in \{6\}$ to $r_s \in \{1,2,3,6\}$ (right). Using more neighborhoods in the mixture improves learning.

# B ADDITIONAL MATERIAL

## B.1 FENCHEL-YOUNG LOSS FOR $K = 1$ IN THE UNCONDITIONAL SETTING

This proposition is analogous to Proposition 3, but in the unconditional setting, when using a data-based initialization method – i.e., the original CD initialization scheme, without persistent Markov chains. See Section B.3 for a detailed discussion about this.

**Proposition 6.** *Let $\boldsymbol{p}_{\boldsymbol{\theta}, \bar{Y}_N}^{(1)}$ denote the distribution of the first iterate of the Markov chain defined by the Markov transition kernel given in Eq. (3), with proposal distribution q and initialized by $\boldsymbol{y}^{(0)} = \boldsymbol{y}_i$, with $i \sim \mathcal{U}(\llbracket 1, N \rrbracket)$. There exists a dataset-dependent regularization $\Omega_{\bar{Y}_N}$ with the following properties: $\Omega_{\bar{Y}_N}$ is $tN / \sum_{i=1}^{N} \mathbb{E}_{q(\boldsymbol{y}_i, \cdot)} ||Y - \boldsymbol{y}_i||_2^2$-strongly convex; it is such that:*

$$\mathbb{E}_{\boldsymbol{p}_{\boldsymbol{\theta}, \bar{Y}_N}^{(1)}} [Y] = \underset{\boldsymbol{\mu} \in \text{conv}\left(\bigcup_{i=1}^{N} \{\mathcal{N}(\boldsymbol{y}_i) \cup \{\boldsymbol{y}_i\}\}\right)}{\text{argmax}} \left\{ \langle \boldsymbol{\theta}, \boldsymbol{\mu} \rangle - \Omega_{\bar{Y}_N}(\boldsymbol{\mu}) \right\};$$

*and the Fenchel-Young loss $L_{\Omega_{\bar{Y}_N}}$ generated by $\Omega_{\bar{Y}_N}$ is $\frac{1}{N} \sum_{i=1}^{N} \mathbb{E}_{q(\boldsymbol{y}_i, \cdot)} ||Y - \boldsymbol{y}_i||_2^2 / t$-smooth in its first argument, and such that $\nabla_{\boldsymbol{\theta}} L_{\Omega_{\bar{Y}_N}}(\boldsymbol{\theta}; \boldsymbol{y}) = \mathbb{E}_{\boldsymbol{p}_{\boldsymbol{\theta}, \bar{Y}_N}^{(1)}} [Y] - \boldsymbol{y}.$*

The proof is given in Section E.7.

## B.2 PROPERTIES OF THE EXPECTED FIRST ITERATE

**Proposition 7.** *Let $\boldsymbol{\theta} \in \mathbb{R}^d$, $\boldsymbol{y} \in \mathcal{Y}$. Let*

$$\mathcal{N}_{better}(\boldsymbol{y}) := \{\boldsymbol{y}' \in \mathcal{N}(\boldsymbol{y}) \mid \langle \boldsymbol{\theta}, \boldsymbol{y}' \rangle + \varphi(\boldsymbol{y}') > \langle \boldsymbol{\theta}, \boldsymbol{y} \rangle + \varphi(\boldsymbol{y})\}$$

*denote the set of improving neighbors of $\boldsymbol{y}$ for the unregularized objective function. We have the following properties:*

$$\mathbb{E}_{\boldsymbol{p}_{\boldsymbol{\theta}, \boldsymbol{y}}^{(1)}} [Y] \in \text{conv}\left(\mathcal{N}(\boldsymbol{y}) \cup \{\boldsymbol{y}\}\right),$$

$$\mathbb{E}_{\boldsymbol{p}_{\boldsymbol{\theta}, \boldsymbol{y}}^{(1)}} [Y] \xrightarrow[t \to 0^+]{} \boldsymbol{y} + \sum_{\boldsymbol{y}' \in \mathcal{N}_{better}(\boldsymbol{y})} q(\boldsymbol{y}, \boldsymbol{y}') \cdot (\boldsymbol{y}' - \boldsymbol{y}),$$

$$and \quad \mathbb{E}_{\boldsymbol{p}_{\boldsymbol{\theta}, \boldsymbol{y}}^{(1)}} [Y] \xrightarrow[t \to \infty]{} \boldsymbol{y} + \sum_{\boldsymbol{y}' \in \mathcal{N}(\boldsymbol{y})} \min \left[q(\boldsymbol{y}, \boldsymbol{y}'), q(\boldsymbol{y}', \boldsymbol{y})\right] \cdot (\boldsymbol{y}' - \boldsymbol{y}).$$

The proof is given in Section E.8. Thus, as the set $\mathcal{N}_{\text{better}}$ is defined according the value of the original, unregularized objective function $\boldsymbol{y} \mapsto \langle \boldsymbol{\theta}, \boldsymbol{y} \rangle + \varphi(\boldsymbol{y})$, the low temperature behavior of the regularized maximizer $\mathbb{E}_{\boldsymbol{p}_{\boldsymbol{\theta}, \boldsymbol{y}}^{(1)}} [Y]$ effectively reflects the fact that the regularization function $\Omega_{\boldsymbol{y}}$ extends the influence of $\varphi$ from the vertices $\mathcal{N}(\boldsymbol{y}) \cup \{\boldsymbol{y}\}$ to their convex hull.

## B.3 MARKOV CHAIN INITIALIZATION

In contrastive divergence (CD) learning, the intractable expectation in the log-likelihood gradient is approximated by short-run MCMC, initialized at the data distribution (Hinton, 2000) (using a Gibbs sampler in the setting of Restricted Boltzmann Machines).

Here, we note, at the $n$-th iteration of gradient descent:

$$\nabla_W L_N(\hat{W}_n) \approx \frac{1}{|B_n|} \sum_{i \in B_n} J_W g_{\hat{W}_n}(\boldsymbol{x}_i) \left( \frac{1}{K} \sum_{k=1}^{K} \boldsymbol{y}_i^{(n+1, k)} - \boldsymbol{y}_i \right),$$

for theconditional setting, with $B_n$ being the mini-batch (or full batch) used at iteration $n$, $\boldsymbol{y}_i$ the ground-truth structure associated to $\boldsymbol{x}_i$ in the dataset, and $\boldsymbol{y}_i^{(n+1, k)}$ the $k$-th iterate of Algorithm 1, with maximization direction $g_{\hat{W}_n}(\boldsymbol{x}_i)$, and initialization point $\boldsymbol{y}_i^{(n+1, 0)}$. We also note:

$$\nabla_{\boldsymbol{\theta}} L_N(\hat{\boldsymbol{\theta}}_n) \approx \frac{1}{K} \sum_{k=1}^{K} \boldsymbol{y}^{(n+1,\,k)} - \bar{Y}_N$$

for the unconditional setting, with $\boldsymbol{y}^{(n+1,\,k)}$ being the $k$-th iterate of Algorithm 1, with maximization direction $\hat{\boldsymbol{\theta}}_n$, and initialization point $\boldsymbol{y}^{(n+1,\,0)}$.

In CD learning of unconditional EBMs (i.e., in our unconditional setting), the Markov Chain is initialized at the empirical data distribution (Hinton, 2000; Carreira-Perpiñán and Hinton, 2005), as explained earlier. Persistent Contrastive Divergence (PCD) learning (Tieleman, 2008) modifies CD by maintaining a persistent Markov chain. Thus, instead of initializing the chain from the data distribution in each iteration, the chain continues from its last state in the previous iteration, by setting $\boldsymbol{y}^{(n+1,\,0)} = \boldsymbol{y}^{(n,\,K)}$. This approach aims to provide a better approximation of the model distribution and to reduce the bias introduced by the initialization of the Markov chain in CD. These are two types of informative initialization methods, which aim at reducing the mixing times of the Markov Chains.

However, neither of these can be applied to the conditional setting, as observed in (Mnih et al., 2012) in the context of conditional Restricted Boltzmann Machines (which are a type of EBMs). Indeed, on the one hand, PCD takes advantage of the fact that the parameter $\hat{\boldsymbol{\theta}}$ does not change too much from one iteration to the next, so that a Markov Chain that has reached equilibrium on $\hat{\boldsymbol{\theta}}_n$ is not far from equilibrium on $\hat{\boldsymbol{\theta}}_{n+1}$. This does not hold in the conditional setting, as each $\boldsymbol{x}_i$ leads to a different $\hat{\boldsymbol{\theta}}_i = g_{\hat{W}}(\boldsymbol{x}_i)$. On the other hand, the data-based initialization method in CD would amount to initialize the chains at the empirical marginal data distribution on $\mathcal{Y}$, and would be irrelevant in a conditional setting, since the distribution we want each Markov Chain to approximate is conditioned on the input $\boldsymbol{x}_i$.

An option is to use persistent chains if training for multiple epochs, and to initialize the Markov Chain associated to $(\boldsymbol{x}_i, \boldsymbol{y}_i)$ for epoch $j$ at the final state of the one associated to the same data point $(\boldsymbol{x}_i, \boldsymbol{y}_i)$ at epoch $j-1$. However, this method is relevant than PCD in the unconditional setting, as $\hat{w}$ changes a lot more in a full epoch than $\hat{\theta}$ in just one gradient step in the unconditional setting. It might be relevant, however, if each epoch consists in a single, full-batch gradient step. Nevertheless, it would require to store a significant number of states $\boldsymbol{y}_i^{(n,\,K)}$ (one for each point in the dataset). The solution we propose, for both full-batch and mini-batch settings, is to initialize the chains at the empirical data distribution conditioned on the input $\boldsymbol{x}_i$, which amounts to initialize them at the ground-truth $\boldsymbol{y}_i$.

This discussion is summed up in Table 4.

Table 4: Possible Markov Chain Initialization Methods under each Learning Setting

| Init. Method / Setting | Unconditional | Conditional, Batch | Conditional, Mini-Batch |
|---|---|---|---|
| **Persistent** | $\boldsymbol{y}^{(n+1,0)} = \boldsymbol{y}^{(n,K)}$ | $\boldsymbol{y}_i^{(n+1,\,0)} = \boldsymbol{y}_i^{(n,\,K)}$ | / |
| **Data-Based** | $\boldsymbol{y}^{(n+1,0)} = \boldsymbol{y}_j$, with $j \sim \mathcal{U}(\llbracket 1, N \rrbracket)$ | $\boldsymbol{y}_i^{(n+1,\,0)} = \boldsymbol{y}_i$ | $\boldsymbol{y}_i^{(n+1,\,0)} = \boldsymbol{y}_i$ |
| **Random** | $\boldsymbol{y}^{(n+1,0)} \sim \mathcal{U}(\mathcal{Y})$ | $\boldsymbol{y}_i^{(n+1,\,0)} \sim \mathcal{U}(\mathcal{Y})$ | $\boldsymbol{y}_i^{(n+1,\,0)} \sim \mathcal{U}(\mathcal{Y})$ |

**Remark 1.** *The use of uniform distributions on $\mathcal{Y}$ for the random initialization method can naturally be replaced by any other different prior distribution.*

## C  DETAILS ON THE DVRPTW

### C.1  OVERVIEW OF THE CHALLENGE.

We evaluate the proposed approach on a large-scale, ML-enriched combinatorial optimization problem: the *EURO Meets NeurIPS 2022 Vehicle Routing Competition* (Kool et al., 2023). In this dynamic vehicle routing problem with time windows (DVRPTW), requests arrive continuously throughout a planning horizon, which is partitioned into a series of delivery waves $\mathcal{W} = \{[\tau_0, \tau_1], [\tau_1, \tau_2], \ldots, [\tau_{|\mathcal{W}|-1}, \tau_{|\mathcal{W}|}]\}$.

At the start of each wave $\omega$, a dispatching and vehicle routing problem must be solved for the set of requests $\mathcal{R}^\omega$ specific to that wave (in which we include the depot $D$), encoded into the system state $\boldsymbol{x}^\omega$. We note $\mathcal{Y}(\boldsymbol{x}^\omega)$ the set of feasible decisions associated to state $\boldsymbol{x}^\omega$.

A feasible solution $\boldsymbol{y}^\omega \in \mathcal{Y}(\boldsymbol{x}^\omega)$ must contain all requests that must be dispatched before $\tau_\omega$ (the rest are postponable), allow each of its routes to visit the requests they dispatch within their respective time windows, and be such that the cumulative customer demand on each of its routes does not exceed a given vehicle capacity. It is encoded thanks to a vector $\left(y_{i,j}^\omega\right)_{i,j \in \mathcal{R}^\omega}$, where $y_{i,j}^\omega = 1$ if the solution contains the directed route segment from $i$ to $j$, and $y_{i,j}^\omega = 0$ otherwise. The set of requests $\mathcal{R}^{\omega+1}$ is obtained by removing all requests dispatched by the chosen solution $\boldsymbol{y}^\omega$ from $\mathcal{R}^\omega$ and adding all new requests which arrived between $\tau_\omega$ and $\tau_{\omega+1}$.

The aim of the challenge is to find an optimal policy $f : \mathcal{X} \to \mathcal{Y}$ assigning decisions $\boldsymbol{y}^\omega \in \mathcal{Y}(\boldsymbol{x}^\omega)$ to system states $\boldsymbol{x}^\omega \in \mathcal{X}$. This can be cast as a reinforcement learning problem:

$$\min_f \mathbb{E}\left[c_{\mathcal{W}}(f)\right], \quad \text{with} \quad c_{\mathcal{W}}(f) := \sum_{\omega \in \mathcal{W}} c(f(\boldsymbol{x}^\omega)),$$

where $c : \boldsymbol{y}^\omega \mapsto \sum_{i,j \in \mathcal{R}^\omega} c_{i,j}\, y_{i,j}^\omega$ gives the routing cost of $\boldsymbol{y}^\omega \in \mathcal{Y}^\omega$ and where $c_{i,j} \geq 0$ is the routing cost from $i$ to $j$. The expectation is taken over full problem instances.

### C.2  REDUCTION TO SUPERVISED LEARNING.

We follow the method of (Baty et al., 2023), which was the winning approach for the challenge. Central to this approach is the concept of prize-collecting dynamic vehicle routing problem with time windows (PC-VRPTW). In this setting, each request $i \in \mathcal{R}^\omega$ is assigned an artificial *prize* $\theta_i^\omega \in \mathbb{R}$, that reflects the benefit of serving it. The prize of the depot $D$ is set to $\theta_D^\omega = 0$. The objective is then to identify a set of routes that maximizes the total prize collected while minimizing the associated travel costs. The model $g_W$ predicts the prize vector $\boldsymbol{\theta}^\omega = g_W(\boldsymbol{x}^\omega)$. Denoting $\varphi(\boldsymbol{y}) := -\langle \boldsymbol{c}, \boldsymbol{y} \rangle$, the corresponding optimization problem can be written as:

$$\widehat{\boldsymbol{y}}(\boldsymbol{\theta}^\omega) = \underset{\boldsymbol{y} \in \mathcal{Y}(\boldsymbol{x}^\omega)}{\mathrm{argmax}} \sum_{i,j \in \mathcal{R}^\omega} \theta_j^\omega\, y_{i,j} - \sum_{i,j \in \mathcal{R}^\omega} c_{i,j}\, y_{i,j} = \langle \boldsymbol{\theta}^\omega, \boldsymbol{y} \rangle + \varphi(\boldsymbol{y}). \tag{12}$$

The overall pipeline is summarized in Fig. 1. Following (Baty et al., 2023), we approximately solve the problem in Eq. (12) using the prize-collecting HGS heuristic (PC-HGS), a variant of hybrid genetic search (HGS) (Vidal, 2022). We denote this approximate solver $\widetilde{\boldsymbol{y}} \approx \widehat{\boldsymbol{y}}$, so that their proposed policy decomposes as $f_W := \widetilde{\boldsymbol{y}} \circ g_W$. The ground-truth routes are created by using an anticipative strategy, i.e., by solving multiple instances where all future information is revealed from the start, and the requests' arrival times information is translated into time windows (thus removing the dynamic aspect of the problem). This anticipative policy, which we note $f^\star$ (which cannot be attained as it needs unavailable information) is thus the target policy imitated by the model – see Section C.8 for more details.

### C.3  PERTURBATION-BASED BASELINE.

In (Baty et al., 2023), a perturbation-based method (Berthet et al., 2020) was used. This method is based on injecting noise in the PC-HGS solver $\widetilde{\boldsymbol{y}}$. Similarly to our approach, the parameters $W$ can

| Name | Description |
|------|-------------|
| relocate | removes request $i$ from its route and re-inserts it before or after request $j$ |
| relocate pair | removes pair of requests $(i, \mathsf{next}(i))$ from their route and re-inserts them before or after request $j$ |
| swap | exchanges the position of requests $i$ and $j$ in the solution |
| swap pair | exchanges the positions of the pairs $(i, \mathsf{next}(i))$ and $(j, \mathsf{next}(j))$ in the solution |
| 2-opt | reverses the route segment between $i$ and $j$ |
| serve request | inserts currently undispatched request $i$ before or after request $j$ |
| remove request | removes currently dispatched request $i$ from the solution |

Table 5: PC-VRPTW Local search moves

then be learned using a Fenchel-Young loss, since the loss is minimized when the perturbed solver correctly predicts the ground truth. However, since $\widetilde{y}$ is not an exact solver, all theoretical learning guarantees associated with this method (e.g., correctness of the gradients) no longer hold.

### C.4 PROPOSED APPROACH.

Our proposed approach instead uses the Fenchel-Young loss associated with the proposed layer, which is minimized when the proposed layer correctly predicts the ground-truth. At inference time, however, we use $f_W := \widetilde{y} \circ g_W$. We use a mixture of proposals, as defined in Algorithm 2. To design each proposal $q_s$, we build randomized versions of moves specifically designed for the prize-collecting dynamic vehicle routing problem with time windows. More precisely, we base our proposals on moves used in the local search part of the PC-HGS algorithm, which are summarized in Table 2. The details of turning these moves into proposal distributions with tractable individual correction ratios are given in Section C.5.

We evaluate three different initialization methods: (i) initialize $\boldsymbol{y}^{(0)}$ by constructing routes dispatching random requests, (ii) initialize $\boldsymbol{y}^{(0)}$ to the ground-truth solution, (iii) initialize $\boldsymbol{y}^{(0)}$ by starting from the dataset ground-truth and applying a heuristic initialization algorithm to improve it. This heuristic initialization, similar to a short local search, is also used by the PC-HGS algorithm $\widetilde{y}$, and is set to take up to half the time allocated to the layer (a limit it does not reach in practice).

### C.5 PROPOSAL DISTRIBUTION DESIGN

**Original deterministic moves.** The selected moves, designed for Local Search algorithms on vehicle routing problems (specifically for the PC-VRPTW for `serve request` and `remove request`), are given in Table 5.

All of these moves (except for `remove request`) involve selecting two clients $i$ and $j$ from the request set $\mathcal{R}^\omega$ (for example, the `relocate` move relocates client $i$ after client $j$ in the solution).

In the Local Search part of the PC-HGS algorithm from Vidal (2022), they are implemented as deterministic functions used within a quadratic loop over clients, and are performed only if they improve the solution's objective value. The search is narrowed down to client pairs $(i, j)$ such that $d(i, j)$ is among the $N_{\mathrm{prox}}$ lowest values in $\{ d(i, k) \mid k \in \mathcal{R}^\omega \setminus \{D, i\} \}$, where $d$ is a problem-specific heuristic distance measure between clients, based on spatial features and time windows, and $N_{\mathrm{prox}}$ is a hyperparameter. These distances are independent from the chosen solution routes (they are computed once at the start of the algorithm, from the problem features), non-negative, and symmetric: $d(i, j) = d(j, i)$.

**Randomization.** In order to transform these deterministic moves into proposals, we first adapt the choice of clients $i$ and $j$, by sampling $i$ uniformly from $V_s^1(\boldsymbol{y})$, which contains the set of valid choices of client $i$ for move $s$ from solution $\boldsymbol{y}$. Then, we sample $j$ from $V_s^2(\boldsymbol{y})[i] \setminus \{i\}$ using the following softmax distribution: $P_s(j \mid i) = \frac{\exp[-d(i,j)/\beta]}{\sum_{k \in V_s^2(\boldsymbol{y})[i] \setminus \{i\}} \exp[-d(i,k)/\beta]}$, where $\beta > 0$ is a neighborhood sampling temperature. The set $V_s^2(\boldsymbol{y})[i]$ contains all valid choices of client $j$ for move $s$ from solution $\boldsymbol{y}$, and is precised along with $V_s^1(\boldsymbol{y})$ in Table 6. We normalize the distance measures inside the softmax, by dividing them by the maximum distance: $d(i, \cdot) \leftarrow d(i, \cdot) / \max_{k \in V_s^2(\boldsymbol{y})[i] \setminus \{i\}} d(i, k)$.

| Move | $V_s^1(\boldsymbol{y})$ | $V_s^2(\boldsymbol{y})[i]$ |
|---|---|---|
| `relocate` | $\mathcal{D}(\boldsymbol{y}) \setminus \mathcal{D}_1(\boldsymbol{y})$ | $\mathcal{D}(\boldsymbol{y})$ |
| `relocate pair` | $\mathcal{D}(\boldsymbol{y}) \setminus \{\mathcal{D}_2(\boldsymbol{y}) \cup \mathcal{D}^{\text{last}}(\boldsymbol{y})\}$ | $\mathcal{D}(\boldsymbol{y}) \setminus \{\mathsf{next}(i)\}$ |
| `swap` | $\mathcal{D}(\boldsymbol{y})$ | $\mathcal{D}(\boldsymbol{y})$ |
| `swap pair` | $\mathcal{D}(\boldsymbol{y}) \setminus \mathcal{D}^{\text{last}}(\boldsymbol{y})$ | $\mathcal{D}(\boldsymbol{y}) \setminus \{\mathcal{D}^{\text{last}}(\boldsymbol{y}) \cup \{\mathsf{prev}(i), \mathsf{next}(i)\}\}$ |
| `2-opt` | $\mathcal{D}(\boldsymbol{y}) \setminus \mathcal{D}_2(\boldsymbol{y})$ | $\mathcal{D}(\boldsymbol{y}) \setminus \mathcal{D}_2(\boldsymbol{y})$ |
| `serve request` | $\bar{\mathcal{D}}(\boldsymbol{y})$ | $\mathcal{D}(\boldsymbol{y}) \cup \mathcal{I}_D(\boldsymbol{y})$ |
| `remove request` | $\{\mathcal{D}(\boldsymbol{y}) \setminus \mathcal{D}_1(\boldsymbol{y})\} \cup \mathcal{I}_1(\boldsymbol{y})$ | |

Table 6: Sets of valid clients for each move. $\mathcal{D}(\boldsymbol{y})$ contains all dispatched clients in solution $\boldsymbol{y}$. $\mathcal{D}_1(\boldsymbol{y})$ contains all dispatched clients that are the only client in their route. $\mathcal{D}_2(\boldsymbol{y})$ contains all dispatched clients that are in a route with 2 clients or less. $\mathcal{D}^{\text{last}}(\boldsymbol{y})$ contains all dispatched clients that are the last of their route. $\bar{\mathcal{D}}(\boldsymbol{y})$ contains all non-dispatched clients. $\mathcal{I}_D(\boldsymbol{y})$ contains the depot of the first empty route, if it exists (all routes may be non-empty), or else is the empty set. $\mathcal{I}_1(\boldsymbol{y})$ contains the only client in the last non-empty route if it contains exactly one client, or else is the empty set.

**Neighborhood graph symmetrization.** Then, we ensure that each individual neighborhood graph $\mathcal{N}_s$ is undirected. This is already the case for the moves `swap`, `swap pair` and `2-opt`, as they are actually involutions (applying the same move on the same couple $(i, j)$ from $\boldsymbol{y}'$ will result in $\boldsymbol{y}$). However, this is obviously not the case for `serve request` and `remove request`. Indeed, if solution $\boldsymbol{y}'$ is obtained from $\boldsymbol{y}$ by removing a dispatched client (respectively serving an non-dispatched one), $\boldsymbol{y}$ cannot be obtained by removing another one (respectively, serving another one). To fix this, we merge these two moves into a single one. First, it evaluates which of the two moves are allowed (i.e., if they are such that $V_s^1(\boldsymbol{y}) \neq \emptyset$). Then, it samples one (the probability of selecting "remove" is chosen to be equal to the number of removable clients divided by the number of removable clients plus the number of servable clients) in the case where both are possible, or else simply performs the only move allowed. Thus, the corresponding neighborhood graph is undirected as it is always possible to perform the reverse operation (as when removing a client, it becomes unserved, thus allowing the `serve request` move from $\boldsymbol{y}'$, and vice-versa). We also allow the `serve request` move to insert a client after the depot of the first empty route, to allow the creation of new routes. In consequence, we allow the `remove request` move to remove the only client in the last non-empty route if it contains exactly one client (to maintain symmetry of the neighborhood graph).

For the `relocate` and `relocate pair` moves, the non-reversibility comes from the fact that they only relocate client $i$ (or clients $i$ and $\mathsf{next}(i)$ in the pair case) after client $j$, so that if client $i$ was the first in its route, relocating it back would be impossible (the depot, which is the start of the route, cannot be selected as $j$). Thus, we allow insertions before clients too, and add a random choice with probability $(\frac{1}{2}, \frac{1}{2})$ to determine if the relocated client(s) will be inserted before or after $j$. We also add this feature to the `serve request` move.

**Correction ratio computation.** Next, we implement the computation of the individual correction ratio $\tilde{\alpha}_s(\boldsymbol{y}, \boldsymbol{y}') = \frac{q_s(\boldsymbol{y}', \boldsymbol{y})}{q_s(\boldsymbol{y}, \boldsymbol{y}')}$ for each proposal $q_s$.

- In the case of `swap` and `2-opt`, we have $\tilde{\alpha}_s(\boldsymbol{y}, \boldsymbol{y}') = 1$. Indeed, let $\boldsymbol{y}'$ be the result of applying one of these moves $s$ on $\boldsymbol{y}$ when sampling $i \in V_s^1(\boldsymbol{y})$ and $j \in V_s^2(\boldsymbol{y})[i] \setminus \{i\}$. We then have:

$$q_s(\boldsymbol{y}, \boldsymbol{y}') = \frac{1}{|V_s^1(\boldsymbol{y})|} \cdot \frac{\exp[-d(i, j)/\beta]}{\sum_{k \in V_s^2(\boldsymbol{y})[i] \setminus \{i\}} \exp[-d(i, k)/\beta]}$$
$$+ \frac{1}{|V_s^1(\boldsymbol{y})|} \cdot \frac{\exp[-d(j, i)/\beta]}{\sum_{k \in V_s^2(\boldsymbol{y})[j] \setminus \{j\}} \exp[-d(j, k)/\beta]},$$

  where the first term accounts for the probability of selecting $i$ then $j$ and the second term accounts for that of selecting $j$ then $i$ (one can easily check that these two cases are the only way of sampling $\boldsymbol{y}'$ from $\boldsymbol{y}$). Then, noticing that we have $|V_s^1(\boldsymbol{y}')| = |V_s^1(\boldsymbol{y})|$, that these moves are involutions (selecting $(i, j)$ or $(j, i)$ from $\boldsymbol{y}'$ is also the only way to sample $\boldsymbol{y}$), and that we have the equalities $V_s^2(\boldsymbol{y})[i] = V_s^2(\boldsymbol{y}')[i]$ and $V_s^2(\boldsymbol{y})[j] = V_s^2(\boldsymbol{y}')[j]$, we actually have $q_s(\boldsymbol{y}', \boldsymbol{y}) = q_s(\boldsymbol{y}, \boldsymbol{y}')$.

- For `swap pair`, the same arguments hold (leading to the same form for $q_s$), except for the equalities $V_s^2(\boldsymbol{y})[i] = V_s^2(\boldsymbol{y}')[i]$ and $V_s^2(\boldsymbol{y})[j] = V_s^2(\boldsymbol{y}')[j]$. Thus, we have the following form for the correction ratio:

$$\frac{q_s(\boldsymbol{y}', \boldsymbol{y})}{q_s(\boldsymbol{y}, \boldsymbol{y}')} = \frac{\sum_{k \in V_s^2(\boldsymbol{y})[i] \setminus \{i\}} \exp\left[-d(i,k)/\beta\right] + \sum_{k \in V_s^2(\boldsymbol{y})[j] \setminus \{j\}} \exp\left[-d(j,k)/\beta\right]}{\sum_{k \in V_s^2(\boldsymbol{y}')[i] \setminus \{i\}} \exp\left[-d(i,k)/\beta\right] + \sum_{k \in V_s^2(\boldsymbol{y}')[j] \setminus \{j\}} \exp\left[-d(j,k)/\beta\right]}.$$

- In the case of `relocate`, let $j'$ denote $\mathsf{next}(j)$ if the selected insertion type was "after", and $\mathsf{prev}(j)$ if it was "before" – where $\mathsf{next}(j) \in \mathcal{R}^\omega$ denotes the request following $j$ in solution $\boldsymbol{y}$, i.e., the only index $k$ such that $\boldsymbol{y}_{j,k} = 1$, and $\mathsf{prev}(j)$ is the one preceding it, i.e., the only $k$ such that $\boldsymbol{y}_{k,j} = 1$. We have:

$$q_s(\boldsymbol{y}, \boldsymbol{y}') = \frac{1}{2} \cdot \frac{1}{|V_s^1(\boldsymbol{y})|} \cdot \frac{\exp\left[-d(i,j)/\beta\right]}{\sum_{k \in V_s^2(\boldsymbol{y})[i] \setminus \{i\}} \exp\left[-d(i,k)/\beta\right]}$$
$$+ \frac{1}{2} \cdot \frac{1}{|V_s^1(\boldsymbol{y})|} \cdot \frac{\exp\left[-d(i,j')/\beta\right]}{\sum_{k \in V_s^2(\boldsymbol{y})[i] \setminus \{i\}} \exp\left[-d(i,k)/\beta\right]}$$

Indeed, if $i$ was relocated *after* $j$, the same solution $\boldsymbol{y}'$ could have been obtained by relocating $i$ *before* $j' = \mathsf{next}(j)$. Similarly, if $i$ was relocated *before* $j$, the same solution $\boldsymbol{y}'$ could have been obtained by relocating $i$ *after* $j' = \mathsf{prev}(j)$. For the reverse move probability, the way of obtaining $\boldsymbol{y}$ from $\boldsymbol{y}'$ is either to select $(i, \mathsf{prev}(i))$ in the after-type insertion case, or $(i, \mathsf{next}(i))$ in the before-type insertion case (where prev and next are taken w.r.t. $\boldsymbol{y}$, i.e., before applying the move). Thus, we have:

$$q_s(\boldsymbol{y}', \boldsymbol{y}) = \frac{1}{2} \cdot \frac{1}{|V_s^1(\boldsymbol{y}')|} \cdot \frac{\exp\left[-d(i, \mathsf{prev}(i)/\beta\right]}{\sum_{k \in V_s^2(\boldsymbol{y}')[i] \setminus \{i\}} \exp\left[-d(i,k)/\beta\right]}$$
$$+ \frac{1}{2} \cdot \frac{1}{|V_s^1(\boldsymbol{y}')|} \cdot \frac{\exp\left[-d(i, \mathsf{next}(i))/\beta\right]}{\sum_{k \in V_s^2(\boldsymbol{y}')[i] \setminus \{i\}} \exp\left[-d(i,k)/\beta\right]}.$$

- For the `relocate pair` move, the exact same reasoning and proposal probability form hold for the forward move, but we have for the reverse direction:

$$q_s(\boldsymbol{y}', \boldsymbol{y}) = \frac{1}{2} \cdot \frac{1}{|V_s^1(\boldsymbol{y}')|} \cdot \frac{\exp\left[-d(i, \mathsf{prev}(i)/\beta\right]}{\sum_{k \in V_s^2(\boldsymbol{y}')[i] \setminus \{i\}} \exp\left[-d(i,k)/\beta\right]}$$
$$+ \frac{1}{2} \cdot \frac{1}{|V_s^1(\boldsymbol{y}')|} \cdot \frac{\exp\left[-d(i, \mathsf{next}(\mathsf{next}(i)))/\beta\right]}{\sum_{k \in V_s^2(\boldsymbol{y}')[i] \setminus \{i\}} \exp\left[-d(i,k)/\beta\right]},$$

as client $\mathsf{next}(i)$ is also relocated.

- For the `serve request` / `remove request` move, we have the forward probability:

$$q_s(\boldsymbol{y}, \boldsymbol{y}') = \frac{\left|\{\mathcal{D}(\boldsymbol{y}) \setminus \mathcal{D}_1(\boldsymbol{y})\} \cup \mathcal{I}_1(\boldsymbol{y})\right|}{\left|\{\mathcal{D}(\boldsymbol{y}) \setminus \mathcal{D}_1(\boldsymbol{y})\} \cup \mathcal{I}_1(\boldsymbol{y})\right| + |\bar{\mathcal{D}}(\boldsymbol{y})|} \times \frac{1}{\left|\{\mathcal{D}(\boldsymbol{y}) \setminus \mathcal{D}_1(\boldsymbol{y})\} \cup \mathcal{I}_1(\boldsymbol{y})\right|}$$
$$= \frac{1}{\left|\{\mathcal{D}(\boldsymbol{y}) \setminus \mathcal{D}_1(\boldsymbol{y})\} \cup \mathcal{I}_1(\boldsymbol{y})\right| + |\bar{\mathcal{D}}(\boldsymbol{y})|}$$

if the chosen move is `remove request`. The expression corresponds to the composition of move choice sampling and uniform sampling over removable clients.

Still in the same case (`remove request` is chosen) and if the removed request $i$ was in $\mathcal{I}_1(\boldsymbol{y})$ (i.e., was the only client in the last non-empty route if the latter contained exactly one

client), we have the reverse move probability:

$$q_s(\boldsymbol{y}', \boldsymbol{y}) = \frac{1}{|\{\mathcal{D}(\boldsymbol{y}') \setminus \mathcal{D}_1(\boldsymbol{y}')\} \cup \mathcal{I}_1(\boldsymbol{y}')| + |\bar{\mathcal{D}}(\boldsymbol{y}')|}$$
$$\times \frac{\exp\left[-\bar{d}(i)/\beta\right]}{\exp\left[-\bar{d}(i)/\beta\right] + \sum_{k \in \mathcal{D}(\boldsymbol{y}')} \exp\left[-d(i,k)/\beta\right]}$$
$$= \frac{1}{|\{\mathcal{D}(\boldsymbol{y}) \setminus \mathcal{D}_1(\boldsymbol{y})\} \cup \mathcal{I}_1(\boldsymbol{y})| + |\bar{\mathcal{D}}(\boldsymbol{y})|}$$
$$\times \frac{\exp\left[-\bar{d}(i)/\beta\right]}{\exp\left[-\bar{d}(i)/\beta\right] + \sum_{\substack{k \in \mathcal{D}(\boldsymbol{y}) \\ k \neq i}} \exp\left[-d(i,k)/\beta\right]}.$$

The expression corresponds to the composition of move choice sampling and softmax sampling of the depot of the first empty route (which was the route of the removed client $i$, so that $\mathcal{I}_D(\boldsymbol{y}') \neq \emptyset$ in this case). We use the average distance to dispatched clients $\bar{d}(i) := \frac{1}{|\mathcal{D}(\boldsymbol{y}')|} \sum_{k \in \mathcal{D}(\boldsymbol{y}')} d(i,k)$ as distance to the depot.

In the case where the removed request $i$ was not in $\mathcal{I}_1(\boldsymbol{y})$, we have instead:

$$q_s(\boldsymbol{y}', \boldsymbol{y}) = \frac{1}{|\{\mathcal{D}(\boldsymbol{y}') \setminus \mathcal{D}_1(\boldsymbol{y}')\} \cup \mathcal{I}_1(\boldsymbol{y}')| + |\bar{\mathcal{D}}(\boldsymbol{y}')|}$$
$$\times \frac{\frac{1}{2} \cdot \exp\left[-d(i, \mathsf{prev}(i))\right] + \frac{1}{2} \cdot \exp\left[-d(i, \mathsf{next}(i))\right]}{\mathbf{1}_{\{\mathcal{I}_D(\boldsymbol{y}') \neq \emptyset\}} \cdot \exp\left[-\bar{d}(i)/\beta\right] + \sum_{k \in \mathcal{D}(\boldsymbol{y}')} \exp\left[-d(i,k)/\beta\right]}$$
$$= \frac{1}{|\{\mathcal{D}(\boldsymbol{y}) \setminus \mathcal{D}_1(\boldsymbol{y})\} \cup \mathcal{I}_1(\boldsymbol{y})| + |\bar{\mathcal{D}}(\boldsymbol{y})|}$$
$$\times \frac{\frac{1}{2} \cdot \exp\left[-d(i, \mathsf{prev}(i))\right] + \frac{1}{2} \cdot \exp\left[-d(i, \mathsf{next}(i))\right]}{\mathbf{1}_{\{\mathcal{I}_D(\boldsymbol{y}') \neq \emptyset\}} \cdot \exp\left[-\bar{d}(i)/\beta\right] + \sum_{\substack{k \in \mathcal{D}(\boldsymbol{y}) \\ k \neq i}} \exp\left[-d(i,k)/\beta\right]}.$$

The right term corresponds to softmax sampling of the previous node with "after" insertion type (which has probability $1/2$) and of the next node with "before" insertion type. The non-emptiness of $\mathcal{I}_D(\boldsymbol{y}')$ is not guaranteed anymore, as all routes might be non-empty (indeed, we did not create an empty one by removing $i$, as $i \in \mathcal{D}(\boldsymbol{y}) \setminus \mathcal{D}_1(\boldsymbol{y})$ in this case).
Similarly, if the chosen move is `serve request`, we have the forward probability:

$$q_s(\boldsymbol{y}, \boldsymbol{y}') = \frac{|\bar{\mathcal{D}}(\boldsymbol{y})|}{|\{\mathcal{D}(\boldsymbol{y}) \setminus \mathcal{D}_1(\boldsymbol{y})\} \cup \mathcal{I}_1(\boldsymbol{y})| + |\bar{\mathcal{D}}(\boldsymbol{y})|}$$
$$\times \frac{\frac{1}{2} \cdot \exp\left[-d(i,j)\right] + \frac{1}{2} \cdot \exp\left[-d(i,j')\right]}{\mathbf{1}_{\{\mathcal{I}_D(\boldsymbol{y}) \neq \emptyset\}} \cdot \exp\left[-\bar{d}(i)/\beta\right] + \sum_{k \in \mathcal{D}(\boldsymbol{y})} \exp\left[-d(i,k)/\beta\right]}$$

if the selected insertion node $j$ is not in $\mathcal{I}_D(\boldsymbol{y})$ (i.e., is not the depot of the first empty route in $\boldsymbol{y}$), where $j' = \mathsf{prev}(j)$ if the insertion type selected was "before" (which has probability $1/2$), and $j' = \mathsf{next}(j)$ if it was "after".
We have instead the forward probability:

$$q_s(\boldsymbol{y}, \boldsymbol{y}') = \frac{1}{|\{\mathcal{D}(\boldsymbol{y}) \setminus \mathcal{D}_1(\boldsymbol{y})\} \cup \mathcal{I}_1(\boldsymbol{y})| + |\bar{\mathcal{D}}(\boldsymbol{y})|}$$
$$\times \frac{\exp\left[-\bar{d}(i)/\beta\right]}{\exp\left[-\bar{d}(i)/\beta\right] + \sum_{k \in \mathcal{D}(\boldsymbol{y})} \exp\left[-d(i,k)/\beta\right]}$$

if the selected insertion node $j$ is in $\mathcal{I}_D(\boldsymbol{y})$ (i.e., is the depot of the first empty route in $\boldsymbol{y}$).
In every case, we have the reverse move probability:

$$q_s(\boldsymbol{y}', \boldsymbol{y}) = \frac{1}{|\{\mathcal{D}(\boldsymbol{y}) \setminus \mathcal{D}_1(\boldsymbol{y})\} \cup \mathcal{I}_1(\boldsymbol{y})| + |\bar{\mathcal{D}}(\boldsymbol{y})|}.$$

In each case, we set $d(i, D) = \infty$ to account for the fact that the depot can never be sampled during the process (except in the `serve request` / `remove request` move, where we allow the depot of the first empty route / last non-empty route to be selected, for which we use the average distance to other requests as explained earlier) – in fact, the distance measure from a client to the depot is not even defined in the original HGS implementation.

The second correction factor needed is $\frac{|Q(\boldsymbol{y})|}{|Q(\boldsymbol{y}')|}$ (see Algorithm 2). We compute it by checking if each move is allowed, i.e., if there exists at least one $i \in V_s^1(\boldsymbol{y})$ such that $V_s^2(\boldsymbol{y})[i] \setminus \{i\} \neq \emptyset$. This can be determined in $\mathcal{O}(\mathcal{R}^\omega)$ for each move.

## C.6 PERFORMANCE METRIC.

As the Fenchel-Young loss $\ell_t$ actually minimized is intractable to compute exactly, we only use the challenge metric. More precisely, we measure the cost relative to that of the anticipative baseline, $\frac{c_\mathcal{W}(f_W) - c_\mathcal{W}(f^\star)}{c_\mathcal{W}(f^\star)}$, which we average over a test dataset of unseen instances.

## C.7 RESULTS.

In Fig. 2, we observe that the initialization method plays an important role, and the ground-truth-based ones greatly outperform the random one.

We observe that the number of Markov iterations $K$ is an important performance factor. Interestingly, the ground-truth initialization significantly improves the learning process for small $K$.

In Table 3, we compare training methods with fixed compute time budget for the layer (perturbed solver or proposed MCMC approach), which is by far the main computational bottleneck. This parameter limits the time allowed for a single forward pass through the combinatorial optimization layer (be it the perturbed inexact oracle or the proposed method). In both cases, the backward pass through the layer is immediate, as a property of the expression of the gradient of Fenchel-Young losses. The models are selected using a validation set and evaluated on the test set. We observe that the proposed approach significantly outperforms the perturbation-based method (Berthet et al., 2020) using $\widetilde{\boldsymbol{y}}$ in low time limit regimes, thus allowing for faster and more efficient training.

Full experimental details and additional results on the impact of temperature are given in Section C.8.

## C.8 ADDITIONAL EXPERIMENTAL DETAILS AND RESULTS FOR SECTION 5.1

**Model, features, dataset, hyperparameters, compute.** Following Baty et al. (2023), the differentiable ML model $g_W$ is implemented as a sparse graph neural network. We also use the same feature set, which represents the system state $\boldsymbol{x}^\omega$ as a vector comprising request-level features, such as coordinates, time windows, demands, travel time to the depot, and quantiles from the distribution of the travel time to all other requests (named *complete* feature set, and described in the Table 4 of their paper). We use the same training, validation, and testing datasets, which are created from 30, 15 and 25 problem instances respectively. The training set uses a sample size of 50 requests per wave, while the rest use 100. The solutions in the training dataset, i.e., the examples from the anticipative strategy $f^\star$ imitated by the model, are obtained by solving the corresponding offline VRPTWs using HGS (Vidal, 2022) with a time limit of 3600 seconds. During evaluation, the PC-HGS solver $\tilde{y}$ is used with a constant time limit of 60 seconds for all models. We use Adam (Kingma and Ba, 2017) together with the proposed stochastic gradient estimators, with a learning rate of $5 \cdot 10^{-3}$. Each training is performed using only a single CPU worker. For Fig. 2, we use a temperature $t = 10^2$. For Table 3, we use 1 Monte-Carlo sample for the perturbation-based method and 1 Markov chain for the proposed approach (in order to have a fair comparison: an equal number of oracle calls / equal compute).

**Statistical significance.** Each training is performed 50 times with the same parameters and different random seeds. Then, the learning curves are averaged, and plotted with a 95% confidence interval. For the results in Table 3, we report the performance of the best model iteration (selected with respect to the validation set) on the test set. This procedure is also averaged over 50 trainings, and reported with 95% confidence intervals.

**Additional results.** In Fig. 15, we report model performance for varying temperature $t$. Interestingly, lower temperatures perform better when using random initialization. In the ground-truth initialization setting, a sweet spot is found at $t = 10^2$, but lower temperatures do not particularly decrease performance.

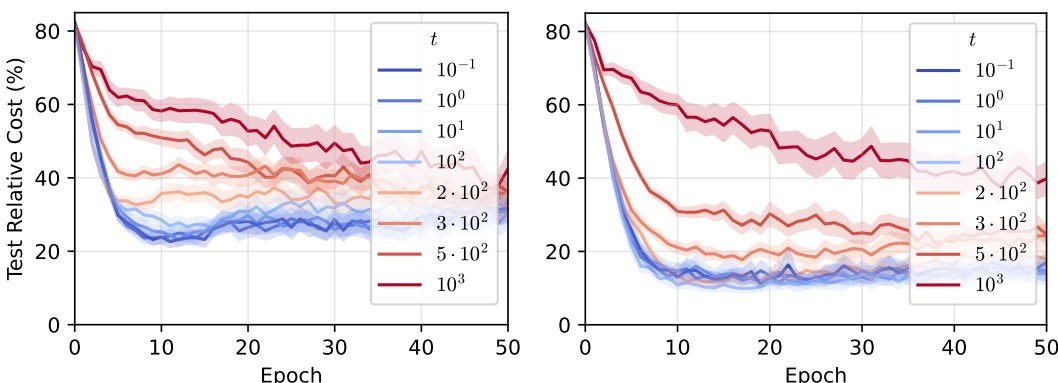

Figure 15: Test relative cost (%). **Left**: varying temperature $t$ with random initialization. **Right**: varying temperature $t$ with ground-truth initialization.

## D  DETAILS ON THE MULTI-DIMENSIONAL KNAPSACK PROBLEM

First, we recall that the combinatorial optimization layer is defined as:

$$\widehat{\boldsymbol{y}}(\boldsymbol{\theta}) \coloneqq \operatorname*{argmax}_{\boldsymbol{y} \in \{0,1\}^d} \sum_{i=1}^{d} \theta_i y_i \quad = \operatorname*{argmax}_{\boldsymbol{y} \in \mathcal{Y}} \langle \boldsymbol{\theta}, \boldsymbol{y} \rangle, \tag{13}$$

$$\text{s.t. } \forall j \in [M], \; \sum_{i=1}^{d} w_{i,j} y_i \leq C_j$$

where $\boldsymbol{\theta} = g_W(\boldsymbol{x}) \in \mathbb{R}^d$ are the item values, $w_{i,j} \geq 0$ is the weight of item $i$ in dimension $j$, and $C_j$ is the capacity of dimension $j$. The feasible set is $\mathcal{Y} \coloneqq \{\boldsymbol{y} \in \{0,1\}^d \mid \forall j \in [M], \; \sum_{i=1}^{d} w_{i,j} y_i \leq C_j\}$.

### D.1  PROPOSAL DISTRIBUTION DESIGN

In this experiment, defined in Section 5.2, we use Algorithm 2 with a mixture of three proposal distributions $q_1$, $q_2$ and $q_3$ ($S = 3$).

Let $\boldsymbol{y} \in \mathcal{Y}$ be a current feasible solution, and let $I(\boldsymbol{y}) = \{i \mid y_i = 1\}$ and $\bar{I}(\boldsymbol{y}) = \{j \mid y_j = 0\}$ denote the indices of selected and unselected items. Given a binary vector $\boldsymbol{y} \in \{0,1\}^d$ and an index $i \in [d]$, we denote by $\boldsymbol{y}_{y_i \to \bar{y_i}}$ the vector where the $i$-th bit is flipped, i.e.:

$$(\boldsymbol{y}_{y_i \to \bar{y_i}})_k = \begin{cases} 1 - y_i & \text{if } k = i, \\ y_i & \text{else.} \end{cases}$$

Given two indices $i, j \in [d]$, we denote by $\boldsymbol{y}_{i \leftrightarrow j} \in \{0,1\}^d$ the vector where the $i$-th and $j$-th bits are swapped, i.e.:

$$(\boldsymbol{y}_{i \leftrightarrow j})_k = \begin{cases} y_j & \text{if } k = i, \\ y_i & \text{if } k = j, \\ y_k & \text{else.} \end{cases}$$

We use a sampling temperature $\beta = 1.0$ and define the following moves:

- **Uniform swap ($q_1$).** The neighborhood $\mathcal{N}_1(\boldsymbol{y})$ consists of all feasible solutions obtained by swapping an active item $i \in I(\boldsymbol{y})$ with an inactive one $j \in \bar{I}(\boldsymbol{y})$, i.e.:

$$\mathcal{N}_1(\boldsymbol{y}) = \{\boldsymbol{y}' \in \mathcal{Y} \mid \exists i \in I(\boldsymbol{y}), j \in \bar{I}(\boldsymbol{y}) : \boldsymbol{y}' = \boldsymbol{y}_{i \leftrightarrow j}\}.$$

  The proposal is uniform over this neighborhood: $\forall \boldsymbol{y}_{i \leftrightarrow j} \in \mathcal{N}_1(\boldsymbol{y}), \ q_1(\boldsymbol{y}, \boldsymbol{y}_{i \leftrightarrow j}) = \frac{1}{|\mathcal{N}_1(\boldsymbol{y})|}$.

- **Guided swap ($q_2$).** Using the same swap neighborhood $\mathcal{N}_2(\boldsymbol{y}) = \mathcal{N}_1(\boldsymbol{y})$, we bias the selection using the predicted item values $\boldsymbol{\theta}$. We sample item $i \in I(\boldsymbol{y})$ to drop with probability $p_{\text{drop}}(i) \propto e^{-\theta_i/\beta}$ and item $j \in \bar{I}(\boldsymbol{y})$ to add with $p_{\text{add}}(j) \propto e^{\theta_j/\beta}$.

  The proposal distribution is therefore: $\forall \boldsymbol{y}_{i \leftrightarrow j} \in \mathcal{N}_1(\boldsymbol{y}), \ q_2(\boldsymbol{y}, \boldsymbol{y}_{i \leftrightarrow j}) \propto \exp\left(\frac{\theta_j - \theta_i}{\beta}\right)$.

- **Guided flip ($q_3$).** The neighborhood $\mathcal{N}_3(\boldsymbol{y})$ consists of all feasible solutions obtained by flipping a single bit $i$, i.e.:

$$\mathcal{N}_3(\boldsymbol{y}) = \{\boldsymbol{y}' \in \mathcal{Y} \mid \exists i \in [d] : \boldsymbol{y}' = \boldsymbol{y}_{y_i \to \bar{y}_i}\}.$$

  We sample index $i$ with probability proportional to $e^{-\theta_i/\beta}$ if $y_i = 1$ (favoring dropping low-value items) and $e^{\theta_i/\beta}$ if $y_i = 0$ (favoring adding high-value items).

  The proposal distribution is therefore: $\forall \boldsymbol{y}_{y_i \to \bar{y}_i} \in \mathcal{N}_3(\boldsymbol{y}), \ q_3(\boldsymbol{y}, \boldsymbol{y}_{y_i \to \bar{y}_i}) \propto \exp\left(\frac{(1-2y_i) \cdot \theta_i}{\beta}\right)$.

## D.2 DATA GENERATION

For the benchmark experiment in Fig. 3, we generate a synthetic dataset of $5,000$ instances using the `PyEPO` library (Tang and Khalil, 2023). We set the problem size to $d = 100$ items and $J = 50$ constraints. For each instance, we sample feature vectors $\boldsymbol{x} \in \mathbb{R}^{64}$ and generate the item values $\boldsymbol{\theta}$ (cost vector) with a polynomial dependence on $\boldsymbol{x}$ of degree $4$ and multiplicative noise $\epsilon = 0.5$. The item weights $w_{i,j}$ are sampled uniformly, and the capacities $C_j$ are generated using a capacity ratio of $0.5$.

To obtain the ground-truth labels $\boldsymbol{y}_i$ for the conditional learning task, we solve each instance using the Gurobi ILP solver with a time limit of 1000ms. The dataset is partitioned into training ($80\%$), validation ($10\%$), and test ($10\%$) sets. We use the validation set to select best model iterations (in terms of relative regret on the validation set), before evaluating their test relative regret.

## D.3 IMPLEMENTATION DETAILS

The predictive model $g_W$ is a Multi-Layer Perceptron (MLP) with two hidden layers of size $64$ and ReLU activations. We train the model for 20 epochs using the Adam (Kingma and Ba, 2017) optimizer with a learning rate of $5 \times 10^{-3}$ and a batch size of $32$.

For the benchmark experiment in Fig. 3, we use a time limit of 1.0ms for both the LS-MCMC layer and the Gurobi ILP solver at training time (at this scale, the solver consistently finds optimal solutions with this time budget).

# E PROOFS

## E.1 PROOF OF EQ. (4)

*Proof.* At fixed temperature $t_k = t$, the iterates of Algorithm 1 (MH case) follow a time-homogenous Markov chain, defined by the following transition kernel $P_{\boldsymbol{\theta},t}$:

$$P_{\boldsymbol{\theta},t}(\boldsymbol{y}, \boldsymbol{y}') = \begin{cases} q(\boldsymbol{y}, \boldsymbol{y}') \min\left[1, \frac{q(\boldsymbol{y}', \boldsymbol{y})}{q(\boldsymbol{y}, \boldsymbol{y}')} \exp\left(\frac{\langle \boldsymbol{\theta}, \boldsymbol{y}' \rangle + \varphi(\boldsymbol{y}') - \langle \boldsymbol{\theta}, \boldsymbol{y} \rangle - \varphi(\boldsymbol{y})}{t}\right)\right] & \text{if } \boldsymbol{y}' \in \mathcal{N}(\boldsymbol{y}), \\ 1 - \sum_{\boldsymbol{y}'' \in \mathcal{N}(\boldsymbol{y})} P_{\boldsymbol{\theta},t}(\boldsymbol{y}, \boldsymbol{y}'') & \text{if } \boldsymbol{y}' = \boldsymbol{y}, \\ 0 & \text{else.} \end{cases}$$

**Irreducibility.** As we assumed the neighborhood graph $G_{\mathcal{N}}$ to be connected and undirected, the Markov Chain is irreducible as we have $\forall \boldsymbol{y} \in \mathcal{Y}, \forall \boldsymbol{y}' \in \mathcal{N}(\boldsymbol{y}), P_{\boldsymbol{\theta},t}(\boldsymbol{y}, \boldsymbol{y}') > 0$.

**Aperiodicity.** For simplicity, we directly assumed aperiodicity in the main text. Here, we show that this is a mild condition, which is verified for instance if there is a solution $\boldsymbol{y} \in \mathcal{Y}$ such that $q(\boldsymbol{y}, \boldsymbol{y}) > 0$. Indeed, we then have:

$$
P_{\boldsymbol{\theta},t}(\boldsymbol{y}, \boldsymbol{y}) = 1 - \sum_{\boldsymbol{y}' \in \mathcal{N}(\boldsymbol{y})} P_{\boldsymbol{\theta},t}(\boldsymbol{y}, \boldsymbol{y}')
$$

$$
= 1 - \sum_{\boldsymbol{y}' \in \mathcal{N}(\boldsymbol{y})} q\left(\boldsymbol{y}, \boldsymbol{y}'\right) \min\left[1, \frac{q(\boldsymbol{y}', \boldsymbol{y})}{q(\boldsymbol{y}, \boldsymbol{y}')} \exp\left(\frac{\langle \boldsymbol{\theta},\, \boldsymbol{y}' \rangle + \varphi(\boldsymbol{y}') - \langle \boldsymbol{\theta},\, \boldsymbol{y} \rangle - \varphi(\boldsymbol{y})}{t}\right)\right]
$$

$$
\geq 1 - \sum_{\boldsymbol{y}' \in \mathcal{N}(\boldsymbol{y})} q\left(\boldsymbol{y}, \boldsymbol{y}'\right)
$$

$$
\geq q(\boldsymbol{y}, \boldsymbol{y}')
$$

$$
> 0.
$$

Thus, we have $P_{\boldsymbol{\theta},t}(\boldsymbol{y}, \boldsymbol{y}) > 0$, which implies that the chain is aperiodic. As an irreducible and aperiodic Markov Chain on a finite state space, it converges to its stationary distribution and the latter is unique (Freedman, 2017). Finally, one can easily check that the detailed balance equation is satisfied for $\pi_{\boldsymbol{\theta},t}$, i.e.:

$$
\forall \boldsymbol{y}, \boldsymbol{y}' \in \mathcal{Y}, \ \pi_{\boldsymbol{\theta},t}(\boldsymbol{y}) P_{\boldsymbol{\theta},t}(\boldsymbol{y}, \boldsymbol{y}') = \pi_{\boldsymbol{\theta},t}(\boldsymbol{y}') P_{\boldsymbol{\theta},t}(\boldsymbol{y}', \boldsymbol{y}),
$$

giving that $\pi_{\boldsymbol{\theta},t}$ is indeed the stationary distribution of the chain, which concludes the proof. $\qquad\square$

### E.2 PROOF OF PROPOSITION 1

*Proof.* Let $\boldsymbol{\theta} \in \mathbb{R}^d$ and $t > 0$. The fact that $\widehat{\boldsymbol{y}}_t(\boldsymbol{\theta}) \in \text{relint}(\mathcal{C}) = \text{relint}(\text{conv}(\mathcal{Y}))$ follows directly from the fact that $\widehat{\boldsymbol{y}}_t(\boldsymbol{\theta})$ is a convex combination of the elements of $\mathcal{Y}$ with positive coefficients, as $\forall \boldsymbol{y} \in \mathcal{Y}, \pi_{\boldsymbol{\theta},t}(\boldsymbol{y}) > 0$.

**Low temperature limit.** Let $\boldsymbol{y}^\star := \text{argmax}_{\boldsymbol{y} \in \mathcal{Y}} \langle \boldsymbol{\theta}, \boldsymbol{y} \rangle + \varphi(\boldsymbol{y})$. The argmax is assumed to be single-valued. Let $\boldsymbol{y} \in \mathcal{Y} \setminus \{\boldsymbol{y}^\star\}$. We have:

$$
\pi_{\boldsymbol{\theta},t}(\boldsymbol{y}) = \frac{\exp\left(\frac{\langle \boldsymbol{\theta}, \boldsymbol{y} \rangle + \varphi(\boldsymbol{y})}{t}\right)}{\sum_{\boldsymbol{y}' \in \mathcal{Y}} \exp\left(\frac{\langle \boldsymbol{\theta}, \boldsymbol{y}' \rangle + \varphi(\boldsymbol{y}')}{t}\right)}
$$

$$
\leq \frac{\exp\left(\frac{\langle \boldsymbol{\theta}, \boldsymbol{y} \rangle + \varphi(\boldsymbol{y})}{t}\right)}{\exp\left(\frac{\langle \boldsymbol{\theta}, \boldsymbol{y}^\star \rangle + \varphi(\boldsymbol{y}^\star)}{t}\right)}
$$

$$
\leq \exp\left(\frac{(\langle \boldsymbol{\theta}, \boldsymbol{y} \rangle + \varphi(\boldsymbol{y})) - (\langle \boldsymbol{\theta}, \boldsymbol{y}^\star \rangle + \varphi(\boldsymbol{y}^\star))}{t}\right)
$$

$$
\xrightarrow[t \to 0^+]{} 0,
$$

as $\langle \boldsymbol{\theta}, \boldsymbol{y} \rangle + \varphi(\boldsymbol{y}) < \langle \boldsymbol{\theta}, \boldsymbol{y}^\star \rangle + \varphi(\boldsymbol{y}^\star)$ by definition of $\boldsymbol{y}^\star$. Thus, we have:

$$
\pi_{\boldsymbol{\theta},t}(\boldsymbol{y}^\star) = 1 - \sum_{\boldsymbol{y} \in \mathcal{Y} \setminus \{\boldsymbol{y}^\star\}} \pi_{\boldsymbol{\theta},t}(\boldsymbol{y}) \xrightarrow[t \to 0^+]{} 1.
$$

Thus, the expectation of $\pi_{\boldsymbol{\theta},t}$ converges to $\boldsymbol{y}^\star$. Naturally, if the argmax is not unique, the distribution converges to a uniform distribution on the maximizing structures.

**High temperature limit.** For all $\boldsymbol{y} \in \mathcal{Y}$, we have:

$$
\pi_{\boldsymbol{\theta},t}(\boldsymbol{y}) = \frac{\exp\left(\frac{\langle \boldsymbol{\theta}, \boldsymbol{y} \rangle + \varphi(\boldsymbol{y})}{t}\right)}{\sum_{\boldsymbol{y}' \in \mathcal{Y}} \exp\left(\frac{\langle \boldsymbol{\theta}, \boldsymbol{y}' \rangle + \varphi(\boldsymbol{y}')}{t}\right)}
$$

$$
\xrightarrow[t \to \infty]{} \frac{1}{|\mathcal{Y}|},
$$

as $\exp(x/t) \xrightarrow[t\to\infty]{} 1$ for all $x \in \mathbb{R}$. Thus, $\pi_{\boldsymbol{\theta},t}$ converges to the uniform distribution on $\mathcal{Y}$, and its expectation converges to the average of all structures.

**Expression of the Jacobian.** Let $A_t : \boldsymbol{\theta} \mapsto t \cdot \log \sum_{\boldsymbol{y} \in \mathcal{Y}} \exp\left(\langle \boldsymbol{\theta}, \boldsymbol{y} \rangle + \varphi(\boldsymbol{y})\right)$ be the cumulant function of the exponential family defined by $\pi_{\boldsymbol{\theta},t}$, scaled by $t$. One can easily check that we have $\nabla_{\boldsymbol{\theta}} A_t(\boldsymbol{\theta}) = \widehat{\boldsymbol{y}}_t(\boldsymbol{\theta})$. Thus, we have $J_{\boldsymbol{\theta}} \widehat{\boldsymbol{y}}_t(\boldsymbol{\theta}) = \nabla^2_{\boldsymbol{\theta}} A_t(\boldsymbol{\theta})$. However, we also have that the hessian matrix of the cumulant function $\boldsymbol{\theta} \mapsto \frac{1}{t} A_t(\boldsymbol{\theta})$ is equal to the covariance matrix of the random vector $\frac{Y}{t}$ under $\pi_{\boldsymbol{\theta},t}$ (Wainwright and Jordan, 2008). Thus, we have:

$$
\begin{aligned}
J_{\boldsymbol{\theta}} \widehat{\boldsymbol{y}}_t(\boldsymbol{\theta}) &= \nabla^2_{\boldsymbol{\theta}} A_t(\boldsymbol{\theta}) \\
&= t \cdot \nabla^2_{\boldsymbol{\theta}} \left( \frac{1}{t} A_t(\boldsymbol{\theta}) \right) \\
&= t \cdot \text{cov}_{\pi_{\boldsymbol{\theta},t}} \left[ \frac{Y}{t} \right] \\
&= \frac{1}{t} \, \text{cov}_{\pi_{\boldsymbol{\theta},t}} [Y] \,.
\end{aligned}
$$

$\square$

### E.3   PROOF OF PROPOSITION 2

*Proof.* Let $K_{\boldsymbol{\theta},t}$ be the Markov transition kernel associated to Algorithm 2, which can be written as:

$$
K_{\boldsymbol{\theta},t}(\boldsymbol{y}, \boldsymbol{y}') = \begin{cases} \sum\limits_{\substack{s \in Q(\boldsymbol{y}) \\ \text{s.t. } q_s(\boldsymbol{y}, \boldsymbol{y}') > 0}} \frac{1}{|Q(\boldsymbol{y})|} q_s(\boldsymbol{y}, \boldsymbol{y}') \min\left(1, \frac{|Q(\boldsymbol{y})|}{|Q(\boldsymbol{y}')|} \cdot \frac{q_s(\boldsymbol{y}', \boldsymbol{y}) \pi_{\boldsymbol{\theta},t}(\boldsymbol{y}')}{q_s(\boldsymbol{y}, \boldsymbol{y}') \pi_{\boldsymbol{\theta},t}(\boldsymbol{y})}\right) & \text{if } \boldsymbol{y}' \in \bar{\mathcal{N}}(\boldsymbol{y}), \\ 1 - \sum_{\boldsymbol{y}'' \in \bar{\mathcal{N}}(\boldsymbol{y})} K_{\boldsymbol{\theta},t}(\boldsymbol{y}, \boldsymbol{y}'') & \text{if } \boldsymbol{y}' = \boldsymbol{y}, \\ 0 & \text{else.} \end{cases}
$$

As $\forall \boldsymbol{y} \in \mathcal{Y}$, $\forall \boldsymbol{y}' \in \bar{\mathcal{N}}(\boldsymbol{y}), K_{\boldsymbol{\theta},t}(\boldsymbol{y}, \boldsymbol{y}') > 0$, the irreducibility of the chain on $\mathcal{Y}$ is directly implied by the connectedness of $G_{\bar{\mathcal{N}}}$.

Thus, we only have to check that the detailed balance equation

$$
\pi_{\boldsymbol{\theta},t}(\boldsymbol{y}) K_{\boldsymbol{\theta},t}(\boldsymbol{y}, \boldsymbol{y}') = \pi_{\boldsymbol{\theta},t}(\boldsymbol{y}') K_{\boldsymbol{\theta},t}(\boldsymbol{y}', \boldsymbol{y})
$$

is satisfied for all $\boldsymbol{y}' \in \bar{\mathcal{N}}(\boldsymbol{y})$. We have:

$$
\pi_{\boldsymbol{\theta},t}(\boldsymbol{y}) K_{\boldsymbol{\theta},t}(\boldsymbol{y}, \boldsymbol{y}') = \sum_{\substack{s \in Q(\boldsymbol{y}) \\ \text{s.t. } q_s(\boldsymbol{y}, \boldsymbol{y}') > 0}} \left[ \frac{q_s(\boldsymbol{y}, \boldsymbol{y}') \pi_{\boldsymbol{\theta},t}(\boldsymbol{y})}{|Q(\boldsymbol{y})|} \min\left(1, \frac{|Q(\boldsymbol{y})|}{|Q(\boldsymbol{y}')|} \cdot \frac{q_s(\boldsymbol{y}', \boldsymbol{y}) \pi_{\boldsymbol{\theta},t}(\boldsymbol{y}')}{q_s(\boldsymbol{y}, \boldsymbol{y}') \pi_{\boldsymbol{\theta},t}(\boldsymbol{y})}\right) \right].
$$

The main point consists in noticing that the undirectedness assumption for each individual neighborhood graph $G_{\mathcal{N}_s}$ implies:

$$
\{s \in Q(\boldsymbol{y}) : q_s(\boldsymbol{y}, \boldsymbol{y}') > 0\} = \{s \in Q(\boldsymbol{y}') : q_s(\boldsymbol{y}', \boldsymbol{y}) > 0\}.
$$

Thus, a simple case analysis on how $|Q(\boldsymbol{y})| q_s(\boldsymbol{y}', \boldsymbol{y}) \pi_{\boldsymbol{\theta},t}(\boldsymbol{y}')$ and $|Q(\boldsymbol{y}')| q_s(\boldsymbol{y}, \boldsymbol{y}') \pi_{\boldsymbol{\theta},t}(\boldsymbol{y})$ compare allows us to observe that the expression of $\pi_{\boldsymbol{\theta},t}(\boldsymbol{y}) K_{\boldsymbol{\theta},t}(\boldsymbol{y}, \boldsymbol{y}')$ is symmetric in $\boldsymbol{y}$ and $\boldsymbol{y}'$, which concludes the proof. $\square$

### E.4   PROOF OF STRICT CONVEXITY

*Proof.* As $A_t$ is a differentiable convex function on $\mathbb{R}^d$ (as the log-sum-exp of such functions), it is an essentially smooth closed proper convex function. Thus, it is such that

$$
\text{relint}\left(\text{dom}((A_t)^*)\right) \subseteq \nabla A_t(\mathbb{R}^d) \subseteq \text{dom}((A_t)^*),
$$

and we have that the restriction of $(A_t)^*$ to $\nabla A_t(\mathbb{R}^d)$ is strictly convex on every convex subset of $\nabla A_t(\mathbb{R}^d)$ (corollary 26.4.1 in Rockafellar (1970)). As the range of the gradient of the cumulant

function $\boldsymbol{\theta} \mapsto A_t(\boldsymbol{\theta})/t$ is exactly the relative interior of the marginal polytope $\text{conv}(\{\boldsymbol{y}/t, \boldsymbol{y} \in \mathcal{Y}\})$ (see appendix B.1 in Wainwright and Jordan (2008)), and $(A_t)^* =: \Omega_t$, we actually have that

$$\text{relint}(\text{dom}(\Omega_t)) \subseteq \text{relint}(\mathcal{C}) \subseteq \text{dom}(\Omega_t),$$

and that $\Omega_t$ is stricly convex on every convex subset of $\text{relint}(\mathcal{C})$, i.e., strictly convex on $\text{relint}(\mathcal{C})$ (as $\text{relint}(\mathcal{C})$ is itself convex).

As $A_t$ is closed proper convex, it is equal to its biconjugate by the Fenchel-Moreau theorem. Thus, we have:

$$A_t(\boldsymbol{\theta}) = \sup_{\boldsymbol{\mu} \in \mathbb{R}^d} \{\langle \boldsymbol{\theta}, \boldsymbol{\mu} \rangle - (A_t)^*(\boldsymbol{\mu})\} = \sup_{\boldsymbol{\mu} \in \mathbb{R}^d} \{\langle \boldsymbol{\theta}, \boldsymbol{\mu} \rangle - \Omega_t(\boldsymbol{\mu})\}.$$

Moreover, as $\nabla A_t(\mathbb{R}^d) = \text{relint}(\mathcal{C})$, we have $||\nabla A_t(\boldsymbol{\theta})|| \leq R_{\mathcal{C}} := \max_{\boldsymbol{\mu} \in \mathcal{C}} ||\boldsymbol{\mu}||$, which gives $\text{dom}(\Omega_t) \subset B(\boldsymbol{0}, R_{\mathcal{C}})$. Thus we can actually write:

$$A_t(\boldsymbol{\theta}) = \sup_{\boldsymbol{\mu} \in B(\boldsymbol{0}, R_{\mathcal{C}})} \{\langle \boldsymbol{\theta}, \boldsymbol{\mu} \rangle - \Omega_t(\boldsymbol{\mu})\},$$

and now apply Danskin's theorem as $B(\boldsymbol{0}, R_{\mathcal{C}})$ is compact, which further gives:

$$\partial A_t(\boldsymbol{\theta}) = \underset{\boldsymbol{\mu} \in B(\boldsymbol{0}, R_{\mathcal{C}})}{\text{argmax}} \{\langle \boldsymbol{\theta}, \boldsymbol{\mu} \rangle - \Omega_t(\boldsymbol{\mu})\},$$

and the fact that $A_t$ is differentiable gives that both sides are single-valued. Moreover, as $\nabla A_t(\mathbb{R}^d) = \text{relint}(\mathcal{C})$, we know that the right hand side is maximized in $\mathcal{C}$, and we can actually write:

$$\nabla A_t(\boldsymbol{\theta}) = \underset{\boldsymbol{\mu} \in \mathcal{C}}{\text{argmax}} \{\langle \boldsymbol{\theta}, \boldsymbol{\mu} \rangle - \Omega_t(\boldsymbol{\mu})\}.$$

We end this proof by noting that a simple calculation yields $\nabla A_t(\boldsymbol{\theta}) = \mathbb{E}_{\pi_{\boldsymbol{\theta},t}}[Y] = \widehat{\boldsymbol{y}}_t(\boldsymbol{\theta})$. The expression of $\nabla_{\boldsymbol{\theta}} \ell_t(\boldsymbol{\theta}\,;\boldsymbol{y})$ follows. $\qquad\square$

> **Remark 2.** *The proposed Fenchel-Young loss can also be obtained via distribution-space regularization. Let $s_{\boldsymbol{\theta}} := (\langle \boldsymbol{\theta}, \boldsymbol{y} \rangle + \varphi(\boldsymbol{y}))_{\boldsymbol{y} \in \mathcal{Y}} \in \mathbb{R}^{|\mathcal{Y}|}$ be a vector containing the score of all structures, and $L_{-tH} : \mathbb{R}^{|\mathcal{Y}|} \times \Delta^{|\mathcal{Y}|} \to \mathbb{R}$ be the Fenchel-Young loss generated by $-tH$, where $H$ is the Shannon entropy. We have $\nabla_{s_{\boldsymbol{\theta}}}(-tH)^*(s_{\boldsymbol{\theta}}) = \pi_{\boldsymbol{\theta},t}$. The chain rule further gives $\nabla_{\boldsymbol{\theta}}(-tH)^*(s_{\boldsymbol{\theta}}) = \mathbb{E}_{\pi_{\boldsymbol{\theta},t}}[Y]$. Thus, we have $\nabla_{\boldsymbol{\theta}} L_{-tH}(s_{\boldsymbol{\theta}}\,;\boldsymbol{p}_{\boldsymbol{y}}) = \nabla_{\boldsymbol{\theta}} \ell_t(\boldsymbol{\theta}\,;\boldsymbol{y})$, where $\boldsymbol{p}_{\boldsymbol{y}}$ is the Dirac distribution on $\boldsymbol{y}$. In the case where $\varphi \equiv 0$ and $t = 1$, we have $\Omega_t(\boldsymbol{\mu}) = -(\max_{\boldsymbol{p} \in \Delta^{|\mathcal{Y}|}} H^s(\boldsymbol{p}) \text{ s.t. } \mathbb{E}_{\boldsymbol{p}}[Y] = \boldsymbol{\mu})$, with $H^s$ the Shannon entropy (Blondel et al., 2020), and $\ell_t$ is known as the CRF loss (Lafferty et al., 2001).*

## E.5 PROOF OF PROPOSITION 4

*Proof.* The proof is exactly the proof of Proposition 4.1 in Berthet et al. (2020), in which the setting is similar, and all the same arguments hold (we also have that $\pi_{\boldsymbol{\theta}_0}$ is dense on $\mathcal{Y}$, giving $\bar{Y}_N \in \text{relint}(\mathcal{C})$ for $N$ large enough). The only difference is the choice of regularization function, and we have to prove that it is also convex and smooth in our case. While the convexity of $\Omega_t$ is directly implied by its definition as a Fenchel conjugate, the fact that is is smooth is due to Theorem 26.3 in Rockafellar (1970) and the essential strict convexity of $A_t$ (which is itself closed proper convex). The latter relies on the fact that $\mathcal{C}$ is assumed to be of full-dimension (otherwise $A_t$ would be linear when restricted to any affine subspace of direction equal to the subspace orthogonal to the direction of the smallest affine subspace spanned by $\mathcal{C}$), which in turn implies that $A_t$ is strictly convex on $\mathbb{R}^d$. Thus, Proposition 4.1 in Berthet et al. (2020) gives the asymptotic normality:

$$\sqrt{N}(\boldsymbol{\theta}_N^\star - \boldsymbol{\theta}_0) \xrightarrow[N \to \infty]{\mathcal{D}} \mathcal{N}\left(\boldsymbol{0}, (\nabla_{\boldsymbol{\theta}}^2 A_t(\boldsymbol{\theta}_0))^{-1} \text{cov}_{\pi_{\boldsymbol{\theta}_0,t}}[Y] (\nabla_{\boldsymbol{\theta}}^2 A_t(\boldsymbol{\theta}_0))^{-1}\right).$$

Moreover, we already derived $\nabla_{\boldsymbol{\theta}}^2 A_t(\boldsymbol{\theta}_0) = \frac{1}{t} \text{cov}_{\pi_{\boldsymbol{\theta}_0,t}}[Y]$ in Section E.2, leading to the simplified asymptotic normality given in the proposition.

$\qquad\square$

*Proof.* The proof consists in bounding the convergence rate of the Markov chain $\left(\boldsymbol{y}^{(k)}\right)_{k\in\mathbb{N}}$ (which has transition kernel $P_{\boldsymbol{\theta},t}$) for all $\boldsymbol{\theta}$, in order to apply Theorem 4.1 in Younes (1998). It is defined as the smallest constant $\lambda_{\boldsymbol{\theta}}$ such that:

$$\exists A > 0 \,:\, \forall \boldsymbol{y} \in \mathcal{Y}, \; |\mathbb{P}(\boldsymbol{y}^{(k)} = \boldsymbol{y}) - \pi_{\boldsymbol{\theta},t}(\boldsymbol{y})| \le A\lambda_{\boldsymbol{\theta}}^k.$$

More precisely, we must find a constant $D$ such that $\exists B > 0 \,:\, \lambda_{\boldsymbol{\theta}} \le 1 - Be^{-D||\boldsymbol{\theta}||}$, in order to impose $K_{n+1} > \left\lfloor 1 + a' \exp\left(2D||\hat{\boldsymbol{\theta}}_n||\right) \right\rfloor$.

A known result gives $\lambda_{\boldsymbol{\theta}} \le \rho(\boldsymbol{\theta})$ with $\rho(\boldsymbol{\theta}) = \max_{\lambda \in S_{\boldsymbol{\theta}}\setminus\{1\}} |\lambda|$ (Madras and Randall, 2002), where $S_{\boldsymbol{\theta}}$ is the spectrum of the transition kernel $P_{\boldsymbol{\theta},t}$ (in this context, $1 - \rho(\boldsymbol{\theta})$ is known as the *spectral gap* of the Markov chain). To bound $\rho(\boldsymbol{\theta})$, we use the results of Ingrassia (1994), which study the Markov chain with transition kernel $P'_{\boldsymbol{\theta},t}$, such that $P_{\boldsymbol{\theta},t} = \frac{1}{2}\left(I + P'_{\boldsymbol{\theta},t}\right)$. It corresponds to the same algorithm, but with a proposal distribution $q'$ defined as:

$$q'\left(\boldsymbol{y},\boldsymbol{y}'\right) = \begin{cases} \frac{1}{d^*} & \text{if } \boldsymbol{y}' \in \mathcal{N}(\boldsymbol{y}), \\ 1 - \frac{d(\boldsymbol{y})}{d^*} & \text{if } \boldsymbol{y}' = \boldsymbol{y}, \\ 0 & \text{else.} \end{cases}$$

As $P'_{\boldsymbol{\theta},t}$ is a row-stochastic matrix, Gershgorin's circle theorem gives that its spectrum is included in the complex unit disc. Moreover, one can easily check that the associated Markov chain is also reversible with respect to $\pi_{\boldsymbol{\theta},t}$, and the corresponding detailed balance equation gives:

$$\forall \boldsymbol{y},\boldsymbol{y}' \in \mathcal{Y}, \; \pi_{\boldsymbol{\theta},t}(\boldsymbol{y})P'_{\boldsymbol{\theta},t}(\boldsymbol{y},\boldsymbol{y}') = \pi_{\boldsymbol{\theta},t}(\boldsymbol{y}')P'_{\boldsymbol{\theta},t}(\boldsymbol{y}',\boldsymbol{y}),$$

which is equivalent to:

$$\forall \boldsymbol{y},\boldsymbol{y}' \in \mathcal{Y}, \; \sqrt{\frac{\pi_{\boldsymbol{\theta},t}(\boldsymbol{y})}{\pi_{\boldsymbol{\theta},t}(\boldsymbol{y}')}} P'_{\boldsymbol{\theta},t}(\boldsymbol{y},\boldsymbol{y}') = \sqrt{\frac{\pi_{\boldsymbol{\theta},t}(\boldsymbol{y}')}{\pi_{\boldsymbol{\theta},t}(\boldsymbol{y})}} P'_{\boldsymbol{\theta},t}(\boldsymbol{y}',\boldsymbol{y})$$

as $\pi_{\boldsymbol{\theta},t}$ has full support on $\mathcal{Y}$, which can be further written in matrix form as:

$$\Pi_{\boldsymbol{\theta}}^{1/2} P'_{\boldsymbol{\theta},t} \Pi_{\boldsymbol{\theta}}^{-1/2} = \Pi_{\boldsymbol{\theta}}^{-1/2} P'^{\top}_{\boldsymbol{\theta},t} \Pi_{\boldsymbol{\theta}}^{1/2},$$

where $\Pi_{\boldsymbol{\theta}} = \text{diag}(\pi_{\boldsymbol{\theta};t})$. Thus, the matrix $\Pi_{\boldsymbol{\theta}}^{1/2} P'_{\boldsymbol{\theta},t} \Pi_{\boldsymbol{\theta}}^{-1/2}$ is symmetric, and the spectral theorem ensures its eigenvalues are real. As it is similar to the transition kernel $P'_{\boldsymbol{\theta},t}$ (with change of basis matrix $\Pi_{\boldsymbol{\theta}}^{-1/2}$), they share the same spectrum $S'_{\boldsymbol{\theta}}$, and we have $S'_{\boldsymbol{\theta}} \subset [-1,1]$. Let us order $S'_{\boldsymbol{\theta}}$ as $-1 \le \lambda'_{\min} \le \cdots \le \lambda'_2 \le \lambda'_1 = 1$. As $P_{\boldsymbol{\theta},t} = \frac{1}{2}\left(I + P'_{\boldsymbol{\theta},t}\right)$, we clearly have $\rho(\boldsymbol{\theta}) = \frac{1+\lambda'_2}{2}$. Thus, we can use Theorem 4.1 of Ingrassia (1994), which gives $\lambda'_2 \le 1 - G \cdot Z(\boldsymbol{\theta})\exp(-m\left(\boldsymbol{\theta}\right))$ (we keep their notations for $Z$ and $m$, and add the dependency in $\boldsymbol{\theta}$ for clarity), where $G$ is a constant depending only on the graph $G_{\mathcal{N}}$, and with:

$$\begin{aligned} Z(\boldsymbol{\theta}) &= \sum_{\boldsymbol{y}\in\mathcal{Y}} \exp\left(\frac{\langle\boldsymbol{\theta},\boldsymbol{y}\rangle + \varphi(\boldsymbol{y})}{t} - \max_{\boldsymbol{y}'\in\mathcal{Y}}\left[\frac{\langle\boldsymbol{\theta},\boldsymbol{y}'\rangle + \varphi(\boldsymbol{y}')}{t}\right]\right) \\ &\ge |\mathcal{Y}|\exp\left(\frac{1}{t}\left[\min_{\boldsymbol{y}\in\mathcal{Y}}\langle\boldsymbol{\theta},\boldsymbol{y}\rangle + \min_{\boldsymbol{y}\in\mathcal{Y}}\varphi(\boldsymbol{y}) - \max_{\boldsymbol{y}'\in\mathcal{Y}}\langle\boldsymbol{\theta},\boldsymbol{y}'\rangle - \max_{\boldsymbol{y}'\in\mathcal{Y}}\varphi(\boldsymbol{y}')\right]\right) \\ &\ge |\mathcal{Y}|\exp\left(-\frac{2R_{\mathcal{C}}}{t}||\boldsymbol{\theta}|| - \frac{2R_{\varphi}}{t}\right), \end{aligned}$$

and:

$$m(\boldsymbol{\theta}) \leq \max_{\boldsymbol{y}\in\mathcal{Y}}\left\{\max_{\boldsymbol{y}'\in\mathcal{Y}}\left[\frac{\langle\boldsymbol{\theta},\,\boldsymbol{y}'\rangle+\varphi(\boldsymbol{y}')}{t}\right]-\frac{\langle\boldsymbol{\theta},\boldsymbol{y}\rangle+\varphi(\boldsymbol{y})}{t}\right\} - 2\min_{\boldsymbol{y}\in\mathcal{Y}}\left\{\max_{\boldsymbol{y}'\in\mathcal{Y}}\left[\frac{\langle\boldsymbol{\theta},\,\boldsymbol{y}'\rangle+\varphi(\boldsymbol{y}')}{t}\right]-\frac{\langle\boldsymbol{\theta},\boldsymbol{y}\rangle+\varphi(\boldsymbol{y})}{t}\right\}$$

$$= \max_{\boldsymbol{y}'\in\mathcal{Y}}\left[\frac{\langle\boldsymbol{\theta},\,\boldsymbol{y}'\rangle+\varphi(\boldsymbol{y}')}{t}\right]-\min_{\boldsymbol{y}\in\mathcal{Y}}\left[\frac{\langle\boldsymbol{\theta},\boldsymbol{y}\rangle+\varphi(\boldsymbol{y})}{t}\right]$$

$$\leq \frac{1}{t}\left(\max_{\boldsymbol{y}'\in\mathcal{Y}}\langle\boldsymbol{\theta},\,\boldsymbol{y}'\rangle+\max_{\boldsymbol{y}'\in\mathcal{Y}}\varphi(\boldsymbol{y}')-\min_{\boldsymbol{y}\in\mathcal{Y}}\langle\boldsymbol{\theta},\,\boldsymbol{y}\rangle-\min_{\boldsymbol{y}\in\mathcal{Y}}\varphi(\boldsymbol{y})\right)$$

$$\leq \frac{2R_{\mathcal{C}}}{t}\|\boldsymbol{\theta}\|+\frac{2R_{\varphi}}{t},$$

where $R_{\mathcal{C}}=\max_{\boldsymbol{y}\in\mathcal{Y}}\|\boldsymbol{y}\|$ and $R_{\varphi}=\max_{\boldsymbol{y}\in\mathcal{Y}}|\varphi(\boldsymbol{y})|$. Thus, we have:

$$\lambda_2' \leq 1 - G|\mathcal{Y}|\exp\left(-\frac{4R_{\varphi}}{t}\right)\exp\left(-\frac{4R_{\mathcal{C}}}{t}\|\boldsymbol{\theta}\|\right),$$

and finally:

$$\lambda_{\boldsymbol{\theta}} \leq 1 - \frac{G|\mathcal{Y}|\exp\left(-\frac{4R_{\varphi}}{t}\right)}{2}\exp\left(-\frac{4R_{\mathcal{C}}}{t}\|\boldsymbol{\theta}\|\right),$$

so taking $D=4R_{\mathcal{C}}/t$ concludes the proof.

$\square$

> **Remark 3.** *The stationary distribution in* Ingrassia *(1994) is defined as proportional to* $\exp\left(-H(\boldsymbol{y})\right)$, *with the assumption that the function $H$ is such that $\min_{\boldsymbol{y}\in\mathcal{Y}}H(\boldsymbol{y})=0$. Thus, we apply their results with*
>
> $$H(\boldsymbol{y}) := \max_{\boldsymbol{y}'\in\mathcal{Y}}\left[\frac{\langle\boldsymbol{\theta},\,\boldsymbol{y}'\rangle+\varphi(\boldsymbol{y}')}{t}\right]-\frac{\langle\boldsymbol{\theta},\boldsymbol{y}\rangle+\varphi(\boldsymbol{y})}{t}$$
>
> *(which gives correct distribution $\pi_{\boldsymbol{\theta},t}$ and respects this assumption), hence the obtained forms for $Z(\boldsymbol{\theta})$ and the upper bound on $m(\boldsymbol{\theta})$.*

### E.7 PROOFS OF PROPOSITION 3 AND PROPOSITION 6

*Proposition 3.* The distribution of the first iterate of the Markov chain with transition kernel defined in Eq. (3) and initialized at the ground-truth structure $\boldsymbol{y}$ is given by:

$$(\boldsymbol{p}_{\boldsymbol{\theta},\boldsymbol{y}}^{(1)})(\boldsymbol{y}') = P_{\boldsymbol{\theta},t}(\boldsymbol{y},\boldsymbol{y}')$$

$$= \begin{cases} q(\boldsymbol{y},\boldsymbol{y}')\min\left[1,\frac{q(\boldsymbol{y}',\boldsymbol{y})}{q(\boldsymbol{y},\boldsymbol{y}')}\exp\left(\left[\langle\boldsymbol{\theta},\boldsymbol{y}'-\boldsymbol{y}\rangle+\varphi(\boldsymbol{y}')-\varphi(\boldsymbol{y})\right]/t\right)\right] & \text{if } \boldsymbol{y}'\in\mathcal{N}(\boldsymbol{y}), \\ 1-\sum_{\boldsymbol{y}''\in\mathcal{N}(\boldsymbol{y})}(\boldsymbol{p}_{\boldsymbol{\theta},\boldsymbol{y}}^{(1)})(\boldsymbol{y}'') & \text{if } \boldsymbol{y}'=\boldsymbol{y}, \\ 0 & \text{else.} \end{cases}$$

Let $\alpha_{\boldsymbol{y}}(\boldsymbol{\theta},\boldsymbol{y}') := \frac{q(\boldsymbol{y}',\boldsymbol{y})}{q(\boldsymbol{y},\boldsymbol{y}')}\exp\left(\left[\langle\boldsymbol{\theta},\boldsymbol{y}'-\boldsymbol{y}\rangle+\varphi(\boldsymbol{y}')-\varphi(\boldsymbol{y})\right]/t\right)$. Define also the following sets:

$$\mathcal{N}_{\boldsymbol{y}}^{-}(\boldsymbol{\theta}) = \{\boldsymbol{y}'\in\mathcal{N}(\boldsymbol{y}) \mid \alpha_{\boldsymbol{y}}(\boldsymbol{\theta},\boldsymbol{y}')\leq 1\}, \quad \mathcal{N}_{\boldsymbol{y}}^{+}(\boldsymbol{\theta}) = \{\boldsymbol{y}'\in\mathcal{N}(\boldsymbol{y}) \mid \alpha_{\boldsymbol{y}}(\boldsymbol{\theta},\boldsymbol{y}')> 1\}.$$

The expectation of the first iterate is then given by:

$$\mathbb{E}_{\boldsymbol{p}_{\boldsymbol{\theta},\boldsymbol{y}}^{(1)}}[Y] = \sum_{\boldsymbol{y}'\in\mathcal{N}(\boldsymbol{y})}(\boldsymbol{p}_{\boldsymbol{\theta},\boldsymbol{y}}^{(1)})(\boldsymbol{y}')\cdot\boldsymbol{y}' + \left(1-\sum_{\boldsymbol{y}''\in\mathcal{N}(\boldsymbol{y})}(\boldsymbol{p}_{\boldsymbol{\theta},\boldsymbol{y}}^{(1)})(\boldsymbol{y}'')\right)\cdot\boldsymbol{y}$$

$$= \boldsymbol{y} + \sum_{\boldsymbol{y}'\in\mathcal{N}(\boldsymbol{y})}(\boldsymbol{p}_{\boldsymbol{\theta},\boldsymbol{y}}^{(1)})(\boldsymbol{y}')\cdot(\boldsymbol{y}'-\boldsymbol{y})$$

$$= \boldsymbol{y} + \sum_{\boldsymbol{y}'\in\mathcal{N}_{\boldsymbol{y}}^{-}(\boldsymbol{\theta})}q(\boldsymbol{y}',\boldsymbol{y})\exp\left(\left[\langle\boldsymbol{\theta},\boldsymbol{y}'-\boldsymbol{y}\rangle+\varphi(\boldsymbol{y}')-\varphi(\boldsymbol{y})\right]/t\right)\cdot(\boldsymbol{y}'-\boldsymbol{y}) + \sum_{\boldsymbol{y}'\in\mathcal{N}_{\boldsymbol{y}}^{+}(\boldsymbol{\theta})}q(\boldsymbol{y},\boldsymbol{y}')\cdot(\boldsymbol{y}'-\boldsymbol{y}).$$

Let now $f_{\boldsymbol{y}} : \mathbb{R}^d \times \mathcal{N}(\boldsymbol{y}) \to \mathbb{R}$ be defined as:

$$f_{\boldsymbol{y}} : (\boldsymbol{\theta}\,;\boldsymbol{y}') \mapsto \begin{cases} t \cdot q(\boldsymbol{y}',\boldsymbol{y}) \exp\left(\left[\langle \boldsymbol{\theta}, \boldsymbol{y}' - \boldsymbol{y}\rangle + \varphi(\boldsymbol{y}') - \varphi(\boldsymbol{y})\right]/t\right) & \text{if } \alpha_{\boldsymbol{y}}(\boldsymbol{\theta},\boldsymbol{y}') \leq 1, \\ t \cdot q(\boldsymbol{y},\boldsymbol{y}')\left(\left[\langle \boldsymbol{\theta}, \boldsymbol{y}' - \boldsymbol{y}\rangle + \varphi(\boldsymbol{y}') - \varphi(\boldsymbol{y})\right]/t + 1 - \log\frac{q(\boldsymbol{y},\boldsymbol{y}')}{q(\boldsymbol{y}',\boldsymbol{y})}\right) & \text{if } \alpha_{\boldsymbol{y}}(\boldsymbol{\theta},\boldsymbol{y}') > 1. \end{cases}$$

Let $F_{\boldsymbol{y}} : \boldsymbol{\theta} \mapsto \langle \boldsymbol{\theta}, \boldsymbol{y}\rangle + \sum_{\boldsymbol{y}' \in \mathcal{N}(\boldsymbol{y})} f_{\boldsymbol{y}}(\boldsymbol{\theta}\,;\boldsymbol{y}')$. We define the target-dependent regularization function $\Omega_{\boldsymbol{y}}$ and the corresponding Fenchel-Young loss as:

$$\Omega_{\boldsymbol{y}} : \boldsymbol{\mu} \mapsto (F_{\boldsymbol{y}})^*(\boldsymbol{\mu}), \qquad L_{\Omega_{\boldsymbol{y}}}(\boldsymbol{\theta}\,;\boldsymbol{y}) \coloneqq (\Omega_{\boldsymbol{y}})^*(\boldsymbol{\theta}) + \Omega_{\boldsymbol{y}}(\boldsymbol{y}) - \langle \boldsymbol{\theta}, \boldsymbol{y}\rangle.$$

- $\underline{\Omega_{\boldsymbol{y}} \text{ is } t/\mathbb{E}_{q(\boldsymbol{y},\cdot)}\|Y - \boldsymbol{y}\|_2^2\text{-strongly convex:}}$

One can easily check that $f_{\boldsymbol{y}}(\,\cdot\,;\boldsymbol{y}')$ is continuous for all $\boldsymbol{y}' \in \mathcal{N}(\boldsymbol{y})$, as it is defined piecewise as continuous functions that match on the junction affine hyperplane defined by:

$$\left\{\boldsymbol{\theta} \in \mathbb{R}^d \mid \alpha_{\boldsymbol{y}}(\boldsymbol{\theta};\boldsymbol{y}') = 1\right\} = \left\{\boldsymbol{\theta} \in \mathbb{R}^d \mid \langle \boldsymbol{\theta}, \boldsymbol{y}' - \boldsymbol{y}\rangle = t \log\frac{q(\boldsymbol{y},\boldsymbol{y}')}{q(\boldsymbol{y}',\boldsymbol{y})} + \varphi(\boldsymbol{y}) - \varphi(\boldsymbol{y}')\right\}.$$

Moreover, we have that $f_{\boldsymbol{y}}(\,\cdot\,;\boldsymbol{y}')$ is actually differentiable everywhere as its gradient can be continuously extended to the junction affine hyperplane with constant value equal to $q(\boldsymbol{y},\boldsymbol{y}')(\boldsymbol{y}' - \boldsymbol{y})$. We now show that $f_{\boldsymbol{y}}(\,\cdot\,;\boldsymbol{y}')$ is $\frac{1}{t}q(\boldsymbol{y},\boldsymbol{y}') \cdot \|\boldsymbol{y}' - \boldsymbol{y}\|^2$-smooth. Indeed, it is defined as the composition of the linear form $\boldsymbol{\theta} \mapsto \langle \boldsymbol{\theta}, \boldsymbol{y}' - \boldsymbol{y}\rangle$ and the function $g : \mathbb{R} \to \mathbb{R}$ given by:

$$g : x \mapsto \begin{cases} t \cdot q(\boldsymbol{y}',\boldsymbol{y}) \exp\left(\left[x + \varphi(\boldsymbol{y}') - \varphi(\boldsymbol{y})\right]/t\right) & \text{if } x \leq t \log\frac{q(\boldsymbol{y},\boldsymbol{y}')}{q(\boldsymbol{y}',\boldsymbol{y})} + \varphi(\boldsymbol{y}) - \varphi(\boldsymbol{y}'), \\ t \cdot q(\boldsymbol{y},\boldsymbol{y}')\left(\left[x + \varphi(\boldsymbol{y}') - \varphi(\boldsymbol{y})\right]/t + 1 - \log\frac{q(\boldsymbol{y},\boldsymbol{y}')}{q(\boldsymbol{y}',\boldsymbol{y})}\right) & \text{if } x > t \log\frac{q(\boldsymbol{y},\boldsymbol{y}')}{q(\boldsymbol{y}',\boldsymbol{y})} + \varphi(\boldsymbol{y}) - \varphi(\boldsymbol{y}'). \end{cases}$$

We begin by showing that $g$ is $\frac{1}{t}q(\boldsymbol{y},\boldsymbol{y}')$-smooth. We have:

$$g' : x \mapsto \begin{cases} q(\boldsymbol{y}',\boldsymbol{y}) \exp\left(\left[x + \varphi(\boldsymbol{y}') - \varphi(\boldsymbol{y})\right]/t\right) & \text{if } x \leq t \log\frac{q(\boldsymbol{y},\boldsymbol{y}')}{q(\boldsymbol{y}',\boldsymbol{y})} + \varphi(\boldsymbol{y}) - \varphi(\boldsymbol{y}'), \\ q(\boldsymbol{y},\boldsymbol{y}') & \text{if } x > t \log\frac{q(\boldsymbol{y},\boldsymbol{y}')}{q(\boldsymbol{y}',\boldsymbol{y})} + \varphi(\boldsymbol{y}) - \varphi(\boldsymbol{y}'). \end{cases}$$

Thus, $g'$ is continuous, and differentiable everywhere except in $x_0 \coloneqq t \log\frac{q(\boldsymbol{y},\boldsymbol{y}')}{q(\boldsymbol{y}',\boldsymbol{y})} + \varphi(\boldsymbol{y}) - \varphi(\boldsymbol{y}')$. Its derivative is given by:

$$g'' : x \mapsto \begin{cases} \frac{1}{t}q(\boldsymbol{y}',\boldsymbol{y}) \exp\left(\left[x + \varphi(\boldsymbol{y}') - \varphi(\boldsymbol{y})\right]/t\right) & \text{if } x \leq t \log\frac{q(\boldsymbol{y},\boldsymbol{y}')}{q(\boldsymbol{y}',\boldsymbol{y})} + \varphi(\boldsymbol{y}) - \varphi(\boldsymbol{y}'), \\ 0 & \text{if } x > t \log\frac{q(\boldsymbol{y},\boldsymbol{y}')}{q(\boldsymbol{y}',\boldsymbol{y})} + \varphi(\boldsymbol{y}) - \varphi(\boldsymbol{y}'). \end{cases}$$

- For $x_1, x_2 \leq x_0$, we have:
$$\begin{aligned} |g'(x_1) - g'(x_2)| &\leq |x_1 - x_2| \sup_{x \in ]-\infty, x_0[} |g''(x)| \\ &= |x_1 - x_2| \lim_{\substack{x \to x_0 \\ x < x_0}} |g''(x)| \\ &= \frac{1}{t}q(\boldsymbol{y},\boldsymbol{y}') \cdot |x_1 - x_2|. \end{aligned}$$

- For $x_1, x_2 \geq x_0$, we trivially have $|g'(x_1) - g'(x_2)| = 0$.

- For $x_1 \leq x_0 \leq x_2$, we have:
$$\begin{aligned} |g'(x_1) - g'(x_2)| &= |(g'(x_1) - g'(x_0)) - (g'(x_2) - g'(x_0))| \\ &\leq |g'(x_1) - g'(x_0)| + |g'(x_2) - g'(x_0)| \\ &\leq \frac{1}{t}q(\boldsymbol{y},\boldsymbol{y}') \cdot |x_1 - x_0| \\ &\leq \frac{1}{t}q(\boldsymbol{y},\boldsymbol{y}') \cdot |x_1 - x_2|. \end{aligned}$$

Thus, we have:

$$\forall x_1, x_2 \in \mathbb{R}, |g'(x_1) - g'(x_2)| \leq \frac{1}{t} q(\boldsymbol{y}, \boldsymbol{y}') \cdot |x_1 - x_2|,$$

and $g$ is $\frac{1}{t} q(\boldsymbol{y}, \boldsymbol{y}')$-smooth. Nevertheless, we have $f_{\boldsymbol{y}}(\,\cdot\,, \boldsymbol{y}') = g(\langle\,\cdot\,, \boldsymbol{y}' - \boldsymbol{y}\rangle)$. Thus, we have, for $\boldsymbol{\theta}_1, \boldsymbol{\theta}_2 \in \mathbb{R}^d$:

$$
\begin{aligned}
||\nabla_{\boldsymbol{\theta}} f_{\boldsymbol{y}}(\boldsymbol{\theta_1}, \boldsymbol{y}') - \nabla_{\boldsymbol{\theta}} f_{\boldsymbol{y}}(\boldsymbol{\theta_2}, \boldsymbol{y}')|| &= ||g'(\langle\boldsymbol{\theta}_1, \boldsymbol{y}' - \boldsymbol{y}\rangle)(\boldsymbol{y}' - \boldsymbol{y}) - g'(\langle\boldsymbol{\theta}_2, \boldsymbol{y}' - \boldsymbol{y}\rangle)(\boldsymbol{y}' - \boldsymbol{y})|| \\
&= |g'(\langle\boldsymbol{\theta}_1, \boldsymbol{y}' - \boldsymbol{y}\rangle) - g'(\langle\boldsymbol{\theta}_2, \boldsymbol{y}' - \boldsymbol{y}\rangle)| \cdot ||\boldsymbol{y}' - \boldsymbol{y}|| \\
&\leq \frac{1}{t} q(\boldsymbol{y}, \boldsymbol{y}') \cdot |\langle\boldsymbol{\theta}_1, \boldsymbol{y}' - \boldsymbol{y}\rangle - \langle\boldsymbol{\theta}_2, \boldsymbol{y}' - \boldsymbol{y}\rangle| \cdot ||\boldsymbol{y}' - \boldsymbol{y}|| \\
&\leq \frac{1}{t} q(\boldsymbol{y}, \boldsymbol{y}') \cdot ||\boldsymbol{y}' - \boldsymbol{y}||^2 \cdot ||\boldsymbol{\theta}_1 - \boldsymbol{\theta}_2||,
\end{aligned}
$$

and $f_{\boldsymbol{y}}(\,\cdot\,, \boldsymbol{y}')$ is $\frac{1}{t} q(\boldsymbol{y}, \boldsymbol{y}') \cdot ||\boldsymbol{y}' - \boldsymbol{y}||^2$-smooth. Thus, recalling that $F_{\boldsymbol{y}}$ is defined as

$$F_{\boldsymbol{y}} : \boldsymbol{\theta} \mapsto \langle\boldsymbol{\theta}, \boldsymbol{y}\rangle + \sum_{\boldsymbol{y}' \in \mathcal{N}(\boldsymbol{y})} f_{\boldsymbol{y}}(\boldsymbol{\theta}; \boldsymbol{y}'),$$

we have that $F_{\boldsymbol{y}}$ is $\sum_{\boldsymbol{y}' \in \mathcal{N}(\boldsymbol{y})} \frac{1}{t} q(\boldsymbol{y}, \boldsymbol{y}') \cdot ||\boldsymbol{y}' - \boldsymbol{y}||^2 = \mathbb{E}_{q(\boldsymbol{y}, \cdot)} ||Y - \boldsymbol{y}||_2^2 / t$-smooth. Finally, as $\Omega_{\boldsymbol{y}} := (F_{\boldsymbol{y}})^*$, Fenchel duality theory gives that $\Omega_{\boldsymbol{y}}$ is $t/\mathbb{E}_{q(\boldsymbol{y}, \cdot)} ||Y - \boldsymbol{y}||_2^2$-strongly convex.

- $\mathbb{E}_{\boldsymbol{p}_{\boldsymbol{\theta}, \boldsymbol{y}}^{(1)}}[Y] = \mathsf{argmax}_{\boldsymbol{\mu} \in \mathsf{conv}(\mathcal{N}(\boldsymbol{y}) \cup \{\boldsymbol{y}\})} \{\langle\boldsymbol{\theta}, \boldsymbol{\mu}\rangle - \Omega_{\boldsymbol{y}}(\boldsymbol{\mu})\}$:

Noticing that $g$ is continuous on $\mathbb{R}$, convex on $\left]-\infty, t \log \frac{q(\boldsymbol{y}, \boldsymbol{y}')}{q(\boldsymbol{y}', \boldsymbol{y})} + \varphi(\boldsymbol{y}) - \varphi(\boldsymbol{y}')\right[$ and on $\left]t \log \frac{q(\boldsymbol{y}, \boldsymbol{y}')}{q(\boldsymbol{y}', \boldsymbol{y})} + \varphi(\boldsymbol{y}) - \varphi(\boldsymbol{y}'), +\infty\right[$, and with matching derivatives on the junction:

$$g'(t) \xrightarrow[t < t \log \frac{q(\boldsymbol{y}, \boldsymbol{y}')}{q(\boldsymbol{y}', \boldsymbol{y})} + \varphi(\boldsymbol{y}) - \varphi(\boldsymbol{y}')]{t \to t \log \frac{q(\boldsymbol{y}, \boldsymbol{y}')}{q(\boldsymbol{y}', \boldsymbol{y})} + \varphi(\boldsymbol{y}) - \varphi(\boldsymbol{y}')} q(\boldsymbol{y}, \boldsymbol{y}'), \quad g'(t) \xrightarrow[t > t \log \frac{q(\boldsymbol{y}, \boldsymbol{y}')}{q(\boldsymbol{y}', \boldsymbol{y})} + \varphi(\boldsymbol{y}) - \varphi(\boldsymbol{y}')]{t \to t \log \frac{q(\boldsymbol{y}, \boldsymbol{y}')}{q(\boldsymbol{y}', \boldsymbol{y})} + \varphi(\boldsymbol{y}) - \varphi(\boldsymbol{y}')} q(\boldsymbol{y}, \boldsymbol{y}'),$$

gives that $g$ is convex on $\mathbb{R}$. Thus, $f_{\boldsymbol{y}}(\,\cdot\,; \boldsymbol{y}')$ is convex on $\mathbb{R}^d$ by composition. Thus,

$$F_{\boldsymbol{y}} : \boldsymbol{\theta} \mapsto \langle\boldsymbol{\theta}, \boldsymbol{y}\rangle + \sum_{\boldsymbol{y}' \in \mathcal{N}(\boldsymbol{y})} f_{\boldsymbol{y}}(\boldsymbol{\theta}; \boldsymbol{y}')$$

is closed proper convex as the sum of such functions. The Fenchel-Moreau theorem then gives that it is equal to its biconjugate. Thus, we have:

$$F_{\boldsymbol{y}}(\boldsymbol{\theta}) = \sup_{\boldsymbol{\mu} \in \mathbb{R}^d} \{\langle\boldsymbol{\theta}, \boldsymbol{\mu}\rangle - (F_{\boldsymbol{y}})^*(\boldsymbol{\mu})\} = \sup_{\boldsymbol{\mu} \in \mathbb{R}^d} \{\langle\boldsymbol{\theta}, \boldsymbol{\mu}\rangle - \Omega_{\boldsymbol{y}}(\boldsymbol{\mu})\}.$$

Nonetheless, the gradient of $F_{\boldsymbol{y}}$ is given by:

$$
\begin{aligned}
\nabla_{\boldsymbol{\theta}} F_{\boldsymbol{y}}(\boldsymbol{\theta}) &= \boldsymbol{y} + \sum_{\boldsymbol{y}' \in \mathcal{N}_{\boldsymbol{y}}^-(\boldsymbol{\theta})} q(\boldsymbol{y}', \boldsymbol{y}) \exp\left(\left[\langle\boldsymbol{\theta}, \boldsymbol{y}' - \boldsymbol{y}\rangle + \varphi(\boldsymbol{y}') - \varphi(\boldsymbol{y})\right] / t\right) \cdot (\boldsymbol{y}' - \boldsymbol{y}) + \sum_{\boldsymbol{y}' \in \mathcal{N}_{\boldsymbol{y}}^+(\boldsymbol{\theta})} q(\boldsymbol{y}, \boldsymbol{y}') \cdot (\boldsymbol{y}' - \boldsymbol{y}) \\
&= \mathbb{E}_{\boldsymbol{p}_{\boldsymbol{\theta}, \boldsymbol{y}}^{(1)}}[Y].
\end{aligned}
$$

Thus, we have $\nabla F_{\boldsymbol{y}}(\mathbb{R}^d) \subset \mathsf{conv}\left(\mathcal{N}(\boldsymbol{y}) \cup \{\boldsymbol{y}\}\right)$, which gives:

$$\forall \boldsymbol{\theta} \in \mathbb{R}^d, ||\nabla F_{\boldsymbol{y}}(\boldsymbol{\theta})|| \leq R_{\mathcal{N}(\boldsymbol{y})} := \max_{\boldsymbol{\mu} \in \mathsf{conv}(\mathcal{N}(\boldsymbol{y}) \cup \{\boldsymbol{y}\})} ||\boldsymbol{\mu}||,$$

so that we have $\mathsf{dom}(\Omega_{\boldsymbol{y}}) \subset B(\mathbf{0}, R_{\mathcal{N}(\boldsymbol{y})})$. Thus we can actually write:

$$F_{\boldsymbol{y}}(\boldsymbol{\theta}) = \sup_{\boldsymbol{\mu} \in B(\mathbf{0}, R_{\mathcal{N}(\boldsymbol{y})})} \{\langle\boldsymbol{\theta}, \boldsymbol{\mu}\rangle - \Omega_{\boldsymbol{y}}(\boldsymbol{\mu})\},$$

and now apply Danskin's theorem as $B(\mathbf{0}, R_{\mathcal{N}(\boldsymbol{y})})$ is compact, which further gives:

$$\partial F_{\boldsymbol{y}}(\boldsymbol{\theta}) = \underset{\boldsymbol{\mu} \in B(\mathbf{0}, R_{\mathcal{N}(\boldsymbol{y})})}{\arg\max} \left\{ \langle \boldsymbol{\theta}, \boldsymbol{\mu} \rangle - \Omega_{\boldsymbol{y}}(\boldsymbol{\mu}) \right\},$$

and the fact that $F_{\boldsymbol{y}}$ is differentiable gives that both sides are single-valued. Moreover, as $\nabla F_{\boldsymbol{y}}(\mathbb{R}^d) \subset$ conv $(\mathcal{N}(\boldsymbol{y}) \cup \{\boldsymbol{y}\})$, we know that the right hand side is maximized in conv $(\mathcal{N}(\boldsymbol{y}) \cup \{\boldsymbol{y}\})$, and we can actually write:

$$\mathbb{E}_{\boldsymbol{p}_{\boldsymbol{\theta}, \boldsymbol{y}}^{(1)}}[Y] = \nabla F_{\boldsymbol{y}}(\boldsymbol{\theta}) = \underset{\boldsymbol{\mu} \in \mathsf{conv}(\mathcal{N}(\boldsymbol{y}) \cup \{\boldsymbol{y}\})}{\arg\max} \left\{ \langle \boldsymbol{\theta}, \boldsymbol{\mu} \rangle - \Omega_{\boldsymbol{y}}(\boldsymbol{\mu}) \right\}.$$

- Smoothness of $L_{\Omega_{\boldsymbol{y}}}(\,\cdot\,; \boldsymbol{y})$ and expression of its gradient:

Based on the above, we have:

$$L_{\Omega_{\boldsymbol{y}}}(\boldsymbol{\theta}\,;\boldsymbol{y}) = F_{\boldsymbol{y}}(\boldsymbol{\theta}) + \Omega_{\boldsymbol{y}}(\boldsymbol{y}) - \langle \boldsymbol{\theta}, \boldsymbol{y} \rangle.$$

Thus, the $\mathbb{E}_{q(\boldsymbol{y}, \cdot)} \|Y - \boldsymbol{y}\|_2^2 / t$-smoothness of $L_{\Omega_{\boldsymbol{y}}}(\,\cdot\,; \boldsymbol{y})$ follows directly from the previously established $\mathbb{E}_{q(\boldsymbol{y}, \cdot)} \|Y - \boldsymbol{y}\|_2^2 / t$-smoothness of $F_{\boldsymbol{y}}$. Similarly, the expression of $\nabla_{\boldsymbol{\theta}} L_{\Omega_{\boldsymbol{y}}}(\boldsymbol{\theta}\,; \boldsymbol{y})$ follows from the previously established expression of $\nabla_{\boldsymbol{\theta}} F_{\boldsymbol{y}}(\boldsymbol{\theta})$, and we have:

$$\nabla_{\boldsymbol{\theta}} L_{\Omega_{\boldsymbol{y}}}(\boldsymbol{\theta}\,;\boldsymbol{y}) = \nabla_{\boldsymbol{\theta}} F_{\boldsymbol{y}}(\boldsymbol{\theta}) - \boldsymbol{y} = \mathbb{E}_{\boldsymbol{p}_{\boldsymbol{\theta}, \boldsymbol{y}}^{(1)}}[Y] - \boldsymbol{y}.$$

$\square$

*Proposition 6.* In the unconditional setting, given a dataset $(\boldsymbol{y}_i)_{i=1}^N$, the distribution of the first iterate of the Markov chain with transition kernel defined in Eq. (3) and initialized by $\boldsymbol{y}^{(0)} = \boldsymbol{y}_i$, with $i \sim \mathcal{U}(\llbracket 1, N \rrbracket)$, is given by:

$$(\boldsymbol{p}_{\boldsymbol{\theta}, \bar{Y}_N}^{(1)})(\boldsymbol{y}) = \sum_{\boldsymbol{y}' \in \mathcal{Y}} \left( \sum_{i=1}^N \mathbf{1}_{\{\boldsymbol{y}_i = \boldsymbol{y}'\}} \cdot \frac{1}{N} \right) P_{\boldsymbol{\theta}, t}(\boldsymbol{y}', \boldsymbol{y})$$

$$= \sum_{\boldsymbol{y}' \in \mathcal{Y}} \left( \sum_{i=1}^N \mathbf{1}_{\{\boldsymbol{y}_i = \boldsymbol{y}'\}} \cdot \frac{1}{N} \right) \boldsymbol{p}_{\boldsymbol{\theta}, \boldsymbol{y}'}^{(1)}(\boldsymbol{y})$$

$$= \frac{1}{N} \sum_{i=1}^N \boldsymbol{p}_{\boldsymbol{\theta}, \boldsymbol{y}_i}^{(1)}(\boldsymbol{y}).$$

Thus, keeping the same notations as in the previous proof, previous calculations give:

$$\mathbb{E}_{\boldsymbol{p}_{\boldsymbol{\theta}, \bar{Y}_N}^{(1)}}[Y] = \frac{1}{N} \sum_{i=1}^N \mathbb{E}_{\boldsymbol{p}_{\boldsymbol{\theta}, \boldsymbol{y}_i}^{(1)}}[Y]$$

$$= \frac{1}{N} \sum_{i=1}^N \nabla_{\boldsymbol{\theta}} F_{\boldsymbol{y}_i}(\boldsymbol{\theta})$$

$$= \nabla_{\boldsymbol{\theta}} \left( \frac{1}{N} \sum_{i=1}^N F_{\boldsymbol{y}_i} \right)(\boldsymbol{\theta}).$$

Let $F_{\bar{Y}_N} := \frac{1}{N} \sum_{i=1}^N F_{\boldsymbol{y}_i}$ Then, the exact same arguments as in the conditional case hold, and the results of Proposition 6 are obtained by replacing $F_{\boldsymbol{y}}$ by $F_{\bar{Y}_N}$ in the proof of Proposition 3, and noticing that the previously shown $\mathbb{E}_{q(\boldsymbol{y}_i, \cdot)} \|Y - \boldsymbol{y}_i\|_2^2 / t$-smoothness of $F_{\boldsymbol{y}_i}$ gives that $F_{\bar{Y}_N}$ is $\frac{1}{N} \sum_{i=1}^N \mathbb{E}_{q(\boldsymbol{y}_i, \cdot)} \|Y - \boldsymbol{y}_i\|_2^2 / t$-smooth. Similar arguments also hold for the regularized optimization formulation, by noting that this time we have $\nabla F_{\bar{Y}_N}(\mathbb{R}^d) \subset \mathsf{conv}\left( \bigcup_{i=1}^N \{\mathcal{N}(\boldsymbol{y}_i) \cup \{\boldsymbol{y}_i\}\} \right)$. $\square$

### E.8 PROOF OF PROPOSITION 7

*Proof.* The first point is directly given by the fact that $\mathbb{E}_{\boldsymbol{p}_{\boldsymbol{\theta},\boldsymbol{y}}^{(1)}}[Y]$ is the expectation of a distribution over $\mathcal{N}(\boldsymbol{y}) \cup \{\boldsymbol{y}\}$. For the second and third points, as derived in Section E.7, we have:

$$\mathbb{E}_{\boldsymbol{p}_{\boldsymbol{\theta},\boldsymbol{y}}^{(1)}}[Y] = \boldsymbol{y} + \sum_{\boldsymbol{y}' \in \mathcal{N}_{\boldsymbol{y}}^{-}(\boldsymbol{\theta})} q(\boldsymbol{y}',\boldsymbol{y}) \exp\left(\left[\langle \boldsymbol{\theta}, \boldsymbol{y}' - \boldsymbol{y}\rangle + \varphi(\boldsymbol{y}') - \varphi(\boldsymbol{y})\right]/t\right) \cdot (\boldsymbol{y}' - \boldsymbol{y}) + \sum_{\boldsymbol{y}' \in \mathcal{N}_{\boldsymbol{y}}^{+}(\boldsymbol{\theta})} q(\boldsymbol{y},\boldsymbol{y}') \cdot (\boldsymbol{y}' - \boldsymbol{y}).$$

Define then:

$$\mathcal{N}_{\text{better}}(\boldsymbol{y}) := \{\boldsymbol{y}' \in \mathcal{N}(\boldsymbol{y}) \mid \langle \boldsymbol{\theta}, \boldsymbol{y}'\rangle + \varphi(\boldsymbol{y}') > \langle \boldsymbol{\theta}, \boldsymbol{y}\rangle + \varphi(\boldsymbol{y})\},$$
$$\mathcal{N}_{\text{worse}}(\boldsymbol{y}) := \{\boldsymbol{y}' \in \mathcal{N}(\boldsymbol{y}) \mid \langle \boldsymbol{\theta}, \boldsymbol{y}'\rangle + \varphi(\boldsymbol{y}') < \langle \boldsymbol{\theta}, \boldsymbol{y}\rangle + \varphi(\boldsymbol{y})\}$$

as the sets of improving and worsening neighbors of $\boldsymbol{y}$ respectively (assuming no neighbor of $\boldsymbol{y}$ has exactly equal objective value for simplicity, which is true almost everywhere w.r.t. $\boldsymbol{\theta} \in \mathbb{R}^d$).

**Low temperature limit.** We have:

$$\mathcal{N}_{\boldsymbol{y}}^{+}(\boldsymbol{\theta}) \xrightarrow[t \to 0^+]{} \mathcal{N}_{\text{better}}(\boldsymbol{y}), \quad \text{and} \quad \mathcal{N}_{\boldsymbol{y}}^{-}(\boldsymbol{\theta}) \xrightarrow[t \to 0^+]{} \mathcal{N}_{\text{worse}}(\boldsymbol{y}).$$

Then, as $x < 0 \implies \exp(x/t) \xrightarrow[t \to 0^+]{} 0$, we have effectively

$$\mathbb{E}_{\boldsymbol{p}_{\boldsymbol{\theta},\boldsymbol{y}}^{(1)}}[Y] \xrightarrow[t \to 0^+]{} \boldsymbol{y} + \sum_{\boldsymbol{y}' \in \mathcal{N}_{\text{better}}(\boldsymbol{y})} q(\boldsymbol{y},\boldsymbol{y}') \cdot (\boldsymbol{y}' - \boldsymbol{y}).$$

**High temperature limit.** As $\forall x \in \mathbb{R}, \exp(x/t) \xrightarrow[t \to \infty]{} 1$, we have:

$$\mathcal{N}_{\boldsymbol{y}}^{+}(\boldsymbol{\theta}) \xrightarrow[t \to \infty]{} \{\boldsymbol{y}' \in \mathcal{N}(\boldsymbol{y}) \mid q(\boldsymbol{y}',\boldsymbol{y}) > q(\boldsymbol{y},\boldsymbol{y}')\}, \quad \text{and} \quad \mathcal{N}_{\boldsymbol{y}}^{-}(\boldsymbol{\theta}) \xrightarrow[t \to \infty]{} \{\boldsymbol{y}' \in \mathcal{N}(\boldsymbol{y}) \mid q(\boldsymbol{y}',\boldsymbol{y}) \leq (\boldsymbol{y},\boldsymbol{y}')\}.$$

Thus, we have:

$$\mathbb{E}_{\boldsymbol{p}_{\boldsymbol{\theta},\boldsymbol{y}}^{(1)}}[Y] \xrightarrow[t \to \infty]{} \boldsymbol{y} + \sum_{\boldsymbol{y}' \mid q(\boldsymbol{y}',\boldsymbol{y}) \leq (\boldsymbol{y},\boldsymbol{y}')} q(\boldsymbol{y}',\boldsymbol{y}) \cdot (\boldsymbol{y}' - \boldsymbol{y}) + \sum_{\boldsymbol{y}' \mid q(\boldsymbol{y}',\boldsymbol{y}) > (\boldsymbol{y},\boldsymbol{y}')} q(\boldsymbol{y},\boldsymbol{y}') \cdot (\boldsymbol{y}' - \boldsymbol{y}),$$

which gives effectively:

$$\mathbb{E}_{\boldsymbol{p}_{\boldsymbol{\theta},\boldsymbol{y}}^{(1)}}[Y] \xrightarrow[t \to \infty]{} \boldsymbol{y} + \sum_{\boldsymbol{y}' \in \mathcal{N}(\boldsymbol{y})} \min\left[q(\boldsymbol{y},\boldsymbol{y}'), q(\boldsymbol{y}',\boldsymbol{y})\right] \cdot (\boldsymbol{y}' - \boldsymbol{y}).$$

$\square$

