# OpenReview forum: "Learning with Local Search MCMC Layers"
_ICLR.cc/2026/Conference — Submitted to ICLR 2026_

### Official Review · Reviewer_vhAR · 2025-10-29

**Soundness:** 4
**Presentation:** 3
**Contribution:** 4
**Rating:** 8
**Confidence:** 2

**Summary:**

This paper leverages the link between local search heuristics and MCMC methods to integrate local search heuristics as differentiable, stochastic layers in neural networks. Fenchel Young loss functions are used as practical stochastic gradient algorithms for both conditional and unconditional settings, and principled convergence guarantees are provided for the algorithm. Empirical results on the dynamic VRP with time windows and predicting binary vectors further validate the effectiveness of the proposed algorithm.

**Strengths:**

1. This paper is based on the very neat idea of linking local search heuristics and MCMC methods to allow local search heuristics as differentiable layers.

2. Principled theoretical analysis are provided, together with convincing empirical results, making the paper very solid.

**Weaknesses:**

While the dynamic VRPTW experiment (5.1) seems solid, the predicting binary vector experiment (5.2) seems quite synthetic. I wonder if the method proposed in this paper can perform competitively for other combinatorial optimization as well (e.g. scheduling, graph coloring, max cut etc).

**Questions:**

1. The authors mention in the future work they plan to extend their framework to large neighborhood search algorithms. I’m quite curious about how they may be able to do it & if the current proposed framework is flexible enough to extend to other iterative combinatorial optimization algorithms (e.g. genetic search, tabu search etc).

2. Can the authors discuss more about the limitation of Alg. 2 (computing the single individual ratio instead of summing the forward probabilities for all move types and the reverse probabilities for all move types)? In what realistic situation could this fails?

---

> ### Author Response · Authors · 2025-11-21
>
> We thank the reviewer for their excellent assessment and for highlighting the "very neat idea," "principled theoretical analysis," and "convincing empirical results." We are glad the reviewer found the paper solid. We address the specific questions and weaknesses below.
>
> ### __Response to Weaknesses (generalization to other problems)__
>
> We appreciate the opportunity to clarify the scope of the method and provide further empirical evidence of its generality.
>
> 1. __New Experiment__. To demonstrate applicability to other standard combinatorial problems, we have added a new experiment on the Multi-Dimensional Knapsack Problem (MKP) (see Section 5.2 and appendix D in the revised submission).
> - __Task:__ Selecting a subset of items to maximize predicted values under multiple resource constraints. This is a fundamental OR problem distinct from routing.
> - __Baselines:__ We compared our method against a broad landscape of differentiable optimization baselines from the PyEPO library: SPO+, PFY, NID, and NCE.
> Results: Our LS-MCMC layer outperforms the baselines in terms of test relative regret, while drastically reducing the computational burden.
> 2. __Purpose of the binary vector experiment.__ We emphasize that the experiment on predicting binary vectors (now Section 5.3) was designed as a controlled environment where the exact partition function and gradients are tractable (which is impossible for most combinatorial problems). This allowed us to theoretically validate that our gradient estimators converge to the true exact gradients. We also provided an experiment on predicting k-subsets (in appendix A), which is a similar controlled environment with additional constraints (binary vectors with $k$ “ones”), where the partition function and gradients can be computed via smoothed dynamic programming.
> 3. __Generalization.__ The proposed method applies seamlessly to any combinatorial optimization problem where a local search oracle (i.e., a natural neighborhood structure) is available. This includes the examples mentioned by the reviewer (scheduling, graph coloring, MaxCut) which all possess standard neighborhood structures (e.g., swap moves, color changes, vertex flips) that are useful for optimization. Our method is particularly relevant in settings where the scale of the problem precludes the use of exact solvers (standard LP/MIP solvers), leaving heuristics such as local search-based algorithms as the only viable option.

---

> > ### Author Response · Authors · 2025-11-21
> >
> > ### __Response to Question 1 (extensions of the framework)__
> >
> > This is a very exciting direction for future research.
> > 1. __Large Neighborhood Search (LNS).__ To extend our framework to LNS-style algorithms, we believe one must move beyond the "accept/reject" framework of Metropolis-Hastings, as it is unable to efficiently leverage local (“large neighborhood-wise”) optimization oracles. Indeed, when a proposal solution $y'$ is obtained from optimizing in the neighborhood of the current solution $y^{(k)}$, the reverse move (getting to $y^{(k)}$ from $y'$ via a similar local optimization step) is in general very unlikely, leading to very inefficient acceptance rates for Metropolis-Hastings.
> > 2. __Other iterative algorithms.__ We believe that extending the framework to other iterative algorithms requires re-thinking the state space of the Markov chain:
> > - Genetic algorithms: The state space can be viewed as the distribution simplex on feasible solutions ($\Delta^{|\mathcal{Y}|}$), rather than the set of feasible solutions $\mathcal{Y}$ itself. Indeed, the algorithm maintains a population of solutions, which can be seen as an empirical distribution on $\mathcal{Y}$. As the population size grows to infinity, the analysis approaches a mean-field limit: existing Markov chain-based analyses of genetic algorithms support this perspective.
> > - Tabu search: For algorithms with memory (like Tabu lists of size $T$), the state space can be modeled as the Cartesian product $\mathcal{Y}^T$. This frames the $T$-step memory process as a standard Markov chain on the history of solutions. The neighborhood of a sequence $(y_t, \dots, y_{t+T-1})$ would then be defined as the set of sequences $(y_{t+1}, \dots, y')$ where $y' \in \mathcal{N}(y_{t+T-1})$.
> > - Iterated local search: For iterated local search-type algorithms, where an iteration corresponds to applying a random perturbation and performing a full (deterministic / hill-climbing) local search, the state space of the Markov chain can be thought of as the set of locally optimal solutions, which is a neighborhood system-dependent, strict subset of $\mathcal{Y}$.
> >
> > ### __Response to Question 2__
> >
> > We thank the reviewer for raising this point, which suggests that our discussion in Section 3.2 regarding the "summing of probabilities" could be interpreted as a limitation of our proposed Algorithm 2.
> >
> > We intended that discussion to highlight the computational intractability of a naive mixture approach (based on Algorithm 1), which would indeed require summing probabilities over all move types. The purpose of Algorithm 2 is precisely to __overcome__ this limitation. To clarify:
> > - The naive approach (based on Algorithm 1): Requires computing the transition probability by summing over all possible neighborhood moves that could lead to the same proposed solution, for both forward and backward moves. This is combinatorially expensive.
> > - Algorithm 2: Solves this by sampling the neighborhood type first, then computing a correction ratio only for that specific move type only.
> >
> > Algorithm 2 is designed to be computationally efficient while remaining theoretically sound. It is, in a way, a generalization of the Metropolis-Hastings algorithm designed to efficiently leverage multiple proposal distributions in a discrete MCMC setting.
> > As shown in Proposition 2, it converges to the unique, correct stationary distribution $\pi_{\theta,t}$ without relying on any approximation. Therefore, there are no realistic situations where it fails to target the correct distribution (provided the aggregate neighborhood graph is connected).
> >
> > We realize this distinction was not sharp enough in the text: we revised Section 3.2 to make it clearer.

---

### Official Review · Reviewer_5HxJ · 2025-10-29

**Soundness:** 2
**Presentation:** 2
**Contribution:** 2
**Rating:** 4
**Confidence:** 3

**Summary:**

This paper aims to integrate neural network training directly into combinatorial optimization processes. Specifically, it introduces a method that enhances local search by incorporating a differentiable layer based on MCMC, and presents the corresponding theoretical background and training procedure. Experiments are conducted on two tasks—VRPTW and binary vector prediction—and in the case of VRPTW, the proposed method outperforms HGS within limited computational budgets.

**Strengths:**

- The paper proposes a novel approach to improving local search performance using a differentiable MCMC layer, which is technically interesting.

- It demonstrates the ability to solve problems with complex constraints, such as VRPTW, within a short amount of time.

**Weaknesses:**

- Similar to traditional heuristic-based local search methods, the performance of the proposed approach heavily depends on how well the local search component is designed. In this sense, the paper does not improve prior NCO methods to better handle complex constraints; rather, it proposes a new type of meta-heuristic that utilizes a differentiable MCMC layer. Considering the ongoing trend of incorporating neural networks more actively in combinatorial optimization, the long-term impact of this work may be somewhat limited.

- Experiments are restricted to VRPTW and binary vector prediction, and for VRPTW, there are no comparison baselines beyond HGS. Incorporating commonly evaluated tasks (e.g. CVRP) and comparing against prior NCO or heuristic methods would enable a more objective assessment of the proposed approach.

- While the method outperforms HGS under strict time limits for VRPTW, its performance becomes inferior when more runtime is allowed. This suggests that although it can quickly generate reasonably good solutions, it may struggle to further refine solutions near optimality or escape local optima compared to traditional meta-heuristic approaches.

- The overall presentation and method complexity may be challenging for NCO researchers to follow, potentially limiting the number of researchers who may be interested in or willing to adopt the approach.

**Questions:**

None

---

> ### Author Response · Authors · 2025-11-21
>
> We thank the reviewer for their time and for acknowledging the novelty of our approach in leveraging MCMC for local search, as well as our ability to handle complex constraints like VRPTW.
>
> We believe there are crucial misunderstandings regarding the positioning of our paper (DFL vs. NCO), the generality of our framework, and the scope of our experiments. We clarify these points below, including results from a new experiment on the Multi-Dimensional Knapsack Problem.
>
> ### __1) Positioning: Distinction from NCO and "Limited Impact"__
>
> The reviewer suggests our method "does not improve prior NCO methods". We respectfully clarify that our work falls under the paradigm of __Decision-Focused Learning (DFL)__ (also called predict-and-optimize), rather than __Neural Combinatorial Optimization (NCO)__.
>
> - In NCO, the goal is to use ML to replace a solver. The model learns the heuristic logic itself to output a solution directly.
> - In the DFL paradigm, the goal is not to replace the solver. The model predicts the parameters (e.g., costs, prizes $\theta$) of the problem based on inputs $x$. These parameters are then fed into a fixed CO algorithm. The pipeline is: ML model $\to$ parameters $\theta$ $\to$ CO layer $\to$ solution $y$.
>
> Our contribution consists in "opening the black box" of the CO algorithm, not to learn the heuristic’s logic (as in NCO), but to enable principled, gradient-based learning of the upstream ML model weights $W$.
>
> __Impact:__ Existing DFL methods lose theoretical guarantees when the solver is inexact. Our contribution is a mathematically rigorous gradient estimator for the vast class of NP-hard problems where neighborhood structures are natural and Local Search heuristics are available. This bridges a major gap in DFL, allowing practitioners to plug existing local search moves into a differentiable layer.
>
> ### __2) Performance vs. Runtime (Comparison with HGS)__
>
> The reviewer notes that while we outperform HGS under strict time limits, we are inferior at longer runtimes, suggesting a struggle to refine solutions. This observation is correct regarding solution quality, but we argue this is a __necessary feature for scalable training__, not a limitation of the method's utility.
>
> In DFL, the optimization layer must be solved for every data sample in every training epoch. This creates a natural bottleneck: consequently, we cannot afford to run a heavy meta-heuristic (like the full HGS) for seconds during the forward pass, as this would make training computationally infeasible. Our contribution provides a generic framework that transforms any problem-specific neighborhood system into a proposal distribution for MCMC.
>
> - __Training (Fast Regime):__ We use this framework to construct a lightweight, stochastic layer that generates meaningful gradients rapidly. As shown in Table 3, our method significantly outperforms the baseline (Perturbed Inexact Oracle) in the 1ms - 100ms regime. This speed is critical for scaling DFL.
> - __Inference (High-Quality Regime):__ Once the model $g_W$ is trained, one can use any solver (including the full HGS) to obtain the final solution, as we do in our experiments ($f_W := \tilde{y} \circ g_W$).
>
> Thus, the "inferior" performance at long runtimes reflects the intentional trade-off of a layer designed for efficient training, while the trained model remains compatible with powerful solvers at inference time.
>
> ### __3) Experiments and Baselines__
>
> The reviewer states that experiments are restricted to VRPTW and binary vectors. Although the initial submission also provides experiments on a k-subsets prediction task (see appendix A), we have added a new experiment on the __Multi-Dimensional Knapsack Problem__, a canonical resource allocation problem distinct from routing. We compare against four DFL baselines from the PyEPO library: SPO+, PFY, NID, and NCE (see details in section 5.2 and appendix D of the revised submission).
>
> __Results__: Our LS-MCMC approach outperforms the baselines in terms of test relative regret, while reducing the computational burden.

---

> ### Comment · Reviewer_5HxJ · 2025-11-27
>
> Additional experiments on the Multidimensional Knapsack Problem (MDKP) have been included, which I believe significantly strengthen the baseline comparison that previously felt insufficient. The rebuttal also convincingly addressed my concerns regarding the contribution of this paper. Accordingly, I will raise my score to 6.

---

### Official Review · Reviewer_eZPo · 2025-11-01

**Soundness:** 3
**Presentation:** 2
**Contribution:** 2
**Rating:** 4
**Confidence:** 3

**Summary:**

This paper proposes a theoretically principled framework that integrates local search heuristics as differentiable layers in neural networks.
By interpreting neighborhood systems used in combinatorial heuristics as proposal distributions, the authors construct MCMC-based layers that remain differentiable even when relying on inexact solvers.
Theoretical analysis establishes connections to Fenchel–Young losses, convergence guarantees under stochastic gradient estimation, and asymptotic properties.
Experiments on the Dynamic Vehicle Routing Problem with Time Windows (DVRPTW) and on a synthetic binary vector prediction task demonstrate the feasibility of the method.

**Strengths:**

1. Well-motivated problem formulation: the paper clearly identifies the key limitation of existing differentiable combinatorial optimization approaches and formulates the challenge of learning with inexact local-search-based oracles in a principled manner.
2. Theoretical guarantees: the proposed method is supported by rigorous theoretical guarantees, including convergence analyses and connections to Fenchel–Young losses. This provides confidence in the soundness and reliability of the approach, even when the optimization oracle is approximate.

**Weaknesses:**

1. Limited coverage of application tasks: although the proposed approach is claimed to be general, experiments are restricted to DVRPTW and a toy binary vector prediction task. As a result, the empirical validation of generality remains narrow.
2. Sparse comparison with existing methods: Table 1 summarizes a broad landscape of differentiable combinatorial optimization methods, yet the experiments compare only against perturbed optimizers (Berthet et al., 2020). While the authors may argue that other approaches rely on exact oracles, this justification should be stated explicitly, and experimental evidence that those methods fail under inexact solvers would strengthen the argument.
3. Limited interpretability of short-time performance: Table 3 shows that the proposed method excels under very tight time budgets (1–100 ms). However, the paper does not clearly explain how this property translates to broader usefulness—e.g., whether such gains are relevant in other tasks or larger-scale industrial settings.

**Questions:**

1. Could the authors include other additional benchmark results (e.g., TSP, scheduling, or knapsack) to test generality of the proposed method?
2. Could the authors include other baselines (in Table 1) to test generality of the proposed method?
3. It would be valuable to discuss whether the observed short-time efficiency carries over to other tasks with combinatorial optimization layer.

---

> ### Author Response · Authors · 2025-11-21
>
> We thank the reviewer for their thoughtful feedback and for recognizing the "well-motivated problem formulation," "rigorous theoretical guarantees," and the "soundness and reliability" of our approach. We agree that demonstrating generality and providing broader comparisons is important. To address this, we have conducted a new experiment on the __Multi-Dimensional Knapsack Problem__ (MKP).
>
> ## __Response to Weaknesses__
>
> ### __Limited coverage of application tasks / Narrow empirical validation__
>
> To address the concern that our experiments were restricted to DVRPTW and a toy task, we have added a full benchmark on the Multi-Dimensional Knapsack Problem (MKP) (see section 5.2 and appendix D in the revision).
> - The MKP is a fundamental NP-hard problem in operations research involving resource allocation under multiple linear constraints. It is structurally distinct from routing (DVRPTW), binary vector prediction, and k-subset prediction (in the appendix).
> - __Results:__ The new experiment demonstrates that our LS-MCMC framework generalizes effectively to constraint-based problems, achieving competitive regret scores while reducing the computational burden.
>
> ### __Sparse comparison with existing methods__
>
> The reviewer noted that we initially compared only against perturbed optimizers. In the new MKP experiment, we explicitly benchmark against a comprehensive suite of methods from the PyEPO library: PFY, SPO+, NID, and NCE (see details in section 5.2 and appendix D of the revised submission).
>
> ### __Broader usefulness of short-time performance__
>
> The reviewer asks how short-time efficiency translates to broader usefulness. We argue that short-time efficiency is critical for the scalability of Decision-Focused Learning (DFL). Indeed, in DFL, the optimization problem sits inside the training loop. It must be solved for every data sample in a batch, for every batch, and every epoch. If the solver takes even 1 second per instance, training on a dataset of 10,000 samples would take hours per epoch just for solver time. Thus, the "inferior" performance at long runtimes reflects the intentional trade-off of a layer designed for efficient training, while the trained model remains compatible with powerful solvers at inference time.
>
> ## __Response to Questions__
>
> - Q1: __Could the authors include other additional benchmark results?__ Yes, we have added the Multi-Dimensional Knapsack Problem (MKP) as a new benchmark. This problem involves packing items into a knapsack with multiple capacity dimensions, testing the framework's ability to handle strict feasibility constraints different from routing logic.
> - Q2: __Could the authors include other baselines?__ Yes, in the MKP experiment, we now compare against SPO+, PFY, NID, and NCE. Sadly, we did not have time to add other baselines for the DVRPTW: indeed, the scale of this experiment in terms of code complexity (complex environment, Python / C++ interface for the solver…) makes the introduction of other baselines in the limited rebuttal period time difficult.
> - Q3: __Short-time efficiency in other tasks?__ The efficiency gains observed in DVRPTW transfer directly to other tasks because they stem from the fundamental design of the layer: replacing a combinatorial search for the optimum (which has unknown, potentially exponential complexity) with a fixed budget of MCMC iterations (linear complexity).

---

### Official Review · Reviewer_SQPE · 2025-11-01

**Soundness:** 3
**Presentation:** 3
**Contribution:** 2
**Rating:** 4
**Confidence:** 3

**Summary:**

The paper introduces Local Search Markov Chain Monte Carlo (MCMC) Layers, a theoretically grounded framework for integrating inexact combinatorial optimisation solvers within neural networks as differentiable, stochastic components. It reinterprets local search as a form of MCMC sampling, transforming neighborhood-based moves into proposal distributions over combinatorial solution spaces. This connection enables traditional non-differentiable optimisation procedures to act as learnable stochastic layers that can provide meaningful gradient estimates for end-to-end model training. The authors demonstrate the theoretical soundness of the framework and validate it empirically on binary vector prediction and a large-scale dynamic vehicle routing problem with time windows.

**Strengths:**

- The core idea of the paper is strong, as it successfully combines two well-established concepts, local search and Markov Chain Monte Carlo, into a coherent framework for differentiable combinatorial optimisation. As well, it is both general and flexible as it supports multiple neighborhood systems and extends beyond prior works.
- The work offers rigorous analysis and solid theoretical guarantees for the proposed framework.
- Despite the technical depth, the paper is clear, well structured, and easy to follow with good explanations and well-presented algorithms and propositions.

**Weaknesses:**

My main concerns are with the empirical section:
- The authors provide empirical results on two cases (dynamic vehicle routing and binary vector prediction), which was effective to show the strength of the claims but yet lack extra experiments to prove the generalisation of the framework to other combinatorial problems aside from routing.
- The evaluation comparison on dynamic vehicles tasks relies solely on the EURO Meets NeurIPS 2022 PC-HGS–based baseline, which, though strong, does limit the depth of the empirical evaluation.

**Questions:**

- As mentioned in the weaknesses section, the experiments primarily focus on binary vector prediction and dynamic vehicle routing. It would be helpful if the authors could discuss or test how the framework might generalise to other combinatorial problems.
- For dynamic vehicles routing, could the authors include additional or more recent differentiable combinatorial optimisation baselines?
- The paper mentions that meaningful gradients can be obtained even with a single MCMC iteration. Could the authors clarify how stable these gradients remain in practice when training end-to-end with few iterations?
- On Table 3, the proposed method performs better under shorter time budgets but converges to similar results as the baseline when given more time. Could the authors please provide more analysis or intuition on what drives this behaviour?

---

> ### Author Response · Authors · 2025-11-21
>
> We thank the reviewer for their positive assessment and for recognizing the "strong core idea," "general and flexible" framework, and "rigorous analysis." We address the concerns regarding empirical evaluation below.
>
> ## __Response to Weaknesses & Questions__
>
> ### __Generalization to other combinatorial problems__
>
> The reviewer noted that our initial experiments focused on routing and binary vector prediction (although the initial submission also contains experiments on k-subsets prediction in the appendix). To prove the generalization of our framework, we have added a new experiment on the Multi-Dimensional Knapsack Problem (MKP) (Section 5.2 and Appendix D in the revised submission).
>
> - __Context:__ MKP is a canonical NP-hard problem involving selecting a subset of items under multiple resource constraints.
> Implementation: We use specific neighborhood structures for MKP (bit swaps and bit flips) and plug them as proposal distributions into our LS-MCMC layer.
> - __Result:__ The method successfully learns to predict item values that minimize regret, demonstrating that the framework adapts to new problems simply by defining appropriate local search moves.
>
> ### __Additional baselines__
>
> In the new MKP experiment, we compared our method against a wide range of differentiable optimization baselines provided by the PyEPO library: SPO+, PFY, NID and NCE (details are in section 5.2 and appendix D of the revised submission).
>
> We hope this addresses the concern about limiting comparison to just PC-HGS. The results show that our method is competitive with these established baselines in terms of regret while offering superior scalability. We did not have time to add other baselines for the DVRPTW: indeed, the scale of this experiment in terms of code complexity (complex environment, Python / C++ interface for the solver…) makes the introduction of other baselines in the limited rebuttal period time difficult.
>
> ### __Stability of gradients with few iterations__
>
> The reviewer asks about gradient stability with single/few MCMC iterations.
> - __Theoretical Justification:__ In Section 4.3 (Proposition 3), we prove that even with a single MCMC iteration ($K=1$), the expected update is the exact gradient of a specific target-dependent Fenchel-Young loss. This means the training signal is a valid gradient for a specific regularized, convex objective.
> - __Empirical Evidence:__ In our dynamic vehicle routing experiment, the learning curve of the LS-MCMC approach with $K=1$ (which is averaged over 50 trainings with different random seeds for statistical significance) shows how smooth the training dynamics are. In our new MKP experiment, we measured the gradient variance (trace of the covariance matrix, estimated mini-batch-wise) throughout training. These new results show that the variance of LS-MCMC gradients is lower than most methods and decreases stably over time, confirming that the local search structure provides a low-variance, reliable signal even with limited iterations.
>
> ### __Performance vs. Time Budgets__
>
> The reviewer asks why our method converges to similar results as the baseline when given more time (Table 3 in original submission).
> The reason is likely that the layer used by the baseline (perturbed PC-HGS) has more powerful exploration capabilities than our Local Search layer, and thus is capable of providing better training signal when given significant time per datapoint. Indeed, the PC-HGS heuristic uses a mixture of Genetic Algorithms, Local Searches, and Large Neighborhood Searches to explore the solution space, while our layer is only Local Search-based.
>
> However, we argue that the objective of our framework is to construct a lightweight, stochastic layer that __generates meaningful gradients rapidly__, as this is the bottleneck to scale the training of ML models that incorporate combinatorial optimization layers. Once the model $g_W$ is trained, however, one can use any solver (including the full PC-HGS, for this particular routing application) to obtain the final solution, as we do in our experiments ($f_W := \tilde{y} \circ g_W$).
>
> Thus, the "inferior" performance at long runtimes reflects the intentional trade-off of a layer designed for efficient training, while the trained model remains compatible with powerful solvers at inference time.

---

> > ### Comment · Reviewer_SQPE · 2025-11-27
> >
> > Thank you for the detailed response. The clarification regarding gradient stability, along with the added knapsack experiment and broader baseline comparisons, resolves my main concerns about the evaluation depth. I have adjusted my rating accordingly.

---

### Meta-Review · Area_Chair_Ry9y · 2026-01-12

**Summary:**

This paper proposes integrating a differentiable, stochastic local search with neural network for optimization with uncertainty. It links local search to MCMC sampling and transforms neighborhood moves into proposal distributions, so that gradient estimates can be derived for end-to-end model training. Extensive theories were provided to show the validity of the method and experiments were conducted on binary vector prediction, dynamic vehicle routing problem, and multi-dimensional knapsack problem (added during rebuttal).

Authors' rebuttal addresses some concerns: 1) adding another multi-dimensional knapsack problem to show the generalization of method; 2) clarifying the performance in Table 3, usefulness of short-time performance and the stability of the LS-MCMC gradients.  However, some concerns remain: 1) more recent differentiable combinatorial optimisation baselines for DVRP are still missing, and only one perturbed optimizer (Berthet et al., 2020) was compared; 2) the performance is limited by local search made differentiable.

Considering the remaining concerns, including the missing baselines for DVRP, which is the most complex and important problem and the limitation of local search itself. I intend to reject this paper which is still not solid at the current version.

**Reviewer Concerns:**

Most reviewers mentioned the two problems in the original submission are not enough. The authors have addressed this comment by adding experiments with a multi-dimensional knapsack problem. Reviewers intended to raise scores by satisfying the new experiments. Authors also explained some questions such as the gradient stability, the potential application with other iterative combinatorial optimization algorithms. However, more baselines are missing for DVRP with time windows. This harms the empirical quality since DVRP is complex and critical to represent the performance and generalization. I also didn't see a reply to the limited performance of local search itself and any empirical results of applying stronger local search heuristics.

**Reviewer Scores:**

Two reviewers who originally gave a score of 4 indicated that they would raise their scores to 6. One reviewer maintained a score of 4. Another reviewer gave a score of 8 but expressed low confidence (the review was quite short and did not provide very useful comments).

---

### Decision · Program_Chairs · 2026-01-26

Reject